# Robust Regression of General ReLUs with Queries

**Ilias Diakonikolas**[*]
University of Wisconsin-Madison
ilias@cs.wisc.edu

**Daniel M. Kane**[*]
University of California, San Diego
dakane@cs.ucsd.edu

**Mingchen Ma**[*]
University of Wisconsin-Madison
mingchen@cs.wisc.edu

## Abstract

We study the task of agnostically learning general (as opposed to homogeneous) ReLUs under the Gaussian distribution with respect to the squared loss. In the passive learning setting, recent work gave a computationally efficient algorithm that uses $\mathrm{poly}(d, 1/\epsilon)$ labeled examples and outputs a hypothesis with error $O(\mathrm{opt}) + \epsilon$, where $\mathrm{opt}$ is the squared loss of the best fit ReLU. Here we focus on the interactive setting, where the learner has some form of query access to the labels of unlabeled examples. Our main result is the first computationally efficient learner that uses $d\,\mathrm{polylog}(1/\epsilon) + \tilde{O}(\min\{1/p, 1/\epsilon\})$ black-box label queries, where $p$ is the bias of the target function, and achieves error $O(\mathrm{opt}) + \epsilon$. We complement our algorithmic result by showing that its query complexity bound is qualitatively near-optimal, even ignoring computational constraints. Finally, we establish that query access is essentially necessary to improve on the label complexity of passive learning. Specifically, for pool-based active learning, any active learner requires $\tilde{\Omega}(d/\epsilon)$ labels, unless it draws a super-polynomial number of unlabeled examples.

## 1 Introduction

ReLU activations play a central role in the design of modern neural networks. A ReLU function $\sigma(W \cdot x - t) = \max\{W \cdot x - t, 0\}$ is specified by a pair of parameters $(W, t)$, where $W \in \mathbb{R}^d$ and $t \in \mathbb{R}$. ReLU regression is the following basic problem: given some form of access to a distribution $D$ over $\mathbb{R}^d \times \mathbb{R}$, output a ReLU with loss that can compete with $\mathrm{opt}$—the loss of the best-fit ReLU with respect to $D$. This fundamental task has been extensively studied in the past decades; see, e.g., [KKSK11, FCG20, DGK+20, VYS21, DKTZ22b, WZDD23, ATV22, GV24] and references therein. Prior algorithmic work has focused on learning from random examples (passive learning) and has obtained efficient learners with error $O(\mathrm{opt}) + \epsilon$, under the assumption that the marginal distribution $D_x$ is well-behaved; in most cases, a standard Gaussian or a structured distribution with similar properties. Such a benchmark is motivated by known computational hardness results for this problem. On the one hand, without any assumption on $D_x$, it is computationally intractable to achieve error $C \cdot \mathrm{opt}$ for any constant $C > 1$ [DKMR22]. On the other hand, even under the standard Gaussian, it is computationally hard to achieve loss $\mathrm{opt} + \epsilon$ [DKZ20, DKR23].

Despite the aforementioned long line of work on this problem, the number of *labeled* examples needed to achieve the desired error guarantee remains poorly understood. For the special case that the target ReLU has negative threshold (i.e., $t \leq 0$) [DKTZ22a, WZDD23] design efficient PAC learning algorithms achieving error $O(\mathrm{opt}) + \epsilon$ with a nearly optimal sample complexity of $d\,\mathrm{polylog}(1/\epsilon)$. The problem becomes much more challenging for *general* ReLUs (with arbitrary bias $t$) and in

---

[*]Equal Contribution

39th Conference on Neural Information Processing Systems (NeurIPS 2025).

particular for $t > 0$. Recently, [GV24, ZWDD25] gave the first efficient PAC learning algorithms that achieve $O(\text{opt}) + \epsilon$ error for general $t$, using $\text{poly}(d, 1/\epsilon)$ labeled examples. In fact, to learn an arbitrary ReLU to error $\epsilon$, even with clean labels, one needs $\tilde{\Theta}(d/\epsilon)$ labeled examples in the passive PAC setting (see, e.g., from Theorem 1.3). Hence, to obtain improved label complexity, one needs to consider stronger data access models that allow some form of adaptive (query) access to the labels.

Learning with queries and (pool-based) active learning are powerful models that can be used to reduce the number of labeled examples needed for various learning tasks. Such models capture the ability to perform experiments, or the availability of expert advice, and are well-motivated by real-worlds applications where unlabeled examples are cheap. A long line of work has shown that in such interactive settings one can significantly reduce the number of labeled examples needed for learning in a variety of settings [BBZ07, DKM05, BL13, ABL17, YZ17, She21, DMRT24, DKM24, KMT24a, GTX$^+$24, LT25].

The focus of this paper is on agnostic learning with black-box query access to the labels. Specifically, for $x \in \mathbb{R}^d$, we can make a black-box query for its label $y(x)$, where $y(x)$ is generated adversarially.

In the context of ReLU regression, in the noiseless setting, the examples we are actually using are those with non-zero labels. This implies that if we have one example with non-zero label, then we only need to make $\tilde{O}(d)$ queries in its neighborhood to collect $d$ examples with non-zero labels. This lead to an algorithm with label complexity of $O(1/p + d)$—beating the label complexity of passive learning. If $p = \Omega(1)$, existing agnostic learning algorithms robustify the above ideas and nearly match such a label complexity, even in the passive setting. However, if $p$ is small (corresponding to large $t > 0$) our understanding of the label complexity is limited. Specifically, there is a large gap in the label complexity of known algorithms for $t > 0$ versus $t < 0$. This discussion motivates the following question:

*What is the label complexity of the general agnostic ReLU regression problem?*

Our main result is the first efficient learning algorithm that solves the general agnostic ReLU regression problem with near-optimal label complexity (see Theorem 3.1 for the formal statement).

**Theorem 1.1.** *Consider the problem of general agnostic ReLU regression with Gaussian marginals. There is an algorithm that makes $M = d\,\text{polylog}(R^2/\epsilon) + \tilde{O}(\min\{1/p, R^2/\epsilon\})$ queries, runs in $\text{poly}(M)$ time, and outputs a ReLU function $h = \sigma(W \cdot x - t)$ such that with high probability, $h$ has error $O(\text{opt}) + \epsilon$. Here, $p$ is the fraction of examples that are labeled non-zero by the optimal ReLU and $R$ is the upper bound for $\|W^*\|$.*

Notably, our algorithmic approach solves not only the ReLU regression problem, but also develops new techniques that can be used to analyze a variety of generalized linear models.

Unlike pool-based active learning, where an algorithm first selects a pool of unlabeled examples $S$ from the marginal distribution $D_x$ and is allowed to query $y(x)$ for $x \in S$, our algorithm makes use of a stronger oracle that is allowed to query *any* desired point in $\mathbb{R}^d$.

To complement our upper bound, we show that the query complexity of our algorithm is information-theoretically nearly optimal (see Theorem 4.1 for the formal statement).

**Theorem 1.2.** *Consider the problem of general agnostic ReLU regression with Gaussian marginals. Suppose that the optimal ReLU has bias $p$ and $\|W^*\| = 1$. Any learning algorithm that learns a hypothesis $\hat{h}$ with error $\tilde{O}(p)$ and succeeds with probability $1/3$ must make $\tilde{\Omega}(1/p^{1-o_d(1)} + d)$ queries, even if $\text{opt} \ll p$.*

Our final lower bound shows that unless the unlabeled dataset is extremely large, no pool-based active learning algorithm is able to achieve the label complexity of our algorithm, even in the realizable setting. This establishes a sharp separation between pool-based active regression and query learning, and resolves the label complexity of ReLU regression (see Theorem 4.2 for the formal statement) .

**Theorem 1.3.** *Consider the problem of general (realizable) ReLU regression with Gaussian marginal. Suppose the optimal ReLU has bias $p$. Any pool-based active learning algorithm that learns a hypothesis $\hat{h}$ with error $\tilde{O}(p)$ from a pool of $m$ unlabled examples drawn from $N(0, I)$ and succeeds with probability $1/3$ must make $\tilde{\Omega}(d/(p \log(m)))$ queries.*

**Preliminaries and Notation**   Here we record the problem definition and basic notation.

**Definition 1.4** (Agnostic ReLU Regression with Queries). *Let $\sigma(z) = \max\{z, 0\}$ be the ReLU function. A labeling function $y(x) : \mathbb{R}^d \to \mathbb{R}$ is a random function that maps each $x \in \mathbb{R}^d$ to an unknown real-valued random variable. For each $h : \mathbb{R}^d \to \mathbb{R}$, denote by $\mathrm{err}(h) = \mathbf{E}_{x \sim N(0,I)} (h(x) - y(x))^2 / 2$, $\mathrm{opt} := \min_{W : \|W\| \leq R, t \geq 0} \mathrm{err}(\sigma(W \cdot x - t))$ and $\sigma^*(x) = \sigma(W^* \cdot x - t^*)$ be any ReLU with error $\mathrm{opt}$. A query takes $x \in \mathbb{R}^d$ as an input and returns a label $y \sim y(x)$. We say that a learning algorithm $\mathcal{A}$ is a constant-factor approximate learner if for every labeling function $y(x)$, and for every $\epsilon, \delta \in (0, 1)$, it outputs some hypothesis $\hat{h} : \mathbb{R}^d \to \mathbb{R}$ by adaptively making queries, such that with probability at least $1 - \delta$, $\mathrm{err}(\hat{h}) \leq O(\mathrm{opt}) + \epsilon$. The query complexity of $\mathcal{A}$ is the total number of queries it uses during the learning process.*

Furthermore, we will without loss of generality assume $\mathrm{opt} \leq \epsilon$, since the final error guarantee is $O(\mathrm{opt} + \epsilon)$. We remark that in some parts of the paper, we will also consider the case where $\sigma$ is not a ReLU but a general function and $W^* \in \mathbb{S}^{d-1}$ In that case, we call the problem agnostic learning (spherical) generalized linear model (GLM) and for a hypothesis $\sigma(w \cdot x)$, we use $\mathrm{err}(w)$ to denote its error accordingly. For a vector $W$ in $\mathbb{R}^d$, we use $\|W\|$ to denote it $\ell_2$ norm and use the lower case $w$ to denote its direction $W / \|W\|$. For a one-dimensional standard normal $z \sim N(0, 1)$ and $t \geq 0$, we denote by $\Phi(t) := \mathbf{Pr}_z(z > t)$ and $\psi(t)$ the value of the density function of $z$ at $t$. For a ReLU activation $\sigma(z - t)$, we denote by $V(t) := \mathbf{E}_{z \sim N(0,1)} \sigma^2(z - t)$ its second moment and call $p = \Phi(t)$ its bias. For a real-valued function $f : \mathbb{R} \to \mathbb{R}$, we denote by $\|f\|_2^2 := \mathbf{E}_{z \sim N(0,1)} f^2(z)$ its squared $L_2$ norm in Gaussian space and for $a \in [0, 1]$, denote by $T_a f(z) := \mathbf{E}_{s \sim N(0,1)} f(az + \sqrt{1 - a^2}s)$.

**Organization of the Paper** In Section 2, we consider a special case of the problem where $t^*$ is given and the optimal vector $W^*$ is restricted over $\mathbb{S}^{d-1}$. We propose a framework showing that a simple projected gradient-descent method converges to a desired solution for learning spherical GLMs, provided a warm-start. As an application, we give an efficient algorithm with optimal query complexity for ReLU regression when $\|W^*\|, t^*$ are given. In Section 3, we tackle the general ReLU regression problem, by discussing the technical difficulties and presenting key components of the problem, combining with the framework of Section 2. In Section 4, we establish our query complexity lower bounds. Due to space limitations, we defer several proofs to the supplementary material.

## 2 Warm Up: Robustly Learning Spherical GLM

As a warm up, we consider the special setting where $\|W^*\| = 1$ and $t^* > 0$ are known. Prior works in the passive setting, such as [GV24, ZWDD25], reduce the problem to this case by guessing $(\|W^*\|, t^*)$ in a brute force way. There are two motivations for studying such a setting. First, as we will discuss in Section 3, although such a special case does not capture the main difficulty of obtaining query-optimal algorithms, it provides important technical components to achieve this goal. Second, when $(\|W^*\|, t^*)$ are known, the problem becomes a special case of agnostic learning of spherical GLMs, where $\sigma$ is a general activation function and is known *precisely* to the learner. In this section, we start by analyzing the task of agnostic learning spherical GLMs in the passive learning setting. We presenting several technical tools that we develop to solve the ReLU regression problem in this special case with an optimal query complexity. Unlike prior works, such as [GV24], which designed a complicated update rule for $w$ (the current direction), we instead focus on the following simple projected gradient descent method:

$$w \leftarrow \mathrm{proj}_{\mathbb{S}^{d-1}} \left( w - \mu \mathrm{proj}_{w^\perp} \nabla_w \mathrm{err}(w) \right). \tag{1}$$

We summarize our main technical contribution in this section informally as follows.

**Theorem 2.1.** *Let $\sigma : \mathbb{R} \to \mathbb{R}$ be any activation function such that $\|\sigma'\|_2^2 = L$, $L > 0$. Let $D$ be any distribution over $\mathbb{R}^d \times \mathbb{R}$ such that $D_x = N(0, I)$ and $\mathrm{err}(w^*) \leq \epsilon$, where $w^* \in \mathbb{S}^{d-1}$ is a direction that achieves the optimal loss. For $\alpha > 1$, suppose we are given any unit vector $w^0 \in \mathbb{S}^{d-1}$ such that $\left\| T_{\sqrt{\cos \theta_0}} \sigma' \right\|_2^2 \geq \|\sigma'\|_2^2 / \alpha$, where $\theta_0 = \theta(w^0, w^*)$. Starting from $w^0$, the update rule (1) gives a direction $\hat{w}$ with error $O(\alpha^2 \epsilon)$.*

Roughly speaking, we show that the update rule (1) has an *initialization-dependent* error guarantee, which holds for even very complicated activation functions that are non-monotone. The quality of the initialization is measured by the ratio $\alpha := \|\sigma'\|_2^2 / \left\| T_{\sqrt{\cos \theta_0}} \sigma' \right\|_2^2$. As we will show later, for general ReLU activations, we are able to efficiently get a $w^0$ with $\alpha = O(1)$, which implies a solution

with error $O(\epsilon)$ if we can implement (1) with high accuracy. Due to space limitations, we defer the technical details and proofs of this section to Appendix B. To analyze the error guarantee, we start with the following simple observation:

$$\text{err}(w) \leq \mathbf{E}_x \left(\sigma(w \cdot x) - \sigma(w^* \cdot x)\right)^2 + \mathbf{E}_x \left(\sigma(w^* \cdot x) - y\right)^2 \leq 2\text{opt} + \mathbf{E}_x \left(\sigma(w \cdot x) - \sigma(w^* \cdot x)\right)^2.$$

So, the central part of the analysis relies on characterizing the noiseless error of a $\sigma(w \cdot x)$, $\ell(w) := \mathbf{E}_{x \sim N(0,I)} \left(\sigma(w \cdot x) - \sigma(w^* \cdot x)\right)^2 / 2$. Prior works usually analyzed $\ell(w)$ via the angle $\theta(w, w^*)$. For example, for the problem of learning homogeneous halfspaces, $\ell(w) = \theta/\pi$. However, in general, $\ell(w)$ does not have such a simple closed-form. In the following lemma, we give an integral expression for $\ell(w)$ over the unit sphere by drawing a connection with the Ornstein–Uhlenbeck semi-group.

**Lemma 2.2** (Noiseless Error Estimation over the Sphere). *Let $\sigma : \mathbb{R} \to \mathbb{R}$ be any activation function such that $\sigma' \in L_2(N(0, I))$ and let $w \in \mathbb{S}^{d-1}$ be any unit vector such that $\theta := \theta(w, w^*) < \pi/2$. Then $\ell(w) = \int_0^\theta \sin s \left\|T_{\sqrt{\cos s}}\sigma'\right\|_2^2 ds \leq (\pi/2)\sin^2\theta \left\|\sigma'\right\|_2^2$.*

To relate Lemma 2.2 with the update rule (1), we use the following two structural lemmas that characterize the progress as well as the noise level during the update in each round.

**Lemma 2.3.** *Let $\sigma : \mathbb{R} \to \mathbb{R}$ be any activation function such that $\sigma' \in L_2(N(0, I))$ and let $w \in \mathbb{S}^{d-1}$ be any unit vector such that $\theta = \theta(w, w^*) < \pi/2$, where $\text{err}(\sigma(w^*)) = \text{opt}$. Write $w^* = aw + bu$, where $u \in \mathbb{S}^{d-1}, u \perp w, a, b \geq 0, a^2 + b^2 = 1$, then, $\text{proj}_{w^\perp} \nabla_w \ell(w) = -b \left\|T_{\sqrt{a}}\sigma'\right\|_2^2 u$.*

**Lemma 2.4.** *Let $\sigma : \mathbb{R} \to \mathbb{R}$ be any activation function such that $\sigma' \in L_2(N(0, I))$ and let $w \in \mathbb{S}^{d-1}$ be any unit vector such that $\theta = \theta(w, w^*) < \pi/2$, where $\text{err}(\sigma(w^*)) = \text{opt} \leq \epsilon$. Then for any $v \in \mathbb{S}^{d-1}$ and $v \perp w$, $|\text{proj}_{w^\perp}(\nabla_w \text{err}(w) - \nabla_w \ell(w)) \cdot v| \leq \sqrt{\epsilon}\|\sigma'\|_2$.*

The intuition here is that in each update round, if the length of the gradient $b\left\|T_{\sqrt{a}}\sigma'\right\|_2^2$ is larger than the noise level $\sqrt{\epsilon}\|\sigma'\|_2$, then the projected gradient descent approach is able to improve the angle between $w$ and $w^*$. We summarize this as Lemma 2.5.

**Lemma 2.5** (Angle Contraction). *Let $w^*, w^{(i)} \in \mathbb{S}^{d-1}$ such that $w^* = aw^{(i)} + bu$, where $u \in \mathbb{S}^{d-1}, u \perp w^{(i)}, a, b \geq 0, a^2 + b^2 = 1$. Let $\theta_i = \theta(w^{(i)}, w^*)$. Let $G \in \mathbb{R}^d$ be a random vector such that with probability 1, $G \perp w^{(i)}$. Let $g$ be the mean of $G$ and $\hat{g} \in \mathbb{R}^d$. Suppose there is some $c > 0$ such that $g \cdot u \geq cb/10, \|g\| \leq cb, \|g - \hat{g}\| \leq bc/40$, then by setting $\mu = c/20$, the update rule $w^{(i+1)} = \text{proj}_{\mathbb{S}^{d-1}}(w^{(i)} + \mu\hat{g})$ satisfies $\sin(\theta_{i+1}/2) \leq \sqrt{1 - \left(\frac{c}{20}\right)^2}\sin(\theta_i/2)$.*

However, the length of the signal $\sin\theta\left\|T_{\sqrt{\cos\theta}}\sigma'\right\|_2^2$ in general is not an increasing function of $\theta$. If $\theta_0$ is not small enough, the noise level could be too high to make the gradient point in the correct direction, making it impossible to reach a desirable error guarantee. On the other hand, by Lemma 2.2, if $\ell(w) \geq \Omega(\alpha^2\epsilon)$, then the noise level $|\text{proj}_{w^\perp}(\nabla_w \text{err}(w) - \nabla_w \ell(w)) \cdot v|$ is at most $\sin\theta\alpha\|\sigma'\|_2^2$. Since $\left\|T_{\sqrt{\cos\theta}}\sigma'\right\|$ is decreasing in $\theta$, this implies that, as long as we have a good initialization, we can make progress and reach a good solution. We summarize this property as Lemma 2.6.

**Lemma 2.6.** *Let $\sigma : \mathbb{R} \to \mathbb{R}$ be any activation function such that $\sigma' \in L_2(N(0, I))$. Let $\alpha > 1$ and $0 < \theta_0 < \pi/2$ such that $\left\|T_{\sqrt{\cos\theta_0}}\sigma'\right\|_2^2 \geq \|\sigma'\|_2^2/\alpha$. Let $w \in \mathbb{S}^{d-1}$ be any unit vector such that $\theta = \theta(w, w^*) < \theta_0$, where $\text{err}(\sigma(w^*)) = \text{opt} \leq \epsilon$. If $\sin^2\theta\|\sigma'\|_2^2 \geq 20\alpha^2\epsilon/\pi$, then for any $v \in \mathbb{S}^{d-1}$ and $v \perp w$, $\|\text{proj}_{w^\perp}(\nabla_w \text{err}(w) - \nabla_w \ell(w))\| \leq \|\text{proj}_{w^\perp}\nabla_w \ell(w)\|/20$. Furthermore, if $\|\text{proj}_{w^\perp}(\nabla_w \text{err}(w) - \nabla_w \ell(w))\| > \|\text{proj}_{w^\perp}\nabla_w \ell(w)\|/20$, then $\text{err}(w) \leq O(\alpha^2\epsilon)$.*

## 2.1 Application: Query Complexity of Agnostic Learning Spherical ReLU with Known Bias

As a direct application of the above framework, we show when $w^* \in \mathbb{S}^{d-1}$ and $t^*$ is given, how to solve the general ReLU regression problem with an optimal query complexity. As discussed above, to ensure we get a solution with error $O(\epsilon)$, we need some $w^0$ with $\|\sigma'\|_2^2 / \left\|T_{\sqrt{\cos\theta_0}}\sigma'\right\|_2^2 = O(1)$. To characterize such a $\theta_0$, we present the following structural lemma.

**Lemma 2.7.** *Let $\sigma$ be an activation function of the form $\sigma(z) = \text{Relu}(z - t^*)$, where $t^* > 0$. If $\sin\theta/2 \leq 1/t^*$, then $\left\|T_{\sqrt{\cos\theta}}\sigma'\right\|_2^2 \geq \|\sigma'\|_2^2/50$.*

This implies that $\theta_0 = O(1/t^*)$ is sufficient for us to reach some $w$ with $O(\epsilon)$ error. But to obtain such a $w^{(0)}$ is also a challenging problem. Our initialization is motivated by the following lemma.

**Lemma 2.8.** *Let $\sigma$ be an activation f the form $\sigma(z) = \mathrm{Relu}(z - t^*)$, where $t^* > 0$. Let $y(x)$ be any labeling function such that $\mathrm{opt} \leq \epsilon$. Let $c > 0$ be a suitably small constant. If $V(t^*) > C\epsilon$, for some large constant $C$, then $\mathbf{Pr}_{x \sim N(0,I)}\left(\bar{y}(x) \neq \mathrm{sign}(w^* \cdot x - t^*)\right) \leq \Phi(t^*)/C'$ for some large constant $C' > 0$, where $\bar{y}(x) := \mathbb{1}\{y(x) > c/(t^*)\}$.*

That is to say, as long as $t^*$ is not too large to make $h \equiv 0$ have error $O(\epsilon)$, the truncated label $\bar{y}(x) := \mathbb{1}\{y(x) > c/t^*\}$ can be seen as a labeling generated by the halfspace $h(x) = \mathrm{sign}(w^* \cdot x - t^*)$ corrupted with $\eta$-level adversarial label noise, such that $\eta/\Phi(t^*) < 1/C$ for some sufficiently large constant $C$. Such an observation is useful, as it allows us to make use of the recent technique developed in [DKM24] to obtain the warm-start using only $\tilde{O}(1/p + d)$ queries.

**Lemma 2.9.** *[Halfspace Initialization via Queries (Theorem 3.8, Theorem F.1 [DKM24])] Let $h^*(x) = \mathrm{sign}(w^* \cdot x - t^*)$, where $t^* \leq O(\sqrt{\log(1/\epsilon)})$ and $y(x) : \mathbb{R}^d \to \pm 1$ be any labeling function such that $\mathbf{Pr}_{x \sim N(0,I)}(h^*(x) \neq y(x)) \leq \Phi(t^*)/C'$ for some large enough constant $C'$. There is an algorithm such that given some $t \in \mathbb{R}$ with $|t - t^*| \leq 1/\log(1/\epsilon)$, it makes $M = \tilde{O}(1/\Phi(t) + d\log(1/\epsilon))$ queries, runs in $\mathrm{poly}(d, M)$ time, and with probability at least $1/\log(1/\Phi(t))$, outputs some $w^{(0)}$ such that $\sin(\theta(w^{(0)}, w^*)/2) \leq \min\{1/t, 1/2\}$.*

However, obtaining such a $w^{(0)}$ is not enough for us to solve the problem with a small label complexity for the following reason: Lemma 2.6 does not bound the number of iterations and the number of labeled examples needed to reach a good solution. Specifically, by Lemma 2.5, the progress made in each round is characterized by the length of the signal $g \cdot u$. If $g \cdot u \approx c \sin \theta$, then the angle decreases by a factor of $(1 - c)$ in each round. If $t^* = 0$, prior works have shown that $c = \Omega(1)$ and $\tilde{O}(d)$ labeled examples are enough to get a good estimation of $g$. The landscape changes dramatically when $t^*$ becomes large. As indicated by Lemma 2.3, for large $t^*$, the length of the gradient is a most $\|\sigma'\|_2^2$, which can be as small as $\mathrm{poly}(\epsilon)$. This implies that, unless we estimate the gradient to very high accuracy and rescale the gradient to a constant length, the progress made in each round is too small. Unfortunately, as the variance of the gradient is much larger than the accuracy we need, this blows up the total query complexity. To overcome this difficulty, we show that for the problem of ReLU regression, the initialization guarantee $\theta_0 \leq O(1/t^*)$ not only lets us make progress in each round, but it also allows us to use queries to boost the length of the gradient while maintain a small variance. Intuitively, every time we estimate the gradient, examples with $w^* \cdot x > t^*$ contribute most of the gradient. When our current $w^{(i)}$ is close to $w^*$, the regions $\{x \mid w^{(i)} \cdot x > t^*\}$ and $\{x \mid w^* \cdot x > t^*\}$ have significant intersection. Thus, we can boost the length of the gradient by querying examples in $\{x \mid w^{(i)} \cdot x > t^*\}$. Furthermore, due to the Lipschitz continuity of $\mathrm{Relu}$, we can maintain small variance for the gradient; and thus $\tilde{O}(d)$ examples suffice for us to accurately estimate the gradient.

**Lemma 2.10.** *Let $\sigma(z) = \mathrm{Relu}(z - t^*)$, with $t^* > 0$. Let $y(x)$ be any labeling function such that $\mathrm{opt} \leq \epsilon$. Let $w \in \mathbb{S}^{d-1}$ be any vector such that $\sin(\theta/2) \leq 1/t^*$. Denote by $G^* \in \mathbb{R}^d$ the random vector $(\sigma(w \cdot x - t^*) - \sigma(w^* \cdot x - t^*)) \mathrm{proj}_{w^\perp} x$ and $G$ the random vector $(\sigma(w \cdot x - t^*) - y(x)) \mathrm{proj}_{w^\perp} x$, where $x \sim N(0,I)\mid_{\{x|w \cdot x > t^*\}}$. Then the following holds: (1) $\mathbf{E}\, G^* = b \left\| T_{\sqrt{a}} \sigma' \right\|_2^2 u/\Phi(t^*)$; (2) $|(\mathbf{E}\, G^* - \mathbf{E}\, G) \cdot v| \leq \sqrt{\epsilon} \|\sigma'\|_2 /\Phi(t^*)$' (3) If $\sin^2 \theta \Phi(t^*) > \epsilon$, then $\mathbf{E}(G \cdot v)^2 \leq \tilde{O}(b^2), \forall v \in \mathbb{S}^{d-1}$.*

The structural results we obtained so far are almost all we need to get a query-optimal algorithm for the spherical case. We refer the reader to Appendix B for the detailed proof. A remaining caveat is that the initialization algorithm for halfspace learning used in [DKM24] only succeeds with $1/\log(1/\epsilon)$ probability in the worst case. This implies that to succeed with a good probability, we need to run the same algorithm $\mathrm{poly} \log(1/\epsilon)$ times, which will give us a list of $\mathrm{poly} \log(1/\epsilon)$ candidate hypotheses. The usual way to find a desired hypothesis from the list is to check their empirical errors via labeled examples. However, this approach will take $\Omega(1/\epsilon^2)$ labeled examples, blowing up the query complexity we have achieved so far. To avoid this, we design a new active testing procedure that only takes $\mathrm{poly} \log(1/\epsilon)$ queries and selects a hypothesis with error $O(\epsilon)$. Our procedure not only works for selecting ReLU activations, but also applies for much general settings.

**Lemma 2.11** (Hypothesis Selection with Queries). *Let $D$ be a distribution over $\mathbb{R}^d \times \mathbb{R}$ and let $D_x$ be the marginal distribution of $D_x$. There is an algorithm that, on input a list of hypotheses*

$h_1, \dots, h_k$ *such that for* $i \in [k]$, $h_i : \mathbb{R}^d \to \mathbb{R}$, $\mathbf{E}_{x \sim D} h_i^2(x)$ *exists, it makes* $\mathrm{poly}(k)$ *queries and returns a hypothesis* $\hat{h}$ *such that* $\mathrm{err}(\hat{h}) \le O(\min_{i \in [k]} \mathbf{E}_{(x,y) \sim D}(y - h_i(x))^2)$.

The intuition of our hypothesis selection algorithm is as follows. To simplify the intuition, we assume $\|h_1\| = \|h_2\|$. For each pair $(i, j)$, we would like to check whether $\|y - h_i\|_2 \le \|y - h_j\|_2$, which is equivalent to checking the sign of the correlation $\mathbf{E}[y(h_i(x) - h_j(x))]$. Naively, this can be done by randomly querying $x \sim D_x$; but since the variance of $y(h_i(x) - h_j(x))$ can be very large, this can blow up the query complexity of the algorithm. Instead, we reweight $D_x$ according to $(h_i(x) - h_j(x))^2$ and use $\tilde{y}(x) \propto y/(h_i(x) - h_j(x))^2$ as our query value. Such a modification keeps the mean we are interested in, but reduces its variance, allowing us to use $\tilde{O}(1)$ queries to solve the task. We defer details of the hypothesis selection algorithm to Appendix B.9.

# 3 Agnostic Learning an Arbitrary ReLU with Optimal Query Complexity

To handle the general case, where $\|W^*\|$ and $t^*$ are unknown, we need to overcome several conceptual and technical difficulties. Prior works in the passive setting solve the general version of the problem by reducing it to the spherical case, by guessing $\|W^*\|$ and $t^*$ up to $\mathrm{poly}(\epsilon/R)$ additive error, running the same algorithm $\mathrm{poly}(R/\epsilon)$ times and doing a hypothesis testing. Unfortunately, this simple approach is prohibitive in our context because it completely ruins the query complexity (even though we are able to solve the spherical setting with optimal query complexity). To achieve an optimal query complexity for the general problem, there are two main obstacles to overcome.

The first one is to find the correct way to update parameters. In the case where $t^* \le 0$, prior works avoid guessing $\|W^*\|$ by replacing the update rule (1) by a standard gradient descent $W \leftarrow W - \mu \nabla_W \mathrm{err}(W)$, and showing that when $\|W - W^*\|$ is large enough to make the noiseless error larger than $\Omega(\epsilon)$, $\nabla_W \mathrm{err}(W) \cdot (W - W^*) \ge \Omega(\|W - W^*\|^2)$; this leads to a constant factor of decay in $\|W - W^*\|$. However, when there is a large threshold, $\|W - W^*\|$ is not the correct quantity that characterizes the noiseless error. Consider the optimal hypothesis $h^* = \sigma(w^* \cdot x - t^*)$ and two other hypotheses $h_1 = \sigma(v \cdot x - t^*)$, $h_2 = (t^*\xi + 1)\sigma(w^* \cdot x - t^*)$, where $\xi > 0$ and $\sin \theta(v, w^*) \approx \xi$. When $t^*$ is large, the parameter distance of $h_2$ is much larger than that of $h_1$; however, it can be checked that the two functions have the same noiseless error, and thus adding the same level of noise can make the two functions indistinguishable. The implication of this phenomenon is that even if the parameter distance $\|W - W^*\|$ is large, we are not able to guarantee the noise rate is small enough to make the fast decay happen.

The second obstacle is how to implement the correct update with a small query complexity. In the spherical setting, we make use of the fact that $\sigma(w^{(i)} \cdot x - t^*) - y(x)$ is small for most queries $x$ to make the variance of the gradient as small as $\sin^2 \theta_i$ (which matches the length of the gradient). The small variance is the key that makes $\tilde{O}(d)$ queries sufficient to improve $w^{(i)}$. However, since $r^*, t^*$ are not part of the input, an inaccurate learned parameter $(r^{(i)}, t^{(i)})$ could make $h(x) - y(x)$ very large, making it impossible to estimate the gradient accurately with few queries. So, when we do the parameter update, the statistics we rely on must have small variance throughout the learning process.

We will require the following notation. For $r > 0, w \in \mathbb{S}^{d-1}, t > 0$, we define hypothesis $h(r, w, t) = \sigma(rw \cdot x - t)$. In particular, we write the optimal hypothesis as $h^* = \sigma(r^*w^* \cdot x - t^*)$. We denote by $\bar{t} := t/r$ the normalized threshold of a ReLU and define the noiseless error of $h(r, w, t)$ as $\ell(r, \bar{w}, t) = \frac{1}{2} \mathbf{E}_{x \sim N(0,I)} \left( \sigma(rw \cdot x - t) - \sigma(r^*w^* \cdot x - t^*) \right)^2$. Our main algorithmic result is an efficient learning algorithm with an optimal query complexity. Due to space limitations, we defer some technical details and proofs from this section to Appendix C.

**Theorem 3.1.** *Consider the problem of agnostic general ReLU regression with queries under the Gaussian distribution. There is an algorithm such that for every labeling function* $y(x)$ *and for every* $\epsilon, \delta \in (0, 1)$, *it makes* $M = \tilde{O}_\delta(\min\{1/p, R^2/\epsilon\} + d \cdot \mathrm{polylog}(R^2/\epsilon))$ *queries, runs in* $\mathrm{poly}(d, M)$ *time, where* $p = \Phi(\bar{t}^*)$ *is the bias of the optimal activation function, and outputs an* $\hat{h}$ *such that with high probability at least* $1 - \delta$, $\mathrm{err}(\hat{h}) \le O(\mathrm{opt}) + \epsilon$.

We remark that the dependence on $R^2$ is due to the natural of rescaling of the error parameter. Similar to the spherical setting, we still need a warm start in order to converge to a good solution. To maintain a truncated label to implement Lemma 2.9, some information about $(r^*, t^*)$ is needed. Here, we will grid $r \in [0, R]$ and $\bar{t} \in [0, O(\sqrt{\log(R^2/\epsilon)})]$, the normalized threshold, to get the initial information. However, instead of using a grid of size $\mathrm{poly}(R/\epsilon)$, we maintain a grid of

size poly $\log(R/\epsilon)$, exponentially smaller than the grid-size used in all prior works. In particular, for parameter $\bar{t}$, we will set $t^{(i)} = (i-1)/\text{polylog}(R/\epsilon), i = 1, \dots, O(\text{polylog}(R/\epsilon))$, while for parameter $r$, we build a two-level non-uniform grid as follows. The first level of the grid is defined as $r_i = 2^{i-1}\epsilon, i = 1, \dots, \log(R^2/\epsilon)$. For each interval $[r_i, r_{i+1}]$, we grid it uniformly into $r_{ij} = r_i + (j-1)r_i/\text{polylog}(R/\epsilon), j = 1, \dots, \text{polylog}(R/\epsilon)$. Such a grid can ensure that one of the grid points $(r, t)$ satisfies $r \le r^* \le 2r$ and $|\bar{t} - \bar{t}^*| \le \text{polylog}(R/\epsilon)$. We show in Lemma 3.2 that such a pair $(r, r\bar{t})$ suffices for us to get an initial direction $w^{(0)}$ as a warm start.

**Lemma 3.2** (Initialization with Raw Knowledge). *Let $h^* = r^*\sigma(\bar{w^*}\cdot x - \bar{t}^*)$ be the optimal hypothesis. Assuming that $(r^*)^2 V(\bar{t}^*) \ge \Omega(\epsilon)$, there is an algorithm such that given parameters $r, t > 0$ with $r \le r^* \le 2r$ and $|t - \bar{t}^*| \le 1/\log(R^2/\epsilon)$, it makes $M = \tilde{O}(1/p + d\log(R^2/\epsilon))$ queries, runs in $\text{poly}(d, M)$ time, and with probability at least $1/\log(1/p)$, outputs some $w^{(0)} \in \mathbb{S}^{d-1}$ such that $\sin(\theta(w^{(0)}, w^*)/2) \le \min\{1/\bar{t}^*, 1/2\}$.*

Although the first-level grid $r_i$ is enough for us to get a warm start $w^{(0)}$, for technical reasons, to obtain an algorithm with an optimal query complexity, it turns out that we need better knowledge about $r^*$. This is the reason why we use a two-level partition. In the rest of the proof, we assume that we have $(r_0, w^{(0)}, t_0)$ such that $|r_0 - r^*| \le r^*/\text{polylog}(R^2/\epsilon), \sin(\theta_0/2) \le 1/\bar{t}^*$ and $|\bar{t}_0 - \bar{t}^*| \le \text{polylog}(R^2/\epsilon)$. We next provide an overview of our algorithm based on this warm start $(r_0, w^{(0)}, t_0)$. As we mentioned earlier, since $\|W - W^*\|$ is not the correct measure for the noiseless error, the noise rate could be large even if $\|W - W^*\|$ is large. To overcome this issue, in each round of the algorithm, we modify the standard gradient descent by decomposing the update into two directions; along direction $w^{(i)}$ and orthogonal to $w^{(i)}$. In other words, we use different statistics to update $(r_i, t_i)$ and $w^{(i)}$ separately in a careful way. The motivation for using such a strategy is due to the following noiseless error decomposition, Lemma 3.3, which implies that the noiseless error can be decomposed into two terms that are independent of each other. Importantly, as long as one of the two terms is suboptimal, we are able to improve it despite the noise from the other direction.

**Lemma 3.3.** *Let $r > 0, w \in \mathbb{S}^{d-1}, t > 0$. Then $\ell(r, w, t) \le (r^*)^2 \int_0^\theta \sin s \left\| T_{\sqrt{\cos s}}\sigma'(z - \bar{t}^*) \right\|_2^2 ds + \mathbf{E}_{z \sim N(0,1)} \left(\sigma(rz - t) - \sigma(r^*z - t^*)\right)^2 .$*

Furthermore, the errors from $(r_i, t_i), w^{(i)}$ can be entangled with each other, which makes the analysis subtle. In particular, the error from one term can make the statistics we use for updating the other terms have a large variance, which could blow-up the query complexity we need. The second motivation of such a strategy is that it provides a clean way to analyze the error entanglement between the two terms. We list the algorithm below as Algorithm 1 and describe the update methods we use. We defer the details of analysis to Appendix C.3.

**Angle Update:** Similar to the spherical setting, we construct the gradient by boosting $\text{proj}_{w^\perp} \nabla_w \text{err}(w)$ via a rejection sampling approach. Given a ReLU activation, $h(r_i, w^{(i)}, t_i)$, define random vectors $G_i^* := \left(h(r_i, w^{(i)}, t_i)(x) - h^*(x)\right) \text{proj}_{(w^{(i)})^\perp}(x)$ and its noisy version $G_i := \left(h(r_i, w^{(i)}, t_i)(x) - y(x)\right) \text{proj}_{(w^{(i)})^\perp}(x)$, where $x \sim N(0, I) \mid_{w^{(i)}\cdot x > \bar{t}_i}$.

To begin with, we present the following lemma that quantifies $G_i$.

**Lemma 3.4.** *Let $h(r_i, \bar{w}^{(i)}, t_i)$. Write $w^* = aw^{(i)} + bu$, where $a, b > 0, a^2 + b^2 = 1$, $u \in \mathbb{S}^{d-1}, u \perp \bar{w}^{(i)}$. Then the following holds: (1) If $|\bar{t}_i - \bar{t}^*| \le 1/\log(R^2/\epsilon)$ and $b \le 1/\bar{t}_i$, then $\mathbf{E}\,G_i^* = -\alpha br^* \left\|T_a\sigma'(z - \bar{t}_i)\right\|^2 u/\Phi(\bar{t}_i)$, where $1/2 < \alpha < 2$. (2) $|(\mathbf{E}\,G^* - \mathbf{E}\,G)\cdot v| \le \sqrt{\epsilon}\,\|\sigma'(z - t_i)\|_2 /\Phi(\bar{t}_i), \forall v \in \mathbb{S}^{d-1}, v \perp w^{(i)}$.*

Lemma 3.4 implies that the error from $(r_i, t_i)$ does not affect the direction of the update $\mathbf{E}\,G$. Combining Lemma 2.2 and Lemma 2.7, it follows that as long as $\theta_i$ is large and contributes $\Omega(\epsilon)$ to the noiseless error, the gradient $G_i$ we construct can improve the angle. However, this does not imply that $\mathbf{E}\,G_i$ can be estimated with only a few queries. To that end, we use the following lemma that quantifies the variance of $G_i$.

**Lemma 3.5.** *Let $h(r_i, \bar{w}^{(i)}, t_i)$ be a ReLU activation. Write $w^* = aw^{(i)} + bu$, where $a, b > 0, a^2 + b^2 = 1, u \in \mathbb{S}^{d-1}, u \perp \bar{w}^{(i)}$. If $|\bar{t}_i - \bar{t}^*| \le 1/\log(R^2/\epsilon), C\epsilon/((r^*)^2\Phi(\bar{t}^*)) \le b^2 \le 1/t_i^2$, then $\mathbf{E}(G_i \cdot v)^2 \le \tilde{O}\left(\mathbf{E}_{z \sim N(0,I)} \left(\sigma(r_iz - t_i) - \sigma(r^*z - t^*)\right)^2 /\Phi(\bar{t}_i) + (r^*)^2b^2\right).$*

Unlike the spherical setting, the variance of $G_i$ depends on the accuracy of $(r_i, t_i)$. In particular, by Lemma 3.3, the contribution to the variance from $(r_i, t_i)$ is proportional to its contribution to

---
**Algorithm 1** QUERYLEARNING(Learn optimal ReLU with a warm start)
---
1: **Input:** $w^{(0)} \in \mathbb{S}^{d-1}$ : unit vector such that $\theta_0 \leq 1/\text{polylog}(R^2/\epsilon)$. $r_0 > 0$ : such that $|r^* - r_0| \leq r^*/\text{polylog}(R^2/\epsilon)$, $t_0 : |t_0 - t^*| \leq \text{polylog}(R^2/\epsilon)$.

2: **Output:** $\hat{h} : \mathbb{R}^d \to \mathbb{R}$, such that $\text{err}(\hat{h}) \leq O(\epsilon)$ with non-trivial probability.

3: $B_0 := r_0^2/\text{polylog}(R^2/\epsilon)$

4: **for** $i = 0, \ldots, T-1$ **do**

5:     Generate $\text{polylog}(R^2/\epsilon)$ samples $x^{(j)} \sim N(0, I) \mid \{w^{(i)} \cdot x > t_i\}$. Query $y(x^{(j)})$ and use them to get an estimate $\hat{g}_i$ for $(\mathbf{E}\, U_i, \mathbf{E}\, F_i)$.

6:     **if** $\|\hat{g}_i\| \geq B_i \text{polylog}(R^2/\epsilon)$ **then**

7:         Set $(r_{i0}, t_{i0}) = (r_i, t_i)$ and update $(r_{ij}, t_{ij}) = (r_{i(j-1)}, t_{i(j-1)}) - (\hat{g}_{ij})/\text{polylog}(R^2/\epsilon)$ until $\|\hat{g}_{ij}\| \leq B_i \text{polylog}(R^2/\epsilon)$

8:     $(r_{i+1}, t_{i+1}) \leftarrow (r_{ij}, t_{ij})$

9:     **for** $j = 1, \ldots, \tilde{O}(d)$ **do**

10:         Generate $x^{(j)} \sim N(0, I) \mid \{w^{(i+1)} \cdot x > t_{i+1}\}$ and query $y(x^{(j)})$

11:     Estimate $\mathbf{E}\, G_i$ via median of mean and get $\hat{G}_i$

12:     $w^{(i+1)} = \text{proj}_{\mathbb{S}^{d-1}} \left( w^{(i)} - \mu \hat{G}_i \right)$

13:     $B_{i+1} = (1 - \rho)B_i$

14: $\hat{w} = w^{(T)}$

15: Build a unit grid of size $1/\text{polylog}(R^2/\epsilon)$ over the ball centered at $(r_T, t_T)$ and randomly select a pair $(\hat{r}, \hat{t})$ from the grid.

16: **return** $h(\hat{r}, \hat{w}, \hat{t})$

---

the noiseless error. When $(r_i, t_i)$ contributes more noiseless error than $w^{(i)}$, $w^{(i)}$ might be updated incorrectly due to the estimation error.

To overcome this difficulty, we make use of a potential analysis by maintaining a *suitably small* upper bound for $(r^*)^2 \sin^2 \theta_i$ and reduce the upper bound in each round of our algorithm by considering the update $w^{(i+1)} = \text{proj}_{\mathbb{S}^{d-1}}(w^{(i+1)} - \mu \hat{G}_i)$ for some suitably small $\mu$, where $\hat{G}_i$ is an estimation for $\mathbf{E}\, G_i$ with $\tilde{O}(d)$ queries. We will show that as long as the error from $(r_i, t_i)$ is within a $\text{polylog}(R^2/\epsilon)$ factor of $B_i^2$, $d\text{polylog}(R^2/\epsilon)$ queries are enough to estimate $\mathbf{E}\, G_i$ and decrease the angle $\theta_i$. So, the key to obatin the correct query complexity is to update $(r_i, t_i)$ correctly, so that it introduces small error throughout the implementation of Algorithm 1.

$(r, t)$ **Update** We will next describe the way we update $(r, t)$. To begin with, we briefly explain the technical difficulty that needs to be handled. To simplify the intuition, we consider a simple one-dimensional setting where the optimal hypothesis is $r^* \sigma(z - \bar{t}^*)$, and we want to learn $(r^*, t^*)$. An immediate observation is that if we know $\bar{t}^*$ to good high accuracy, then applying a binary search over $r$ via querying examples with $z > \bar{t}^*$, we can learn $r^*$ efficiently. However, learning $t^*$ with few queries with an inaccurate learned parameter $r$ is challenging, because even estimating the bias of the target ReLU needs many samples. In halfspace learning, [DKM24] overcomes the difficulty by querying examples for which $|z - \bar{t}| < B \ll 1$. Such a method can zoom the error from $(\bar{t} - \bar{t}^*)$ and $(\bar{t} - \bar{t}^*)/B$, which allows us to make a binary search to find $\bar{t}^*$ with very few queries. Unfortunately, this method is not robust to label noise in $l_2$ loss. Intuitively, queries are more sensitive when the labels are real-valued instead of binary. For the $l_2$ loss, the noise rate could be very high within any small region, making this approach fail.

To bypass this difficulty, for a given ReLU activation $h(r_i, w^{(i)}, t_i)$, we consider the following two quantities: $U_i^* := (h(r_i, w^{(i)}, t_i) - h^*)(w^{(i)} \cdot x)$, $F_i^* := -(h(r_i, w^{(i)}, t_i) - h^*)$, and their noisy version $U_i, F_i$. Here we consider $x \sim N(0, I) \mid \{w^{(i)} \cdot x > \bar{t}_i\}$. Intuitively, if $(r_i, t_i)$ are close to $(r^*, t^*)$, the task for optimizing them can be approximately seen as optimizing the following quadratic function $Z(r, t) := \mathbf{E}_{x \sim N(0, I)} \left((rz - t) - (r^* z - t^*)\right)^2 \mathbb{1}(z > \bar{t}^*)$. Such a quantity nearly characterizes the contribution of $(r_i, t_i)$ to the noiseless error as well as the contribution of $(r_i, t_i)$ to the variance of the gradient we use for updating $w^{(i)}$. Furthermore, if $\theta_i$ has already been updated in a reasonable range, then $(\mathbf{E}\, U_i, \mathbf{E}\, F_i)$ is very close to the gradient of $Z(r_i, t_i)$. This gives the intuition to estimate $(\mathbf{E}\, U_i, \mathbf{E}\, F_i)$ and run a standard gradient descent update to improve $(r_i, t_i)$. To formalize

this intuition, we need to overcome two technical challenges. First, unlike the angle update, where the error from $(r_i, t_i)$ does not affect the update direction $\mathbf{E}\, G_i$, the mismatch of $w^{(i)}, w^*$ does affect $\mathbf{E}\, U_i^*, \mathbf{E}\, F_i^*$, even if we ignore the noise and estimation error. This is because the mismatch between $w^{(i)}$ and $w^*$ also contributes to $U_i, F_i$, which could vanish or even reverse the gradient for updating $(r_i, t_i)$. We defer the quantitative evaluation for the noise level and variance for $U_i, F_i$ to Appendix C. Fortunately, such forms of error are only $O(r^* \sin\theta_i)$, which means that as long as $w^{(i)}$ is updated to a reasonable range, we are safe to update $(r_i, t_i)$. So, we will simultaneously maintain $w^{(i)}$ within a reasonable region and only update $(r_i, t_i)$, when the angle is within a reasonable accuracy. The second technique issue is because when the true threshold is large, $Z(r, t)$ is an ill-conditioned function. Due to the presence of noise, the ill-condition of $Z(r, t)$ implies that even if $w^{(T)}$ has already been updated to a desirable accuracy, the gradient descent update can only guarantee that $(r_T, t_T)$ is $O(\epsilon \mathrm{polylog}(R^2/\epsilon)/\Phi(\tilde{t^*}))$ close to $(r^*, t^*)$ in terms of squared norm. Fortunately, there are only two parameters we need to worry about and they are already very close to $(r^*, t^*)$; randomly selecting a pair of parameters from their neighborhoods gives us a good hypothesis with enough probability.

## 4 Label Complexity Lower Bounds And the Necessity of Using Queries

Here we present our label complexity lower bounds. We defer the detailed proofs in this section to Appendix D. We start with an information-theoretic lower bound in the query setting.

**Theorem 4.1** (Query Complexity Lower Bound). *Consider the problem of agnostic ReLU regression with queries with a restriction that the optimal ReLU satisfies $\|W^*\| \leq 1$ and has bias at least $p$. Any learning algorithm that outputs a hypothesis with error less than $O(p/\log^2(p))$ with probability $1/3$, must make at least $\tilde{\Omega}(1/p^{1-o_d(1)} + d)$ queries. Furthermore, this holds even if $\mathrm{opt} \leq 2^{-\Omega(d^{1/4})}p$.*

The proof of Theorem 4.1 can be broken down into two parts. We first consider the lower bound of $\Omega(d)$. Such a lower bound even holds for an easier problem, which is the standard linear regression problem in the realizable setting. Suppose that we have made $r$ queries $x^{(1)}, \ldots, x^{(r)}$ so far, and denote by $L$ the subspace spanned by them. Consider a Bayesian setting, where $w^* \sim \mathbb{S}^{d-1}$. Suppose we know $w_L^*$, then for a new example $x$, by symmetry, no hypothesis will have better error than the hypothesis $w_L^* \cdot x$. In particular, if $x \sim N(0, I)$, this implies that no hypothesis has error better than $1 - \|w_L^*\|_2^2$. For any possible subspace $L$ with dimension $r$, in expectation $\|\mathrm{proj}_L w^*\|_2^2 = r/d$, which implies that unless $\Omega(d)$ queries are made, an algorithm must incur $\Omega(1)$ error. On the other hand, consider a hypothesis $\sigma(w^* \cdot x - t^*)$ with bias $p$. We want to tell whether the hypothesis is $0$ or not. Notice that in the realizable setting, if we make a query $x$ and find $y(x) = 0$, then by making another query in the opposite direction (but very far) we can easily verify whether $h \equiv 0$. However, with only a tiny fraction of adversarial label noise, we can corrupt all examples far from $0$ to have $0$ label. Now if $w^* \sim \mathbb{S}^{d-1}$, then only examples in a small cap with volume $p^{1-o(1)}$ have non-zero label; thus, unless $1/p^{1-o(1)}$ queries are made, we are not able to solve the distinguishing problem.

Our next lower bound shows that no pool-based active learner can achieve the query complexity of our query learner, even in the realizable setting.

**Theorem 4.2** (Label Complexity Lower Bound, Active Learning). *Consider the problem of realizable pool-based active ReLU regression with a restriction that the optimal ReLU has bias $p$. Any active learning algorithm $\mathcal{A}$ that makes less than $\tilde{O}(d/(p\log(m)))$ label queries over $S$, a set of $m$ i.i.d. points drawn from $N(0, I)$, will with probability at least $2/3$ output a hypothesis $\hat{h}$ with error $\tilde{\Omega}(p)$.*

While a number of prior works [Das04, HY15, KMT24b, DKM24] established query complexity lower bounds for active classification problems, few techniques could be directly applied to the regression setting. An immediate obstacle is that the behavior of an adaptive algorithm can be much more complicated when the label is changed from binary to continuous. An observation, inspired by prior works for proving label complexity lower bounds for learning halfspaces[Das04, DKM24], is that given a pool of $m$ unlabeled examples, learning a hypothesis with queries is not harder than using queries to find $d$ examples with non-zero labels. So we will focus on the hardness of this easier problem. By Yao's minimax principle, we consider a deterministic algorithm that solves the problem, while $w^* \sim \mathbb{S}^{d-1}$ for large enough $d$. Suppose that a deterministic algorithm wants to use $r$ queries to find $k$ examples with non-zero labels from a pool of $m$ unlabeled examples drawn from $N(0, I)$. Given a set of $m$ unlabeled examples, and any fixed $w^*$, the behavior of $\mathcal{A}$ can be uniquely described

as a path $P = ((x^1, y^1), \ldots, (x^r, y^r))$ according to its responses. In particular, if the algorithm successfully finds $k$ examples with non-zero responses, then there exist $k$ indices $i_1, \ldots, i_k$ at which the responses are $y^{i_1}, \ldots, y^{i_k} > 0$. Since there are $\binom{r}{k}$ such tuples, to argue that the algorithm has a large probability of failure, it suffices to argue that for a fixed tuple the probability of realization is very small. Proving such a statement turns out to be challenging, due to the rich behavior of the algorithm. In the binary classification setting, for each fixed tuple, the corresponding unlabeled examples $(x^{i_1}, \ldots, x^{i_k})$ are unique, since $y \in \{0, 1\}$. That is, to prove a lower bound for classification problem, we only need to argue that the probability these $k$ examples are all positive is small. Such a strategy does not work for regression, as the next example not only depends on whether the previous example is positive, but also depends on the full value of $y$. This results in many possible realizations of $((x^{i_1}, y^{i_1}), \ldots, (x^{i_k}, y^{i_k}))$. So we need to bound the integral of their density function over all possible outcomes. Consider a realization of the event $((x^{i_1}, y^{i_1}), \ldots, (x^{i_k}, y^{i_k}))$, with $y^{i_j} > 0$. Such a realization completely characterizes a vector $w_L \in L$, where $L$ is the subspace spanned by these $k$ queries. One observation is that if we change the basis of $L$ by defining $b_1 = x^{(i_1)} / \|x^{(i_1)}\|$ and $b_j = \text{proj}_{L_{j-1}} x^{(i_j)} / \|\text{proj}_{L_{i-1}} x^{(i_j)}\|$, where $L_{i-1} = \textbf{span}\{x^{(i_1)}, \ldots, x^{(i_{j-1})}\}$, then $w_L$ can be described as a $k$-dimensional vector $v(w_L) := (w_L \cdot b_1, \ldots, w_L \cdot b_k)$. Importantly, the value of the density function of the event is exactly the density of the event that the first $k$ coordinates of a random unit random vector equal to $v(w_L)$. On the other hand, due to the tree-structure of the algorithm, given any possible $v(w_L)$, we can decode it to reconstruct the corresponding $((x^{i_1}, y^{i_1}), \ldots, (x^{i_k}, y^{i_k}))$. Denote by $S_w$ the set of all possible $v(w_L)$. This means the probability we are interested in is exactly equal to the probability that $\text{proj}_{L_0}(w^*) \in S_w$, where $L_0$ is the span of the first $k$ standard basis vectors. To derive an upper bound for this probability, it is sufficient to find a superset of $S_w$. Our observation is that if $x^{i_1}, \ldots, x^{i_k}$ are orthogonal, then $\|w_L\|^2 \geq k(t^*)^2$, which also implies that $\|v(w_L)\|^2 \geq k(t^*)^2$. Furthermore, as long as $m$ is not as large as $2^{\Omega(d)}$, for $k$ chosen to be slightly smaller than $d$, every $k$-tuple of examples from the pool are nearly orthogonal. This implies that the norm of $v(w_L)$ can be lower bounded uniformly, which suffices to bound above the target probability by $O(p \log(1/p))^k$. Since there are at most $\binom{r}{k}$ tuples we care about and $k$ is slightly smaller than $d$, a carefully chosen $r$ concludes the proof.

## Acknowledgement

Ilias Diakonikolas was supported by NSF Medium Award CCF-2107079, NSF Award CCF-1652862 (CAREER), a Sloan Research Fellowship, and a DARPA Learning with Less Labels (LwLL) grant. Daniel M. Kane was supported by NSF Medium Award CCF-2107547 and NSF Award CCF-1553288 (CAREER). Mingchen Ma was supported by NSF Award CCF-2144298

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

# Supplementary Material

**Structure of Supplementary Material**    The supplementary material is organized as follows. In Appendix A, we record the notation and mathematical background required for our technical sections. In Appendix B, we present omitted details from Section 2. In Appendix C, we provide a full version of Section 3, including a detailed discussion on the update rule for $w$ and $(r, t)$, missing technical lemmas and proofs, and the proof of Theorem 3.1. Finally, in Appendix D, we give the detailed proofs of our lower bound results from Section 4.

## A    Preliminaries and Related Background

### A.1    Problem Definitions and Notation

We first define the task of Agnostic Learning with queries and the corresponding task in the active learning setting.

**Definition A.1** (Agnostic ReLU Regression with Queries)**.** *Let* $\sigma(z) = \max\{z, 0\}$ *be the ReLU function. A labeling function* $y(x) : \mathbb{R}^d \to \mathbb{R}$ *is a random function that maps each* $x \in \mathbb{R}^d$ *to an unknown real-valued random variable. For each* $h : \mathbb{R}^d \to \mathbb{R}$*, denote by* $\mathrm{err}(h) = \mathbf{E}_{x \sim N(0,I)} \left(h(x) - y(x)\right)^2 / 2$*,* $\mathrm{opt} := \min_{W : \|W\| \le R, t \ge 0} \mathrm{err}(\sigma(W \cdot x - t))$ *and* $\sigma^*(x) = \sigma(W^* \cdot x - t^*)$ *be any ReLU with error* $\mathrm{opt}$*. A query takes* $x \in \mathbb{R}^d$ *as an input and returns a label* $y \sim y(x)$*. We say that a learning algorithm* $\mathcal{A}$ *is a constant-factor approximate learner if for every labeling function* $y(x)$*, and for every* $\epsilon, \delta \in (0, 1)$*, it outputs some hypothesis* $\hat{h} : \mathbb{R}^d \to \mathbb{R}$ *by adaptively making queries, such that with probability at least* $1 - \delta$*,* $\mathrm{err}(\hat{h}) \le O(\mathrm{opt}) + \epsilon$*. The query complexity of* $\mathcal{A}$ *is the total number of queries it uses during the learning process.*

**Definition A.2** (Pool-Based Active Learning for Agnostic ReLU Regression)**.** *Let* $\sigma : \mathbb{R} \to \mathbb{R}$ *be a known activation function and* $w^* \in \mathbb{S}^{d-1}$ *be a unit vector. Let* $D$ *be a distribution over* $\mathbb{R}^d \times \{\pm 1\}$ *such that* $D_x$*, the marginal distribution over* $x$*, is the standard Gaussian distribution* $N(0, I)$*. For each* $h : \mathbb{R}^d \to \mathbb{R}$*, denote by* $\mathrm{err}(h) = \mathbf{E}_{(x,y) \sim D} \left(h(x) - y\right)^2 / 2$*,* $\mathrm{opt} := \min_{w : \|w\| \le R} \mathrm{err}(\sigma(w \cdot x))$ *and* $\sigma^*(x) = \sigma(W^* \cdot x - t^*)$ *be any ReLU with error* $\mathrm{opt}$*. Let* $S$ *be a set of* $m$ *i.i.d. labeled examples drawn from* $D$*. An active learning algorithm (with label query access) is given* $S$ *but with hidden labels and is allowed to make a label query for each* $x \in S$ *and observe its label* $y(x)$*. We say that a learning algorithm* $\mathcal{A}$ *is a constant-factor approximate learner if for every distribution* $D$ *and for every* $\epsilon, \delta \in (0, 1)$*, it outputs some* $\hat{h} \in H$ *by adaptively making label queries over a set of* $m$ *examples drawn i.i.d. from* $D$*, such that with probability at least* $1 - \delta$*,* $\mathrm{err}(\hat{h}) \le O(\mathrm{opt}) + \epsilon$*. The label complexity of* $\mathcal{A}$ *is the total number label queries made over* $S$ *during the learning process.*

We will without loss of generality assume $\mathrm{opt} \le \epsilon$, since the final error guarantee is $O(\mathrm{opt} + \epsilon)$. We remark that in some parts of the paper, we will also consider the case where $\sigma$ is not a ReLU but a general function and $W^* \in \mathbb{S}^{d-1}$. In that case, we call the problem agnostic learning (spherical) generalized linear model (GLM) and for a hypothesis $\sigma(w \cdot x)$, we use $\mathrm{err}(w)$ to denote its error accordingly. For a vector $W$ in $\mathbb{R}^d$, we use $\|W\|$ to denote it $\ell_2$ norm and use the lower case $w$ to denote its direction $W / \|W\|$. For a one-dimensional standard random variable $z \sim N(0, 1)$ and $t \ge 0$, we denote by $\Phi(t) := \mathbf{Pr}_z(z > t)$ and $\psi(t)$ the value of the density function of $z$ at $t$. For a ReLU activation $\sigma(z - t)$, we denote by $V(t) := \mathbf{E}_{z \sim N(0,1)} \sigma^2(z - t)$ its second moment and call $p$ its bias. For a real valued function $f : \mathbb{R} \to \mathbb{R}$, we denote by $\|f\|_2^2 := \mathbf{E}_{z \sim N(0,1)} f^2(z)$ its squared $L_2$ norm in Gaussian space and for $a \in [0, 1]$, denote by $T_a f(z) := \mathbf{E}_{s \sim N(0,1)} f(az + \sqrt{1 - a^2} s)$.

### A.2    Background on Ornstein–Uhlenbeck Semigroup

Here we provide basic background on the Ornstein–Uhlenbeck Semigroup. We refer the readers to [O'D14] for additional background.

**Definition A.3** (Ornstein–Uhlenbeck Semigroup)**.** *Let* $\rho \in (0, 1)$*. The Ornstein–Uhlenbeck semigroup* $T_\rho : L_2 \to L_2$ *is a linear mapping that maps a function* $f$ *to a function* $T_\rho f$ *defined as*

$$T_\rho f(z) = \mathop{\mathbf{E}}_{s \sim N(0,1)} f(\rho z + \sqrt{1 - \rho^2} s).$$

**Definition A.4** (Ornstein–Uhlenbeck Operator). *The Ornstein–Uhlenbeck operator $L : L_2 \to L_2$ is a linear mapping that maps a function $f$ to a function $Lf$ defined as*

$$Lf(z) = \frac{dT_\rho f}{d\rho}\mid_{\rho=1}(z)$$

We list several facts about the Ornstein–Uhlenbeck semigroup and Ornstein–Uhlenbeck operator.

**Fact A.5** ([O'D14]). *Let $f, g \in L_2(N)$. The follows statements hold.*

1. *For $\rho \in [0,1]$, $\|T_\rho f\|_2^2$ is a non-decreasing function with respect to $\rho$.*

2. *For $a, b \in [0,1]$, $\mathbf{E}_{z \in N(0,1)} T_a f(z) T_b f(z) = \left\| T_{\sqrt{ab}} f(z) \right\|_2^2$*

3. *If $f$ is a differentiable function, then for $\rho \in (0,1)$ $(T_\rho f(z))' = \rho T_\rho f'(z)$.*

4. *$\frac{dT_\rho f}{d\rho} = \frac{1}{\rho} L T_\rho f$*

5. *$\mathbf{E}_{z \sim N(0,1)} (f(z) L T_\rho g(z)) = \mathbf{E}_{z \sim N(0,1)} (f'(z)(T_\rho g(z))')$*

### A.3 Background on Gaussian Integral

In this section, we provide background on Gaussian Integral. Let $z \sim N(0,1)$ be the standard normal random variable. For $t \geq 0$, we denote by $\Phi(t) := \mathbf{Pr}_z(z > t)$ and $\psi(t)$ the value of the density function of $z$ at $t$. For a ReLU activation $\sigma(z-t)$, we denote by $V(t) := \mathbf{E}_{z \sim N(0,1)} \sigma^2(z-t)$ its second moment. We provide detailed characterization for $\Phi(t), \psi(t), V(t)$. First, we provide the following fact that characterizes $\Phi(t)$.

**Fact A.6** (Komatsu's Inequality). *For any $t \geq 0$, $\Phi(t)$ can be bounded as*

$$\sqrt{\frac{2}{\pi}} \frac{\exp(-t^2/2)}{t + \sqrt{t^2+4}} \leq \Phi(t) \leq \sqrt{\frac{2}{\pi}} \frac{\exp(-t^2/2)}{t + \sqrt{t^2+2}}.$$

Next, we provide the following fact calculated by [GV24] that relates $V(t), \Phi(t), \psi(t)$.

**Fact A.7** (see, e.g., Appendix A, [GV24]). *For any $t \geq 0$, the following fact holds.*

$$\mathop{\mathbf{E}}_{z \sim N(0,1)} z \mathbb{1}(z > t) = \psi(t)$$

$$\mathop{\mathbf{E}}_{z \sim N(0,1)} z(z-t)\mathbb{1}(z > t) = \Phi(t)$$

$$\mathop{\mathbf{E}}_{z \sim N(0,1)} z^2 \mathbb{1}(z > t) = \Phi(t) + t\psi(t) = (1 + o_t(1))t^2 \Phi(t)$$

$$V(t) = (t^2 + 1)\Phi(t) - t\psi(t) = (2 + o_t(1))\Phi(t)/t^2$$

For large $t > 0$, it is useful to mention the following asymptotic relation between $\Phi(t)$ and $\psi(t)$,

$$\frac{\Phi(t)}{\psi(t)} \sim \frac{1}{t} - \frac{1}{t^3} + \frac{3}{t^5} - \cdots.$$

The following useful Stein's lemma will also be frequently used in our proofs.

**Fact A.8** (Stein's Lemma). *Let $x \sim N(\mu, \sigma^2 I)$ be a Gaussian vector in $\mathbb{R}^d$. Let $g : \mathbb{R}^d \to \mathbb{R}$ such that $\mathbf{E}_{x \sim N(\mu, \sigma^2 I)} g(x)x$ and $\mathbf{E}_{x \sim N(\mu, \sigma^2 I)} \nabla g(x)$ exist. Then the following fact holds*

$$\mathop{\mathbf{E}}_{x \sim N(\mu, \sigma^2 I)} g(x)(x - \mu) = \sigma^2 \mathop{\mathbf{E}}_{x \sim N(\mu, \sigma^2 I)} \nabla g(x).$$

## B    Omitted Details from Section 2

Here we provide the full details omitted from Section 2.

## B.1 Proof of Lemma 2.2

We provide the proof of Lemma 2.2 and its restatement as Lemma B.1.

**Lemma B.1** (Restatement of Lemma 2.2). *Let $\sigma : \mathbb{R} \to \mathbb{R}$ be any activation function such that $\sigma' \in L_2(N)$ and let $w \in \mathbb{S}^{d-1}$ be any unit vector such that $\theta := \theta(w, w^*) < \pi/2$, then*

$$\ell(w) = \int_0^\theta \sin s \left\| T_{\sqrt{\cos s}} \sigma' \right\|_2^2 ds \le \frac{\pi}{2} \sin^2 \theta \left\| \sigma' \right\|_2^2.$$

*Proof of Lemma 2.2.* We write $w^* = aw + bu$, where $a, b > 0, a^2 + b^2 = 1$ and $u \in \mathbb{S}^{d-1}, u \perp w$. Notice that if $x \sim N(0, I)$, then $z = w \cdot x$ and $s = u \cdot x$ are independent standard one dimensional normal random variables.

$$\ell(w) = \frac{1}{2} \mathop{\mathbf{E}}_{x \sim N(0,I)} \left( \sigma(w \cdot x) - \sigma(w^* \cdot x) \right)^2 = \mathop{\mathbf{E}}_{z \sim N(0,1)} \sigma^2(z) - \mathop{\mathbf{E}}_{x \sim N(0,I)} \sigma(w \cdot x) \sigma(w^* \cdot x)$$

$$= \mathop{\mathbf{E}}_{z \sim N(0,1)} \sigma^2(z) - \mathop{\mathbf{E}}_{x \sim N(0,I)} \sigma(w \cdot x) \sigma(aw \cdot x + bu \cdot x)$$

$$= \mathop{\mathbf{E}}_{z \sim N(0,1)} \sigma^2(z) - \mathop{\mathbf{E}}_{z,s \sim N(0,1)} \sigma(z) \sigma(az + bs) = \mathop{\mathbf{E}}_{z \sim N(0,1)} \sigma(z) \left( \sigma(z) - T_a \sigma(z) \right)$$

$$= \mathop{\mathbf{E}}_{z \sim N(0,1)} \sigma(z) \int_a^1 \frac{dT_s \sigma(z)}{ds} ds = \mathop{\mathbf{E}}_{z \sim N(0,1)} \sigma(z) \int_a^1 \frac{1}{s} L T_s \sigma(z) ds = \int_a^1 \mathop{\mathbf{E}}_{z \sim N(0,1)} \sigma(z) \frac{1}{s} L T_s \sigma(z) ds \tag{2}$$

$$= \int_a^1 \mathop{\mathbf{E}}_{z \sim N(0,1)} \sigma'(z) \frac{1}{s} (T_s \sigma(z))' ds = \int_a^1 \mathop{\mathbf{E}}_{z \sim N(0,1)} \sigma'(z) T_s \sigma'(z) ds \tag{3}$$

$$= \int_a^1 \left\| T_{\sqrt{s}} \sigma' \right\|_2^2 ds = \int_0^\theta \sin s \left\| T_{\sqrt{\cos s}} \sigma' \right\|_2^2 ds.$$

Here, (2) follows Item 4 and (3) holds because of Item 3 and Item 5. The last equation follows a change of variable. By the monotone property of $\|T_\rho \sigma'\|$, and $\theta \le \pi/2 \sin \theta$ for $\theta \in (0, \pi/2)$, we obtain that

$$\ell(w) = \int_0^\theta \sin s \left\| T_{\sqrt{\cos s}} \sigma' \right\|_2^2 ds \le \frac{\pi}{2} \sin^2 \theta \left\| \sigma' \right\|_2^2.$$

$\square$

## B.2 Proof of Lemma 2.3

Here we provide the proof of Lemma 2.3 and its restatement Lemma B.2.

**Lemma B.2.** *Let $\sigma : \mathbb{R} \to \mathbb{R}$ be any activation function such that $\sigma' \in L_2(N(0,I))$ and let $w \in \mathbb{S}^{d-1}$ be any unit vector such that $\theta = \theta(w, w^*) < \pi/2$, where $\mathrm{err}(\sigma(w^*)) = \mathrm{opt}$. Write $w^* = aw + bu$, where $u \in \mathbb{S}^{d-1}, u \perp w, a, b \ge 0, a^2 + b^2 = 1$, then, $\mathrm{proj}_{w^\perp} \nabla_w \ell(w) = -b \left\| T_{\sqrt{a}} \sigma' \right\|_2^2 u$.*

*Proof of Lemma 2.3.* By Fact A.5 and Fact A.8, we have the following calculation

$$\mathrm{proj}_{w^\perp} \nabla_w \ell(w) = \mathrm{proj}_{w^\perp} \mathop{\mathbf{E}}_{x \sim N(0,I)} \left( \sigma(w \cdot x) - \sigma(w^* \cdot x) \right) \sigma'(w \cdot x) x$$

$$= \mathrm{proj}_{w^\perp} \mathop{\mathbf{E}}_{x \sim N(0,I)} \sigma(w \cdot x) \sigma'(w \cdot x) x - \mathrm{proj}_{w^\perp} \mathop{\mathbf{E}}_{x \sim N(0,I)} \sigma(w^* \cdot x) \sigma'(w \cdot x) x$$

$$= -\mathrm{proj}_{w^\perp} \mathop{\mathbf{E}}_{x \sim N(0,I)} \sigma(w^* \cdot x) \sigma'(w \cdot x) x = -\mathrm{proj}_{w^\perp} \mathop{\mathbf{E}}_{x \sim N(0,I)} \nabla_x \left( \sigma(w^* \cdot x) \sigma'(w \cdot x) \right)$$

$$= -\mathrm{proj}_{w^\perp} \mathop{\mathbf{E}}_{x \sim N(0,I)} \left( \sigma'(w^* \cdot x) \sigma'(w \cdot x) w^* + \sigma(w^* \cdot x) \sigma''(w \cdot x) w \right)$$

$$= -\mathop{\mathbf{E}}_{x \sim N(0,I)} \sigma'(w^* \cdot x) \sigma'(w \cdot x) bu = -\mathop{\mathbf{E}}_{z,s \sim N(0,1)} \sigma'(az + bs) \sigma'(z) bu$$

$$= -\mathop{\mathbf{E}}_{z \sim N(0,1)} T_a \sigma'(z) \sigma'(z) bu = -b \left\| T_{\sqrt{a}} \sigma' \right\|_2^2 u.$$

$\square$

### B.3   Proof of Lemma 2.4

We next give the proof for Lemma 2.4 and its restatement as Lemma B.3.

**Lemma B.3.** *Let $\sigma : \mathbb{R} \to \mathbb{R}$ be any activation function such that $\sigma' \in L_2(N(0, I))$ and let $w \in \mathbb{S}^{d-1}$ be any unit vector such that $\theta = \theta(w, w^*) < \pi/2$, where $\mathrm{err}(\sigma(w^*)) = \mathrm{opt} \leq \epsilon$. Then for any $v \in \mathbb{S}^{d-1}$ and $v \perp w$, $|\mathrm{proj}_{w^\perp} (\nabla_w \mathrm{err}(w) - \nabla_w \ell(w)) \cdot v| \leq \sqrt{\epsilon} \, \|\sigma'\|_2$.*

*Proof of Lemma 2.4.*  We notice that

$$\mathrm{proj}_{w^\perp} (\nabla_w \mathrm{err}(w) - \nabla_w \ell(w)) = \mathrm{proj}_{w^\perp} \mathop{\mathbf{E}}_{x \sim N(0,I)} (\sigma(w^* \cdot x) - y) \, \sigma'(w \cdot x) x.$$

For every $v \in \mathbb{S}^{d-1}$ and $v \perp w$, we have

$$
\begin{aligned}
|\mathrm{proj}_{w^\perp} (\nabla_w \mathrm{err}(w) - \nabla_w \ell(w)) \cdot v| &= \left| \mathop{\mathbf{E}}_{x \sim N(0,I)} (\sigma(w^* \cdot x) - y) \, \sigma'(w \cdot x)(x \cdot v) \right| \\
&\leq \sqrt{\mathop{\mathbf{E}}_{x \sim N(0,I)} (\sigma(w^* \cdot x) - y)^2} \sqrt{\mathop{\mathbf{E}}_{x \sim N(0,I)} (\sigma'(w \cdot x)(x \cdot v))^2} \\
&= \sqrt{\mathrm{opt}} \sqrt{\mathop{\mathbf{E}}_{z,s \sim N(0,1)} (\sigma'(z)s)^2} \leq \sqrt{\epsilon} \, \|\sigma'\|_2 \, .
\end{aligned}
$$

Here, in the first inequality, we use Holder's inequality and in the last inequality, we use the fact that $z, s$ are independent. $\qquad\square$

### B.4   Proof of Lemma 2.5

We next give the proof of the angle contraction lemma as follows.

**Lemma B.4** (Angle Contraction). *Let $w^*, w^{(i)} \in \mathbb{S}^{d-1}$ such that $w^* = aw^{(i)} + bu$, where $u \in \mathbb{S}^{d-1}, u \perp w^{(i)}, a, b \geq 0, a^2 + b^2 = 1$. Let $\theta_i = \theta(w^{(i)}, w^*)$. Let $G \in \mathbb{R}^d$ be a random vector such that with probability 1, $G \perp w^{(i)}$. Let $g$ be the mean of $G$ and $\hat{g} \in \mathbb{R}^d$. Suppose there is some $c > 0$ such that $g \cdot u \geq cb/10, \|g\| \leq cb, \|g - \hat{g}\| \leq bc/40$, then by setting $\mu = c/20$, the update rule $w^{(i+1)} = \mathrm{proj}_{\mathbb{S}^{d-1}}(w^{(i)} + \mu\hat{g})$ satisfies $\sin(\theta_{i+1}/2) \leq \sqrt{1 - \left(\frac{c}{20}\right)^2} \sin(\theta_i/2)$.*

*Proof of Lemma 2.5.*

$$\left\| w^{(i+1)} - w^* \right\|^2 = \left\| \mathrm{proj}_{\mathbb{S}^{d-1}}(w^{(i)} + \mu\hat{g}) - \mathrm{proj}_{\mathbb{S}^{d-1}}(w^*) \right\|^2 \leq \left\| w^{(i)} + \mu\hat{g} - w^* \right\|^2 .$$

It remains to upper bound $\left\| w^{(i)} + \mu\hat{g} - w^* \right\|^2$. We have

$$
\begin{aligned}
\left\| w^{(i)} + \mu\hat{g} - w^* \right\|^2 &= \left\| w^{(i)} - w^* \right\|^2 + 2\mu\hat{g} \cdot (w^{(i)} - w^*) + \mu^2 \|\hat{g}\|^2 \\
&= \left\| w^{(i)} - w^* \right\|^2 - 2\mu\hat{g} \cdot w^* + \mu^2 \|\hat{g}\|^2 \\
&= \left\| w^{(i)} - w^* \right\|^2 - 2\mu g \cdot w^* + 2\mu(g - \hat{g}) \cdot w^* + \mu^2 \|\hat{g}\|^2 \\
&= \left\| w^{(i)} - w^* \right\|^2 - 2\mu g \cdot w^* + 2\mu(g - \hat{g}) \cdot bu + \mu^2 \|\hat{g}\|^2 \\
&\leq \left\| w^{(i)} - w^* \right\|^2 - 2\mu g \cdot w^* + 2\mu b \|g - \hat{g}\| + \mu^2 \|\hat{g}\|^2 \, . \\
&= \left\| w^{(i)} - w^* \right\|^2 - 2\mu b g \cdot u + 2\mu b \|g - \hat{g}\| + \mu^2 \|\hat{g}\|^2 \, .
\end{aligned}
$$

Here, in the second equality, we use the fact that $\hat{g} \perp w^{(i)}$ and in the fourth equality, we use the fact that $(g - \hat{g}) \cdot w^* = (g - \hat{g}) \cdot aw^{(i)} + (g - \hat{g}) \cdot bu = (g - \hat{g}) \cdot bu$.

Notice that $\left\|w^{(i)} - w^*\right\| = 2\sin\frac{\theta_i}{2}$ and $b = \sin\theta_i$. By choosing $\mu = 1/C \le c/20$ We have

$$(2\sin\frac{\theta_{i+1}}{2})^2 \le (2\sin\frac{\theta_i}{2})^2 - \mu c b^2/5 + \mu c b^2/20 + \mu^2 c^2 b^2$$

$$\le (2\sin\frac{\theta_i}{2})^2 - \mu c \sin^2\frac{\theta_i}{2}/5 = 4(1 - \frac{\mu c}{20})\sin^2\frac{\theta_i}{2} = 4(1 - \left(\frac{c}{20}\right)^2)\sin^2\frac{\theta_i}{2}$$

This implies $\sin(\theta_{i+1}/2) \le \sqrt{1 - \left(\frac{c}{20}\right)^2}\sin(\theta_i/2)$. $\qquad\qquad\square$

## B.5 Proof of Lemma 2.6

For convenience, we restate the lemma below.

**Lemma B.5.** *Let $\sigma : \mathbb{R} \to \mathbb{R}$ be any activation function such that $\sigma' \in L_2(N(0, I))$. Let $\alpha > 1$ and $0 < \theta_0 < \pi/2$ such that $\left\|T_{\sqrt{\cos\theta_0}}\sigma'\right\|_2^2 \ge \|\sigma'\|_2^2/\alpha$. Let $w \in \mathbb{S}^{d-1}$ be any unit vector such that $\theta = \theta(w, w^*) < \theta_0$, where $\mathrm{err}(\sigma(w^*)) = \mathrm{opt} \le \epsilon$. If $\sin^2\theta\|\sigma'\|_2^2 \ge 20\alpha^2\epsilon/\pi$, then for any $v \in \mathbb{S}^{d-1}$ and $v \perp w$, $\left\|\mathrm{proj}_{w^\perp}(\nabla_w\mathrm{err}(w) - \nabla_w\ell(w))\right\| \le \left\|\mathrm{proj}_{w^\perp}\nabla_w\ell(w)\right\|/20$. Furthermore, if $\left\|\mathrm{proj}_{w^\perp}(\nabla_w\mathrm{err}(w) - \nabla_w\ell(w))\right\| > \left\|\mathrm{proj}_{w^\perp}\nabla_w\ell(w)\right\|/20$, then $\mathrm{err}(w) \le O(\alpha^2\epsilon)$.*

*Proof of Lemma 2.6.* Since $\sin^2\theta\|\sigma'\|_2^2 \ge 20\alpha^2\epsilon$, we know that $\sqrt{\epsilon} \le \sin\theta\|\sigma'\|_2/\sqrt{20\alpha^2}$. By Lemma 2.4, we know that

$$\left\|\mathrm{proj}_{w^\perp}(\nabla_w\mathrm{err}(w) - \nabla_w\ell(w)) \cdot v\right\| \le \sqrt{\epsilon}\|\sigma'\|_2 \le \sin\theta\|\sigma'\|^2/\sqrt{20\alpha^2}$$

$$\le \sin\theta\left\|T_{\sqrt{\cos\theta}}\sigma'\right\|_2^2/\sqrt{20} = \left\|\mathrm{proj}_{w^\perp}\nabla_w\ell(w)\right\|/\sqrt{20}.$$

Here, the last inequality holds by the monotone property of $\|T_\rho\sigma'\|$. On the other hand, if $\left\|\mathrm{proj}_{w^\perp}(\nabla_w\mathrm{err}(w) - \nabla_w\ell(w))\right\| > \left\|\mathrm{proj}_{w^\perp}\nabla_w\ell(w)\right\|/\sqrt{20}$, by Lemma 2.2, we know that

$$\mathrm{err}(w) \le \mathop{\mathbf{E}}_x\left(\sigma(w \cdot x) - \sigma(w^* \cdot x)\right)^2 + \mathop{\mathbf{E}}_x\left(\sigma(w^* \cdot x) - y\right)^2 \le 2\mathrm{opt} + \mathop{\mathbf{E}}_x\left(\sigma(w \cdot x) - \sigma(w^* \cdot x)\right)^2$$

$$\le 2\mathrm{opt} + \pi\sin^2\theta\|\sigma'\|_2^2 \le O(\alpha^2\epsilon).$$

$\qquad\qquad\square$

Combining Lemma 2.2, Lemma 2.3, Lemma 2.4, Lemma 2.6, we know that the update rule (1) satisfies the following property. Suppose we are given a warm start $w^{(0)}$ such that $\left\|T_{\sqrt{\cos\theta_0}}\sigma'\right\|_2^2 \ge \|\sigma'\|_2^2/\alpha$. As long as the current $w$ has large angle, and thus has large noiseless error, the noise rate is much smaller than the length of the gradient used in the update and the angle can be improved. On the other hand, once the length of the gradient is small and the noise level is large, it must be the case that the angle is small enough so that the error of the current hypothesis is as small as $O(\alpha^2\epsilon)$. We next use this property to show that when $\|W^*\| = 1$ and $t^* > 0$ is given, we are able to solve the ReLU regression problem with label complexity $\tilde{O}(1/p + d\,\mathrm{polylog}(1/\epsilon))$.

## B.6 Proof of Lemma 2.8

We start by proving that using a method of label truncation, we are able to reduce the initialization for ReLU regression to the initialization for agnostic learning of halfspaces. We present the proof of Lemma 2.8 and its restatement Lemma B.6.

**Lemma B.6.** *Let $\sigma$ be an activation f the form $\sigma(z) = \mathrm{Relu}(z - t^*)$, where $t^* > 0$. Let $y(x)$ be any labeling function such that $\mathrm{opt} \le \epsilon$. Let $c > 0$ be a suitably small constant. If $V(t^*) > C\epsilon$, for some large constant $C$, then $\mathbf{Pr}_{x \sim N(0,I)}(\bar{y}(x) \ne \mathrm{sign}(w^* \cdot x - t^*)) \le \Phi(t^*)/C'$ for some large constant $C' > 0$, where $\bar{y}(x) := \mathbb{1}\{y(x) > c/(t^*)\}$.*

The idea of the proof is as follows. Since the labels are continuous, for those examples with ground truth labels very close to the threshold $c/(t^*)$, we are not able to control their behavior. But on the other hand, for those examples with ground truth labels that are far from the threshold, to change their pseudo-label $\bar{y}(x)$, the adversary must add high-level noise to them. We will show that as long as $h \equiv 0$ does not have error $O(\epsilon)$, we can reduce the problem to halfspace learning by carefully choosing $c$.

*Proof of Lemma 2.8.* We partition $\mathbb{R}^d$ into three regions $I_1 := \{x \mid w^* \cdot x > t^* + 2c/(t^* + 1)\}, I_2 := \{x \mid w^* \cdot x < t^*\}$ and $J := \{x \mid t^* < w^* \cdot x < t^* + 2c/(t^* + 1)\}$. Since $\mathrm{opt} = \mathbf{E}_{x \sim N(0,I)} \left( \sigma(w^* \cdot x - t^*) - y \right)^2 / 2 < \epsilon$, we have

$$
\begin{aligned}
\epsilon &\geq \underset{x \sim N(0,I)}{\mathbf{E}} \left( \sigma(w^* \cdot x - t^*) - y \right)^2 / 2 \\
&\geq \underset{x \sim N(0,I)}{\mathbf{E}} \mathbb{1}\{x \in I_1 \cup I_2\} \left( \sigma(w^* \cdot x - t^*) - y \right)^2 / 2 \\
&\geq \underset{x \sim N(0,I)}{\mathbf{E}} \left( \mathbb{1}\{x \in I_1, y(x) < c/t^*\} + \mathbb{1}\{x \in I_2, y(x) > c/t^*\} \right) \left( \sigma(w^* \cdot x - t^*) - y \right)^2 / 2 \\
&\geq \frac{c^2}{2(t^* + 1)^2} \underset{x \sim N(0,I)}{\mathbf{Pr}} \left( x \in I_1 \cup I_2, y(x) \neq \bar{y}(x) \right)
\end{aligned}
$$

Since $\mathbf{E}_{x \sim N(0,I)} \, \sigma(w^* \cdot x - t^*)^2 \geq C\epsilon$, we obtain that

$$
\underset{x \sim N(0,I)}{\mathbf{Pr}} \left( x \in I_1 \cup I_2, y(x) \neq \bar{y}(x) \right) \leq \frac{2(t^* + 1)^2}{c^2} \epsilon \leq \frac{2(t^* + 1)^2 \, \mathbf{E}_{x \sim N(0,I)} \, \sigma(w^* \cdot x - t^*)^2}{Cc^2} \leq \Phi(t^*)/C_1,
$$

for some large enough $C_1 > 0$.

On the other hand,

$$
\underset{x \sim N(0,I)}{\mathbf{Pr}} \left( x \in J, y(x) \neq \bar{y}(x) \right) \leq \underset{x \sim N(0,I)}{\mathbf{Pr}} \left( x \in J \right) \leq \frac{2c}{(t^* + 1)} \frac{1}{\sqrt{2\pi}} \exp\left( -\frac{(t^*)^2}{2} \right) \leq 10c\Phi(t^*).
$$

Thus,

$$
\begin{aligned}
\underset{x \sim N(0,I)}{\mathbf{Pr}} \left( \bar{y}(x) \neq \mathrm{sign}(w^* \cdot x - t^*) \right) &= \underset{x \sim N(0,I)}{\mathbf{Pr}} \left( x \in J, y(x) \neq \bar{y}(x) \right) + \underset{x \sim N(0,I)}{\mathbf{Pr}} \left( x \in I_1 \cup I_2, y(x) \neq \bar{y}(x) \right) \\
&\leq \Phi(t^*)/C_1 + 10c\Phi(t^*) = \Phi(t^*)/C''
\end{aligned}
$$

for some large enough $C'' > 0$. $\qquad\square$

## B.7 Proof of Lemma 2.7

Using the initialization technique recently developed in [DKM24], with label complexity $\tilde{O}(1/p + d\,\mathrm{polylog}(1/\epsilon))$, we are able to get a $w^{(0)}$ with $\theta_0 \leq O(1/t^*)$. We next show that such an angle satisfies Lemma 2.6 for $\alpha = O(1)$. We provide the proof of Lemma 2.7 and its restatement as Lemma B.7.

**Lemma B.7.** *Let $\sigma = \sigma_{t^*}^*$ be the optimal activation function of the form $\sigma(w^* \cdot x - t^*)$, where $\sigma$ is the ReLU function, $w^* \in \mathbb{S}^{d-1}$ and $t^* > 0$ is known. If $\sin(\theta/2) \leq 1/t^*$, then $\left\| T_{\sqrt{\cos\theta}} \sigma' \right\|_2^2 \geq \|\sigma'\|_2^2 / 50$.*

*Proof of Lemma 2.7.* Let $w \in \mathbb{S}^{d-1}$ be any direction such that $\sin(\theta/2) \leq 1/t^*$. Write $w^* = aw + bu$, where $u \in \mathbb{S}^{d-1}, u \perp w, a, b \geq 0, a^2 + b^2 = 1$. By Lemma 2.5, we know that

$$
\underset{x \sim N(0,I)}{\mathbf{E}} \sigma(w^* \cdot x - t^*) \sigma'(w^* \cdot x - t^*)(x \cdot u) = b \left\| T_{\sqrt{\cos\theta}} \sigma' \right\|_2^2.
$$

To show $\left\| T_{\sqrt{\cos\theta}} \sigma' \right\|_2^2 \geq \|\sigma'\|_2^2 / 50$, it is sufficient to show that $\mathbf{E}_{x \sim N(0,I)} \, \sigma(w^* \cdot x - t^*) \sigma'(w^* \cdot x - t^*)(x \cdot u) \geq b \|\sigma'\|_2^2 / 50$. Write $z = w \cdot x, s = u \cdot x$. Notice that $z, s$ are independent one-dimensional normal random variables. We have

$$
\begin{aligned}
\underset{x \sim N(0,I)}{\mathbf{E}} \sigma(w^* \cdot x - t^*) \sigma'(w \cdot x - t^*)(x \cdot u) &= \underset{z,s}{\mathbf{E}} \, \sigma(az + bs - t^*) \sigma'(z - t^*)s \\
&= \underset{z}{\mathbf{E}} \, \sigma'(z - t^*) \underset{s}{\mathbf{E}} \, \sigma(az + bs - t^*)s \\
&= b \underset{z}{\mathbf{E}} \, \sigma'(z - t^*) \underset{s}{\mathbf{E}} \, \sigma'(az + bs - t^*).
\end{aligned}
$$

Notice that $\sigma'(z - t^*) = \mathbb{1}(z > t^*)$. We have

$$
\begin{aligned}
b \underset{z}{\mathbf{E}} \sigma'(z - t^*) \underset{s}{\mathbf{E}} \sigma'(az + bs - t^*) &= b \int_{t^*}^{\infty} \left( \sigma'(z - t^*) \right)^2 \psi(z) \underset{s}{\mathbf{E}} \frac{\sigma'(az + bs - t^*)}{\sigma'(z - t^*)} dz \\
&= b \int_{t^*}^{\infty} \left( \sigma'(z - t^*) \right)^2 \psi(z) \underset{s}{\mathbf{E}} \frac{\sigma'(s - \frac{t^* - az}{b})}{\sigma'(z - t^*)} dz \\
&= b \int_{t^*}^{\infty} \left( \sigma'(z - t^*) \right)^2 \psi(z) \underset{s}{\mathbf{E}} \sigma'(s - \frac{t^* - az}{b}) dz \\
&\geq b \int_{t^*}^{\infty} \left( \sigma'(z - t^*) \right)^2 \psi(z) \underset{s}{\mathbf{E}} \sigma'(s - \frac{t^* - at^*}{b}) dz \\
&= b \int_{t^*}^{\infty} \left( \sigma'(z - t^*) \right)^2 \psi(z) dz \underset{s \sim N(0,1)}{\mathbf{Pr}} (s > \frac{b^2 t^*}{b(1 + a)}).
\end{aligned}
$$

Since $\sin(\theta/2) < 1/t^*$, we know that $b < 2/t^*$, which implies that $\mathbf{Pr}_{s \sim N(0,1)}(s > \frac{b^2 t^*}{b(1+a)}) \geq \mathbf{Pr}_s(s > 2) \geq 0.02$. Thus, we obtain that

$$
\underset{x \sim N(0,I)}{\mathbf{E}} \sigma(w^* \cdot x - t^*) \sigma'(w^* \cdot x - t^*)(x \cdot u) \geq b \left\| \sigma' \right\|_2^2 / 50.
$$

$\square$

### B.8   Proof of Lemma 2.10

Here we show how to use queries to boost the gradient used by (1). The idea behind our proof is that if we consider the random vector $G = (\sigma(w \cdot x - t^*) - y(x)) \operatorname{proj}_{w^\perp} x$, where $x \sim N(0, I) \mid_{\{x \mid w \cdot x > t^*\}}$, then with the warm-start we have that the expectation of such a random vector plays the same role as the gradient $\nabla_w \operatorname{err}(w)$, but has larger length and small variance. This allows us to estimate it to a desired accuracy with few queries. We provide the proof of Lemma 2.10 and its restatement Lemma B.8.

**Lemma B.8.** *Let* $\sigma(z) = \operatorname{Relu}(z - t^*)$, *with* $t^* > 0$. *Let* $y(x)$ *be any labeling function such that* $\operatorname{opt} \leq \epsilon$. *Let* $w \in \mathbb{S}^{d-1}$ *be any vector such that* $\sin(\theta/2) \leq 1/t^*$. *Denote by* $G^* \in \mathbb{R}^d$ *the random vector* $(\sigma(w \cdot x - t^*) - \sigma(w^* \cdot x - t^*)) \operatorname{proj}_{w^\perp} x$ *and* $G$ *the random vector* $(\sigma(w \cdot x - t^*) - y(x)) \operatorname{proj}_{w^\perp} x$, *where* $x \sim N(0, I) \mid_{\{x \mid w \cdot x > t^*\}}$. *Then the following holds:*

1. $\mathbf{E} G^* = b \left\| T_{\sqrt{a}} \sigma' \right\|_2^2 u / \Phi(t^*)$.

2. $|(\mathbf{E} G^* - \mathbf{E} G) \cdot v| \leq \sqrt{\epsilon} \left\| \sigma' \right\|_2 / \Phi(t^*)$.

3. *If* $\sin^2 \theta \Phi(t^*) > \epsilon$, *then* $\mathbf{E}(G \cdot v)^2 \leq \tilde{O}(b^2), \forall v \in \mathbb{S}^{d-1}$.

*Proof of Lemma 2.10.* We first prove the first item.

$$
\begin{aligned}
\mathbf{E} G^* &= \underset{x \sim N(0,I)|_{\{x \mid w \cdot x > t^*\}}}{\mathbf{E}} (\sigma(w \cdot x - t^*) - \sigma(w^* \cdot x - t^*)) \operatorname{proj}_{w^\perp} x \\
&= \underset{x \sim N(0,I)}{\mathbf{E}} \mathbb{1}\{w \cdot x > t^*\} (\sigma(w \cdot x - t^*) - \sigma(w^* \cdot x - t^*)) \operatorname{proj}_{w^\perp} x / \Phi(t^*) \\
&= \underset{x \sim N(0,I)}{\mathbf{E}} \sigma'(w \cdot x - t^*) (\sigma(w \cdot x - t^*) - \sigma(w^* \cdot x - t^*)) \operatorname{proj}_{w^\perp} x / \Phi(t^*) \\
&= -b \left\| T_{\sqrt{a}} \sigma' \right\|_2^2 / \Phi(t^*) u
\end{aligned}
$$

Here, the last equation follows from Lemma 2.3.

We next prove the second item.

$$|\mathbf{E}(G^* - G) \cdot v| = \left| \underset{x \sim N(0,I)|_{\{x|w\cdot x > t^*\}}}{\mathbf{E}} (y(x) - \sigma(w^* \cdot x - t^*)) (x \cdot v)) \right|$$

$$= \left| \underset{x \sim N(0,I)}{\mathbf{E}} \mathbb{1}\{w \cdot x > t^*\} (y(x) - \sigma(w^* \cdot x - t^*)) (x \cdot v)) \right| / \Phi(t^*)$$

$$= \left| \underset{x \sim N(0,I)}{\mathbf{E}} \sigma'(w \cdot x - t^*) (y(x) - \sigma(w^* \cdot x - t^*)) (x \cdot v)) \right| / \Phi(t^*)$$

$$\leq \sqrt{\epsilon} \|\sigma'\|_2 / \Phi(t^*)$$

Here the last inequality follows from Lemma 2.4.

Finally, we control the variance of $G$. We have

$$\mathbf{E}(G \cdot v)^2 \leq 2\mathbf{E}((G - G^*) \cdot v)^2 + 2\mathbf{E}(G^* \cdot v)^2.$$

We bound the two terms separately.

$$\mathbf{E}(G^* \cdot v)^2 = \underset{x \sim N(0,I)|_{\{x|w\cdot x > t^*\}}}{\mathbf{E}} (\sigma(w \cdot x - t^*) - \sigma(w^* \cdot x - t^*))^2 (x \cdot v)^2$$

$$= \underset{x \sim N(0,I)}{\mathbf{E}} \mathbb{1}\{w \cdot x > t^*\} (\sigma(w \cdot x - t^*) - \sigma(w^* \cdot x - t^*))^2 (x \cdot v)^2 / \Phi(t^*)$$

$$\leq \underset{x \sim N(0,I)}{\mathbf{E}} \mathbb{1}\{w \cdot x > t^*\} ((w - w^*) \cdot x)^2 (x \cdot v)^2 / \Phi(t^*)$$

Recall that $w^* = aw + bu$, where $u \in \mathbb{S}^{d-1}, u \perp w, a, b \geq 0, a^2 + b^2 = 1$. We have

$$\mathbf{E}(G^* \cdot v)^2 \leq \underset{x \sim N(0,I)}{\mathbf{E}} \mathbb{1}\{w \cdot x > t^*\} \left((1-a)^2 (w \cdot x)^2 + b^2 (u \cdot x)^2\right) (x \cdot v)^2 / \Phi(t^*)$$

$$= \underset{x \sim N(0,I)}{\mathbf{E}} \mathbb{1}\{w \cdot x > t^*\}(1-a)^2 (w \cdot x)^2 (x \cdot v)^2 / \Phi(t^*) + \mathbb{1}\{w \cdot x > t^*\} \left(b^2 (u \cdot x)^2\right) (x \cdot v)^2 / \Phi(t^*)$$

$$\leq O(b^4) \underset{z,s}{\mathbf{E}} \mathbb{1}\{z > t^*\} z^2 s^2 / \Phi(t^*) + b^2 \underset{z,s,r}{\mathbf{E}} \mathbb{1}\{z > t^*\} s^2 r^2 / \Phi(t^*) \leq O(b^2)$$

In the second last inequality above, we use the fact that $1 - \cos\theta \leq O(\sin^2 \theta)$ and $\mathbf{E}_z \mathbb{1}\{z > t^*\} z^2 \leq O((t^*)^2 \phi(t))$ and $b \leq O(1/t^*)$.

We next bound the first term. We have

$$\mathbf{E}\left((G^* - G) \cdot v\right)^2 = \left| \underset{x \sim N(0,I)|_{\{x|w\cdot x > t^*\}}}{\mathbf{E}} (y(x) - \sigma(w^* \cdot x - t^*))^2 (x \cdot v)^2) \right|$$

$$= \underset{x \sim N(0,I)}{\mathbf{E}} (y(x) - \sigma(w^* \cdot x - t^*))^2 (x \cdot v)^2 \mathbb{1}\{w \cdot x > t^*\} / \Phi(t^*)$$

Let $M > 0$ be a threshold such that $\mathbf{E}_{z \sim N(0,1)} z^2 \mathbb{1}\{|z| > M\} \leq \epsilon$. Notice that if we set $y' = \text{sign}(y) \min\{|y|, M\}$, then we will only introduce at most $\epsilon$ error. So, we can without loss of generality, assume $|y| \leq M$. In particular, for ReLU activation, $M \leq O(\sqrt{\log(1/\epsilon)})$. Based on this, we obtain that

$$\mathbf{E}\left((G^* - G) \cdot v\right)^2 = \underset{x \sim N(0,I)}{\mathbf{E}} (y(x) - \sigma(w^* \cdot x - t^*))^2 (x \cdot v)^2 \mathbb{1}\{w \cdot x > t^*\} \mathbb{1}\{(x \cdot v) \leq M\} / \Phi(t^*)$$

$$+ \underset{x \sim N(0,I)}{\mathbf{E}} (y(x) - \sigma(w^* \cdot x - t^*))^2 (x \cdot v)^2 \mathbb{1}\{w \cdot x > t^*\} \mathbb{1}\{(x \cdot v) > M\} / \Phi(t^*)$$

$$\leq \text{opt} M^2 / \Phi(t^*) + 4M^2 \underset{x \sim N(0,I)}{\mathbf{E}} (x \cdot v)^2 \mathbb{1}\{w \cdot x > t^*\} \mathbb{1}\{(x \cdot v) > M\} / \Phi(t^*)$$

$$\leq \epsilon M^2 / \Phi(t^*) + 4\epsilon M^2 \leq \sin^2\theta M^2 = \tilde{O}(b^2) .$$

$\square$

## B.9 Proof of Lemma 2.11

In this section, we present the proof of Lemma 2.11 and the corresponding hypothesis selection algorithm. For convenience, we first state a formal version of Lemma 2.11 as follows.

**Lemma B.9** (Hypothesis Selection with Queries). *Let $D$ be a distribution over $\mathbb{R}^d \times \mathbb{R}$ and let $D_x$ be the marginal distribution of $D_x$. There is an algorithm that, on input a list of hypotheses $h_1, \ldots, h_k$ such that for $i \in [k]$, $h_i : \mathbb{R}^d \to \mathbb{R}$, $\mathbf{E}_{x \sim D}\, h_i^2(x)$ exists, it makes $\mathrm{poly}(k)$ queries and returns a hypothesis $\hat{h}$ such that $\mathrm{err}(\hat{h}) \leq O(\min_{i \in [k]} \mathbf{E}_{(x,y) \sim D}(y - h_i(x))^2)$.*

As a subroutine, we will use later, we present the following well-known median-of-mean estimator.

**Lemma B.10** (Median of Means Estimation). *Let $G \in \mathbb{R}^d$ be a random vector such that for every $i \in [d]$, $\mathbf{E}(G \cdot x_i)^2 \leq B^2$. Then there is an estimator that takes $M = O(\log(1/\delta)dm)$, i.i.d. samples from $G$ and computes a vector $g \in \mathbb{R}^d$ such that with probability at least $1 - \delta$, $\|g - \mathbf{E}\,G\| \leq B/\sqrt{m}$.*

In this hypothesis selection problem, we only need to use the version for $d = 1$. Roughly speaking, if a random variable has a bounded variance, then very few samples suffice to estimate its mean. We present the following algorithm.

---

**Algorithm 2** HYPOTHESISSELECTION(Select a good hypothesis from a list of hypothesis)

---

1: **Input:** $h_1, \ldots, h_k$ such that for $i \in [k]$, $h_i : \mathbb{R}^d \to \mathbb{R}$,, $D_x$ a marginal distribution over $\mathbb{R}^d$
2: **Output:** $\hat{h} : \mathbb{R}^d \to \mathbb{R}$, such that $\mathrm{err}(\hat{h}) \leq O(\min_{i \in [k]} \mathbf{E}_{(x,y) \sim D}(y - h_i(x))^2)$ with non-trivial probability.
3: Let $d(x)$ be the density function of $D_x$ at $x$
4: Create an empty graph $G$ with a set of node $[k]$
5: **for** each $(i, j), i \neq j \in [k]$ **do**
6:     For each pair of $(i, j)$ create a hypothesis $g_{ij}(x) = \frac{h_i(x) - h_j(x)}{\|h_i - h_j\|}$
7:     Denote by $D_{ij}$ the distribution over $x$ with density proportional to $g_{ij}^2(x)d(x)$
8:     Use the median of mean method to estimate $\mathbf{E}_{x \sim D_{ij}}(y(x) - \frac{h_i + h_j}{2})g_{ij}(x)/g_{ij}^2(x)$ with $\mathrm{poly}(k)$ samples
9:     If the estimated result is more than $\|h_i - h_j\|$, draw an edge from $i$ to $j$
10: Return any $h_i$ such that $i$ is in a source strongly connected component of $G$

---

*Proof of Lemma 2.11.* We first observe that

$$\mathrm{err}(h_i) - \mathrm{err}(h_j) = 2 \mathop{\mathbf{E}}_{x \sim D_x} y(h_j - h_i) + \|h_i\|^2 - \|h_j\|^2 = 2 \mathop{\mathbf{E}}_{x \sim D_x}(y - \frac{h_i + h_j}{2})(h_j - h_i),$$

which means to compare the error of $h_i, h_j$, it is sufficient to check the sign of $\mathbf{E}_{x \sim D_x}(y - \frac{h_i + h_j}{2})(h_j - h_i)$. And this is equivalent to check the sign of $\mathbf{E}_{x \sim D_x}(y - \frac{h_i + h_j}{2})g_{ij}$. We remark that by our construction, $\|g_{ij}\|_2 = 1$. Notice that

$$\mathop{\mathbf{E}}_{x \sim D_{ij}}(y - \frac{h_i + h_j}{2})g_{ij}(x)/g_{ij}^2(x) = \mathop{\mathbf{E}}_{x \sim D_x} g_{ij}(x)\frac{(y - \frac{h_i + h_j}{2})}{g_{ij}^2(x)}g_{ij}^2(x) = \mathop{\mathbf{E}}_{x \sim D_x}(y - \frac{h_i + h_j}{2})g_{ij}(x).$$

Furthermore, consider the variance of the above quantity, we have

$$\mathop{\mathbf{E}}_{x \sim D_{ij}}(y - \frac{h_i + h_j}{2})^2 g_{ij}^2(x)/g_{ij}^4(x) = \mathop{\mathbf{E}}_{x \sim D_x}(y - \frac{h_i + h_j}{2})^2 \frac{g_{ij}^2(x)}{g_{ij}^4(x)}g_{ij}^2(x) = \left\|(y - \frac{h_i + h_j}{2})\right\|_2^2.$$

By Lemma B.10, we know that with $\mathrm{poly}(k)$ samples we are able to estimate $\mathbf{E}_{x \sim D_x}(y - \frac{h_i + h_j}{2})g_{ij}(x)$, with error $\left\|(y - \frac{h_i + h_j}{2})\right\|/\mathrm{poly}(k)$. We consider a fixed pair of $(h_i, h_j)$. Assume $\left\|(y - \frac{h_i + h_j}{2})\right\| \geq \|h_i - h_j\|$, then we have

$$\|h_i - y\| \leq \left\|(y - \frac{h_i + h_j}{2})\right\| + \frac{\|(h_i - h_j)\|}{2} \leq \frac{3}{4}(\|h_i - y\| + \|h_j - y\|),$$

which implies $\|h_i - y\| \leq 3 \|h_j - y\|$. By symmetry, we have $\|h_j - y\| \leq 3 \|h_i - y\|$. This implies if $\|h_i - y\|^2 \geq 10 \|h_j - y\|^2$, then we must have $\left\|(y - \frac{h_i+h_j}{2})\right\| \leq \|h_i - h_j\|$. We next prove that if $\|h_i - y\|^2 \geq 10 \|h_j - y\|^2$, then there must be an edge from $j$ to $i$ but no edge from $i$ to $j$. Recall that

$$\operatorname*{\mathbf{E}}_{x \sim D_x} (y - \frac{h_i + h_j}{2})(h_j - h_i) = \operatorname*{\mathbf{E}}_{x \sim D_x} y(h_j - h_i) - \frac{\|h_j\|^2 - \|h_i\|^2}{2}$$

$$= \frac{1}{2}(\|h_i - y\|^2 - \|h_j - y\|^2 + \|h_j\|^2 - \|h_i\|^2) - \frac{\|h_j\|^2 - \|h_i\|^2}{2} \geq \frac{9}{2} \|h_j - y\|^2$$

Since

$$\|h_i - h_j\| \geq \|h_i - y\| - \|h_j - y\| \geq (\sqrt{10} - 1) \|h_j - y\|,$$

we have $\|h_j - y\| \leq \|h_i - h_j\|/2$. This gives

$$\operatorname*{\mathbf{E}}_{x \sim D_x} (y - \frac{h_i + h_j}{2})(h_j - h_i) \geq \frac{9}{8} \|h_i - h_j\|^2.$$

Thus, when $\|h_i - y\|^2 \geq 10 \|h_j - y\|^2$, we have $\mathbf{E}_{x \sim D_x}(y - \frac{h_i+h_j}{2})g_{ji} \geq 9 \|h_i - h_j\|/8$ and the estimation of this is larger than $\|h_i - h_j\|$ with high probability. Thus, there must be an edge from $j$ to $i$ and no edge from $i$ to $j$.

On the other hand, if $\left\|(y - \frac{h_i+h_j}{2})\right\| \geq k^2 \|h_i - h_j\|$, then we have

$$\|h_i - y\| \leq \left\|(y - \frac{h_i + h_j}{2})\right\| + \frac{\|(h_i - h_j)\|}{2} \leq \left(\frac{1}{2} + \frac{1}{4k^2}\right)(\|h_i - y\| + \|h_j - y\|),$$

which implies $\|h_i - y\| \leq (1 + 1/2k^2) \|h_j - y\|$. By symmetry, we have $\|h_j - y\| \geq (1 - 1/2k^2) \|h_i - y\|$. If $\|h_i - y\|^2 \leq \|h_j - y\|^2$, then we have $\mathbf{E}\, yg_{ji} \leq 0$. Since we have estimated $\mathbf{E}\, yg_{ji}$ up to error $\left\|y - \frac{h_i + h_{i+1}}{2}\right\|/k^2$, this implies unless $\|h_i - y\|$ and $\|h_j - y\|$ are within a factor of $1 \pm 1/2k^2$ of each other, there will be no edge from $j$ to $i$. Thus, if there is an edge from $j$ to $i$, then it must be the case that $\|h_i - y\| \geq \|h_j - y\| (1 - 1/2k^2)$.

Now, we consider any source strongly connected $S$ in $G$. For every pair of $(i, j)$ in $S$, there is a cycle $C$ that contains $(i, j)$. Without loss of generality, we write $C = (1 \rightarrow 2, \ldots, m \rightarrow 1)$, where $m \leq k$. We will show that every $h_i, i \in C$ has similar error. We prove this via contradiction. Assuming there is a pair of $(i, j)$ in the cycle such that $\|y - h_i\|^2 \geq 25 \|y - h_i\|^2$, then there must be an edge from $i$ to $j$. Without loss of generality, we assume $i = m$ and $j = 1$, otherwise we can prove the same statement over a smaller cycle. Since for every $i \leq m - 1$, there is an edge from $i$ to $i + 1$, this implies

$$\|h_{m-1} - y\| \geq \left(1 - \frac{1}{2k^2}\right)^k \|h_1 - y\| \geq \|h_1 - y\|/2 \geq \frac{5}{2} \|h_m - y\|,$$

which implies that there should not be an edge from $m - 1$ to $m$. Thus, for each pair of $(i, j)$ in a cycle $C$, $\|h_i - y\|^2 \leq 25 \|h_j - y\|^2$. This implies that any source strongly connected component $S$ cannot contain a vertex $j$ such that $\|y - h_j\|^2 \geq 25 \min_{i \in [k]} \mathbf{E}_{(x,y) \sim D}(y - h_i(x))^2$. To see why this is true, we consider two cases. Let $i^*$ be the vertex such that $\|h_{i^*} - y\| = \min_{i \in [k]} \mathbf{E}_{(x,y) \sim D}(y - h_i(x))^2$. If $S$ contains $i^*$, since $S$ is a strongly connected component, each pair of $(i, i^*)$ is contained within some cycle and their error must be within a factor of 25. On the other hand, if $i^* \notin S$ and $S$ contains some $j$ such that $\|y - h_j\|^2 \geq 25 \min_{i \in [k]} \mathbf{E}_{(x,y) \sim D}(y - h_i(x))^2$, which implies that there must be an edge from $i^*$ to $j$ and $S$ cannot be a source strongly connected component. Notice that every time we make a comparison between $h_i, h_j$, we only use $\operatorname{poly}(k)$ samples, thus the total number of queries we make is $\operatorname{poly}(k)$.

Since for every pair of nodes $(i, j)$, there is a cycle such that $(i, j)$ are both in the cycle. In this case, $\|y - h_i\|$ and $\|h_j - y\|$ are within a factor. Let $i^*$ be the index such that $\|h_{i^*} - y\|$ is minimized. We claim that $C$ does not contain some $j$ such that $\|h_j - y\|^2 \geq 10 \|h_{i^*} - y\|^2$. If so, then there must be an edge from $i^*$ to $j$. Since $C$ is a source strongly connected component, then $i^* \in C$, which implies that $(i, i^*)$ must be in the same cycle, which gives a contradiction. $\qquad\square$

## B.10 Proof of Theorem B.11

Finally, we give a query-optimal learning algorithm that solves the general ReLU regression problem for the special case where $\|W^*\| = 1$ and $t^* > 0$ is given.

**Theorem B.11** (Query Learning for Spherical ReLU). *Consider the problem of agnostic PAC learning GLM with membership queries under the Gaussian distribution. Suppose the optimal activation function $\sigma = \sigma_{t^*}^*$ is of the form $\sigma(w^* \cdot x - t^*)$, where $\sigma$ is the ReLU function, $w^* \in \mathbb{S}^{d-1}$ and $t^* \in \mathbb{R}$ is known, there is an algorithm such that for every labeling function $y(x)$ and for every $\epsilon, \delta \in (0, 1)$, it makes $M = \tilde{O}_\delta(\min\{1/p, 1/\epsilon\} + d \cdot \mathrm{polylog}(1/\epsilon))$ memberships queries, runs in $\mathrm{poly}(d, M)$ time, where $p = \Phi(t^*)$ is the bias of the optimal activation function $\sigma^*$, and outputs an $\hat{h} = \sigma(\hat{w} \cdot x - t^*)$, $\hat{w} \in \mathbb{S}^{d-1}$, such that with probability at least $1 - \delta$, $\mathrm{err}(\hat{h}) \leq O(\mathrm{opt}) + \epsilon$.*

---

**Algorithm 3** SPHERICALLEARNING(Learn $w^*$ over the unit sphere)

---

1: **Input:** error parameter $\epsilon \in (0, 1)$, confidence parameter $\delta \in (0, 1)$
2: **Output:** $\hat{h} : \mathbb{R}^d \to \mathbb{R}$, such that $\mathrm{err}(\hat{h}) \leq O(\epsilon)$ with non-trivial probability.
3: Call INITIALIZATION$(t^*)$ to get $w^{(0)}$ and return $\hat{h}(x) = 0$ if no $w^{(0)}$ is returned.
4: **for** $i = 0, \ldots, T - 1$ **do**
5:     **for** $j = 1, \ldots, \tilde{O}(d)$ **do**
6:         Generate $z^{(j)} > 0$ with probability $\phi(t^* + z^{(i)})/\Phi(t^*)$ and $u^{(j)} \perp w^{(i)} \sim N(0, I)$
7:         Query $y(x^{(j)})$, where $z^{(j)} w^{(i)} + u^{(j)}$
8:         Estimate $\mathbf{E} \frac{1}{m} \sum_{j=1}^m \left(\sigma(w^{(i)} \cdot x^{(j)} - t^*) - y(x^{(j)})\right) u^{(j)}$ via median of mean and get $g^{(i)}$
9:     $w^{(i+1)} = \mathrm{proj}_{\mathbb{S}^{d-1}} \left(w^{(i)} + \mu g^{(i)}\right)$
10: **return** $w^{(T)}$
11: **procedure** INITIALIZATION(Find a warm start $w^{(0)}$ given $t^*$)
12:     **Input:** $t^* > 0$
13:     **Output:** $w^{(0)} \in \mathbb{S}^{d-1}$ such that $\theta(w^{(0)}, w^*) \leq 1/(2t^*)$ or assert $\mathrm{err}(0) \leq O(\epsilon)$.
14:     **if** $\Phi(t^*)/(t^*)^2 \leq O(\epsilon)$ **then**
15:         Assert $\mathrm{err}(0) \leq O(\epsilon)$.
16:     **else**
17:         Run the initialization algorithm for query learning halfspaces (Algorithm 2/Algorithm 5 in [DKM24]) by simulating binary membership query with $\bar{y} := \mathbb{1}\{y > c/(t^*)\}$ for small constant $c > 0$ and denote the return of the algorithm by $w^{(0)}$.

---

*Proof of Theorem B.11.* In each round of Algorithm 3, we use $\theta_i$ to denote $\theta(w^{(i)}, w^*)$ and we denote by $G_i^* \in \mathbb{R}^d$ the random vector be the random vector $\left(\sigma(w^{(i)} \cdot x - t^*) - \sigma(w^* \cdot x - t^*)\right) \mathrm{proj}_{(w^{(i)})^\perp} x$ and $G_i$ be the random vector $\left(\sigma(w^{(i)} \cdot x - t^*) - y(x)\right) \mathrm{proj}_{(w^{(i)})^\perp} x$, where $x \sim N(0, I)\mid_{\{x \mid w^{(i)} \cdot x > t^*\}}$.

By Lemma 2.8 and Lemma 2.9, we know that the subroutine INITIALIZATION takes $\tilde{O}(1/p + d \log(1/\epsilon))$ queries and output a unit vector $w^{(0)}$ such that with probability at least $\log(1/\Phi(t^*))$, $\sin(\theta_0/2) \leq 1/t^*$. In the rest of the proof, we assume INITIALIZATION succeeds. Let $\alpha = 50$. By Lemma 2.7, let $\alpha = 50$, we know that if $\theta_i \leq \theta_0$, then $\left\|T_{\sqrt{\cos \theta_i}} \sigma'\right\|_2^2 \geq \|\sigma'\|_2^2 / \alpha$. Let $\phi^* \in (0, \pi/2)$ such that $\sin^2 \phi^* = 20\alpha^2 \epsilon/(\pi \|\sigma'\|_2^2)$. Lemma 2.6 implies that if $\theta_i \leq \phi^*$, then $\mathrm{err}(w) \leq O(\alpha^2 \epsilon) = O(\epsilon)$, since $\alpha = O(1)$.

Recall that for activation function $\sigma(z) = ReLU(z - t^*)$, we have $\|\sigma'\|_2^2 = \Phi(t^*)$. We will show that for if $\theta_i \leq \theta_0$ and $\theta_i \geq \phi^*$, then with high probability, $\theta_{i+1} \leq (1 - 1/C)\theta_i$ for some constant $C > 1$. Write $w^* = aw^{(i)} + bu$, where $u \in \mathbb{S}^{d-1}, a, b \geq 0, a^2 + b^2 = 1$. By Lemma 2.10, we known that

$$\mathbf{E}\, G_i^* = \mathrm{proj}_{(w^{(i)})^\perp} \nabla_w \ell(w^{(i)})/\Phi(t^*) = b \left\|T_{\sqrt{a}} \sigma'\right\|_2^2 u/\Phi(t^*)$$

$$\|(\mathbf{E}\, G^* - \mathbf{E}\, G)\| = \|\mathrm{proj}_{w^\perp}(\nabla_w \mathrm{err}(w) - \nabla_w \ell(w))\| / \Phi(t^*) \leq \sqrt{\epsilon} \|\sigma'\|_2 / \Phi(t^*)$$

By Lemma 2.6, we know that if $\phi^* < \theta_i \leq \theta_0$, then $\|(\mathbf{E}\,G^* - \mathbf{E}\,G)\| \leq \|\mathbf{E}\,G_i\|/20$. This implies that

$$\mathbf{E}\,G_i \cdot u = \mathbf{E}\,G_i \cdot u = \mathbf{E}\,G_i^* \cdot u + \mathbf{E}(G_i - G_i^*) \cdot u \geq \frac{19}{20}\|\mathbf{E}\,G_i\| \geq 19b\alpha^{-1}/20.$$

Furthermore, since $\|\mathbf{E}\,G\| \leq \frac{21}{20}\|\mathbf{E}\,G_i\|$, Lemma 2.5, we know that by estimating $G$ upto error $\|\mathbf{E}\,G\|/40$, then $\theta_{i+1} \leq (1 - 1/C)\theta_i$ for some large constant $C > 0$. By Lemma 2.10, we know that $\mathbf{E}(G \cdot v)^2 \leq O(b^2 \log(1/\epsilon)), \forall v \in \mathbb{S}^{d-1}$. By Lemma B.10, we know that in each round of update, we only need $\tilde{O}(d)$ queries. Since we only need $\tilde{O}(\log(1/\epsilon))$ rounds to make $\theta_i \leq \theta^*$, the query complexity of Algorithm 3 is $\tilde{O}(1/\Phi(t^*) + d\mathrm{poly}\log(1/\epsilon))$. $\qquad\square$

## C   Omitted Proofs from Section 3

In this section, we provide the missing details for solving the general ReLU regression problem.

### C.1   Proof of Lemma 3.2

We start with the initialization algorithm. We provide the proof of Lemma 3.2 and its restatement Lemma C.1 as follows. The key point of the lemma is that if we have a reasonable initial knowledge about $r^*, t^*$, we are still able to do the initialization.

**Lemma C.1** (Initialization with Raw Knowledge). *Let $\sigma$ be the ReLU activation function. Let $h^* = r^*\sigma(\bar{w}^* \cdot x - \bar{t}^*)$ be the optimal hypothesis. Suppose that $(r^*)^2 V(\bar{t}^*) \geq \Omega(\epsilon)$, there is an algorithm such that given parameter $r, t > 0$ such that $r \leq r^* \leq 2r$ and $|t - \bar{t}^*| \leq 1/\log(R^2/\epsilon)$, it makes $M = \tilde{O}(1/p + d\log(R^2/\epsilon))$, runs in $\mathrm{poly}(d, M)$ time, and with probability at least $1/\log(1/p)$, outputs some $w^{(0)} \in \mathbb{S}^{d-1}$ such that $\sin(\theta(w^{(0)}, w^*)/2) \leq \min\{1/\bar{t}^*, 1/2\}$.*

*Proof of Lemma 3.2.* We consider the truncated label $\bar{y}(x) = \mathbb{1}\{y(x) > cr/t\}$, for a suitably small constant $c > 0$. We will show that the truncated label $\bar{y}(x)$ can be seen as generated from the halfspace $h^*(x) = \mathrm{sign}(w^* \cdot x - \bar{t}^*)$ corrupted with adversarial label noise with level at most $\Phi(\bar{t}^*)/C$ for some large enough constant $C$. By the assumption that $(r^*)^2 V(\bar{t}^*) \geq \Omega(\epsilon)$, we have $\bar{t}^* \leq O(\sqrt{\log(R^2/\epsilon)})$, otherwise, $h(x) \equiv 0$ has error $O(\epsilon)$. On the other hand, we can assume $\bar{t}^* \geq 1$, otherwise, estimating $\mathbf{E}_{x \sim N(0,I)}\,y(x)x$ with constant error is enough to get a $w^{(0)}$ such that $\sin\theta(w^{(0)}, w^*) \leq 1/C$. With these assumptions, we use a similar argument as we did for the proof of Lemma 2.8 but with a more careful analysis.

We partition $\mathbb{R}^d$ into three regions $I_1 := \{x \mid w^* \cdot x > \bar{t}^* + 2c/t\}, I_2 := \{x \mid w^* \cdot x < \bar{t}^*\}$ and $J := \{x \mid \bar{t}^* < w^* \cdot x < \bar{t}^* + 2c/t\}$. Since $\mathrm{opt} = \mathbf{E}_{x \sim N(0,I)}\left(r^*\sigma(w^* \cdot x - \bar{t}^*) - y\right)^2/2 < \epsilon$, we have

$$\begin{aligned}
\epsilon &\geq \mathop{\mathbf{E}}_{x \sim N(0,I)} \left(r^*\sigma(w^* \cdot x - \bar{t}^*)\right)^2/2 \\
&\geq \mathop{\mathbf{E}}_{x \sim N(0,I)} \mathbb{1}\{x \in I_1 \cup I_2\}\left(r^*\sigma(w^* \cdot x - \bar{t}^*) - y\right)^2/2 \\
&\geq \mathop{\mathbf{E}}_{x \sim N(0,I)} \left(\mathbb{1}\{x \in I_1, y(x) < cr/t\} + \mathbb{1}\{x \in I_2, y(x) > cr/t\}\right)\left(r^*\sigma(w^* \cdot x - \bar{t}^*) - y\right)^2/2 \\
&\geq \frac{c^2 r^2}{2t^2} \mathop{\mathbf{Pr}}_{x \sim N(0,I)} (x \in I_1 \cup I_2, y(x) \neq \bar{y}(x))
\end{aligned}$$

Since $\mathbf{E}_{x \sim N(0,I)}(r^*)^2\sigma(w^* \cdot x - \bar{t}^*)^2 \geq C\epsilon$, we obtain that

$$\mathop{\mathbf{Pr}}_{x \sim N(0,I)} (x \in I_1 \cup I_2, y(x) \neq \bar{y}(x)) \leq \frac{2t^2}{r^2 c^2}\epsilon \leq \frac{2t^2\,\mathbf{E}_{x \sim N(0,I)}(r^*)^2\sigma(w^* \cdot x - \bar{t}^*)^2}{Cc^2} \leq \Phi(\bar{t}^*)/C_1,$$

for some large enough $C_1 > 0$. Here, we use the fact that $r \leq r^* \leq 2r, \bar{t}^* \leq O(\sqrt{\log(R^2/\epsilon)})$ and $|t - \bar{t}^*| \leq 1/\log(R/\epsilon)$.

On the other hand,

$$\mathop{\mathbf{Pr}}_{x \sim N(0,I)} (x \in J, y(x) \neq \bar{y}(x)) \leq \mathop{\mathbf{Pr}}_{x \sim N(0,I)} (x \in J) \leq \frac{2c}{t}\frac{1}{\sqrt{2\pi}}\exp\left(-\frac{(\bar{t}^*)^2}{2}\right) \leq 10c\Phi(\bar{t}^*).$$

Thus,

$$\Pr_{x \sim N(0,I)} (\bar{y}(x) \neq \text{sign}(w^* \cdot x - \bar{t}^*)) = \Pr_{x \sim N(0,I)} (x \in J, y(x) \neq \bar{y}(x)) + \Pr_{x \sim N(0,I)} (x \in I_1 \cup I_2, y(x) \neq \bar{y}(x))$$
$$\leq \Phi(\bar{t}^*)/C_1 + 10c\Phi(\bar{t}^*) = \Phi(\bar{t}^*)/C''$$

for some large enough $C'' > 0$. By Lemma 2.9, we are able to efficiently obtain a direction $w^0$ that satisfies the statement of Lemma 3.2. $\square$

## C.2 Proof of Lemma 3.3

We next present the proof of Lemma 3.3 and its restatement Lemma C.2 to decompose the noiseless error for the general ReLU regression problem.

**Lemma C.2.** *Consider the problem of agnostic ReLU regression with queries. Let $r > 0, w \in \mathbb{S}^{d-1}, t > 0$ and $\theta(w, \bar{w}^*) = \theta$. If $t^* \leq O(\sqrt{\log(R^2/\epsilon)})$ and $|t - t^*| \leq 1/\log(R^2/\epsilon)$, then*

$$\ell(r, w, t) \leq (r^*)^2 \int_0^\theta \sin s \left\| T_{\sqrt{\cos s}} \sigma'(z - \bar{t}^*) \right\|_2^2 ds + \mathbb{E}_{z \sim N(0,I)} (\sigma(rz - t) - \sigma(r^*z - t^*))^2.$$

*Proof of Lemma 3.3.* To simplify the notation, we denote by $h' = h(r^*, w, t^*)$. Notice that

$$\ell(r, \bar{w}, t) = \frac{1}{2} \mathbb{E}_{x \sim N(0,I)} (h - h^*)^2 = \frac{1}{2} \mathbb{E}_{x \sim N(0,I)} (h - h' + h' - h^*)^2 \leq \mathbb{E}_{x \sim N(0,I)} (h - h')^2 + (h' - h^*)^2$$

On the one hand, we have

$$\mathbb{E}_{x \sim N(0,I)} (h - h^*)^2 = \mathbb{E}_{x \sim N(0,I)} (\sigma(rw \cdot x - t) - \sigma(r^*w \cdot x - t^*))^2 = \mathbb{E}_{z \sim N(0,1)} (\sigma(rz - t) - \sigma(r^*z - t^*))^2$$

On the other hand, we have

$$\mathbb{E}_{x \sim N(0,I)} (h - h^*)^2 = \mathbb{E}_{x \sim N(0,I)} (\sigma(r^*w \cdot x - t^*) - \sigma(r^*w^* \cdot x - t^*))^2$$
$$= (r^*)^2 \mathbb{E}_{x \sim N(0,I)} (\sigma(w \cdot x - \bar{t}^*) - \sigma(w^* \cdot x - \bar{t}^*))^2$$
$$= (r^*)^2 \int_0^\theta \sin s \left\| T_{\sqrt{\cos s}} \sigma'(z - \bar{t}^*) \right\|_2^2 ds.$$

Here, the last equation follows Lemma 2.2. This implies

$$\ell(r, w, t) \leq (r^*)^2 \int_0^\theta \sin s \left\| T_{\sqrt{\cos s}} \sigma'(z - \bar{t}^*) \right\|_2^2 ds + \mathbb{E}_{z \sim N(0,I)} (\sigma(rz - t) - \sigma(r^*z - t^*))^2.$$

$\square$

## C.3 Overview of Query Learning Algorithm

In this section, we present the analysis for the algorithm corresponding to Theorem 3.1. For convenience, we present Algorithm 4 as follows and give a brief overview of it.

**Theorem C.3.** *Consider the problem of agnostic general ReLU regression with queries under the Gaussian distribution. There is an algorithm such that for every labeling function $y(x)$ and for every $\epsilon, \delta \in (0,1)$, it makes $M = \tilde{O}_\delta(\min\{1/p, R^2/\epsilon\} + d \cdot \text{polylog}(R^2/\epsilon))$ queries, runs in $\text{poly}(d, M)$ time, where $p = \Phi(\bar{t}^*)$ is the bias of the optimal activation function, and outputs an $\hat{h}$ such that with high probability at least $1 - \delta$, $\text{err}(\hat{h}) \leq O(\text{opt}) + \epsilon$.*

We remark that the dependence on $R^2/\epsilon$ is due to the natural scaling of the squared $\ell_2$ loss. If we want to learn $R\sigma(w^* \cdot x - t^*)$ up to error $\epsilon$, this is equivalent to learning $\sigma(w^* \cdot x - t^*)$ to error $\epsilon/R^2$.

Given a ReLU activation, $h(r_i, w^{(i)}, t_i)$, define random vectors

$$G_i^* := \left( h(r_i, w^{(i)}, t_i)(x) - h^*(x) \right) \text{proj}_{(w^{(i)})^\perp}(x)$$

and its noisy version

$$G_i := \Big( h(r_i, w^{(i)}, t_i)(x) - y(x) \Big) \operatorname{proj}_{(w^{(i)})^\perp}(x) \,,$$

where $x \sim N(0, I) \mid_{w^{(i)} \cdot x > \bar{t}_i}$. Define random variables as follows

$$U_i^* := (h(r_i, w^{(i)}, t_i) - h^*)(w^{(i)} \cdot x), \ F_i^* := -(h(r_i, w^{(i)}, t_i) - h^*),$$

and denote by $U_i, F_i$, their noisy version, namely

$$U_i := (h(r_i, w^{(i)}, t_i) - y)(w^{(i)} \cdot x), \ F_i := -(h(r_i, w^{(i)}, t_i) - y) \,.$$

---

**Algorithm 4** QUERYLEARNING(Learn optimal ReLU with a warm start)

---

1: **Input:** $w^{(0)} \in \mathbb{S}^{d-1}$ : unit vector such that $\theta_0 \leq 1/\operatorname{polylog}(R^2/\epsilon)$. $r_0 > 0$ : such that $|r^* - r_0| \leq r^*/\operatorname{polylog}(R^2/\epsilon)$, $t_0 : |t_0 - t^*| \leq \operatorname{polylog}(R^2/\epsilon)$.
2: **Output:** $\hat{h} : \mathbb{R}^d \to \mathbb{R}$, such that $\operatorname{err}(\hat{h}) \leq O(\epsilon)$ with non-trivial probability.
3: $B_0 := r_0^2/\operatorname{polylog}(R^2/\epsilon)$
4: **for** $i = 0, \dots, T-1$ **do**
5:      Generate $\operatorname{polylog}(R^2/\epsilon)$ samples $x^{(j)} \sim N(0, I) \mid \{w^{(i)} \cdot x > t_i\}$. Query $y(x^{(j)})$ and use them to get an estimate $\hat{g}_i$ for $(\mathbf{E}\, U_i, \mathbf{E}\, F_i)$.
6:      **if** $\|\hat{g}_i\| \geq B_i \operatorname{polylog}(R^2/\epsilon)$ **then**
7:          Set $(r_{i0}, t_{i0}) = (r_i, t_i)$ and update $(r_{ij}, t_{ij}) = (r_{i(j-1)}, t_{i(j-1)}) - (\hat{g_{ij}})/\operatorname{polylog}(R^2/\epsilon)$ until $\|\hat{g}_{ij}\| \leq B_i \operatorname{polylog}(R^2/\epsilon)$
8:      $(r_{i+1}, t_{i+1}) \leftarrow (r_{ij}, t_{ij})$
9:      **for** $j = 1, \dots, \tilde{O}(d)$ **do**
10:          Generate $x^{(j)} \sim N(0, I) \mid \{w^{(i+1)} \cdot x > t_{i+1}\}$ and query $y(x^{(j)})$
11:      Estimate $\mathbf{E}\, G_i$ via median of mean and get $\hat{G}_i$
12:      $w^{(i+1)} = \operatorname{proj}_{\mathbb{S}^{d-1}} \Big( w^{(i)} - \mu \hat{G}_i \Big)$
13:      $B_{i+1} = (1 - \rho) B_i$
14: $\hat{w} = w^{(T)}$
15: Build a unit grid of size $1/\operatorname{polylog}(R^2/\epsilon)$ over the ball centered at $(r_T, t_T)$ and randomly select a pair $(\hat{r}, \hat{t})$ from the grid.
16: **return** $h(\hat{r}, \hat{w}, \hat{t})$

---

We now proceed with an intuitive explanation of Algorithm 4 and its analysis. The detailed analysis is carried out in the rest of the section. We are given a tuple of initial parameters $(r_0, w^{(0)}, t_0)$ such that each of them are close to the true parameters up to a $1 \pm 1/\operatorname{polylog}(R^2/\epsilon)$ factor. We remark that by Lemma 3.2, we are only able to get some $w^{(0)}$ such that $\theta_0 \leq 1/\bar{t}^*$, but as we will show in Appendix C.4, since the other parameters are close enough, by updating $w^{(0)}$ for $\log \log(R^2/\epsilon)$ rounds, we are able to achieve this guarantee. We will maintain an upper bound $B_i^2$ for $(r^*)^2 \sin^2 \theta$ and reduce this upper bound stably in each round of the algorithm. In our analysis, we will show that as long as the error from $(r_i, t_i)$ is within a $\operatorname{polylog}(R^2/\epsilon)$ factor of $B_i^2$, we are able to make roughly $d\operatorname{polylog}(R^2/\epsilon)$ queries to safely improve $w^{(i)}$ and decrease the angle $\theta_i$ by a small constant factor.

Regarding the parameters $(r_i, t_i)$, since they are close to $(r^*, t^*)$, the task for optimizing them can be approximately seen as optimizing the following quadratic function $Z(r, t) := \mathbf{E}_{x \sim N(0, I)} \left( (rz - t) - (r^* z - t^*) \right)^2 \mathbb{1}(z > \bar{t}^*)$. Such a quantity nearly characterizes the contribution of $(r_i, t_i)$ to the noiseless error as well as the contribution of $(r_i, t_i)$ to the variance of the gradient we use for updating $w^{(i)}$. As we will show in Appendix C.5, if $B_i^2 \Phi(\bar{t}^*) \leq Z(r_i, t_i)/\operatorname{polylog}(R^2/\epsilon)$, i.e., $\theta_i$ has already been updated in a reasonable range, then $(\mathbf{E}\, U_i, \mathbf{E}\, F_i)$ is very close to the gradient of $Z(r_i, t_i)$. Thus, we are able to tell whether $Z(r_i, t_i)$ is desirable by checking the norm of $(\mathbf{E}\, U_i, \mathbf{E}\, F_i)$ due to the nature of quadratic minimization. When $Z(r_i, t_i)$ is large, we use a standard gradient descent to update these parameters such that $Z(r_i, t_i)$ is controlled by $B_{i+1}$ within a polylogarithmic factor. This allows us to safely update $w^{(i)}$ via gradient descent. When $Z(r_i, t_i)$ is already close to $B_{i+1}$, we do not update it. We remark that due to the noise and the ill-condition of $Z$ for large $t^*$, the

step size we choose is $1/\mathrm{polylog}(R^2/\epsilon)$ instead of a constant. After at most $\mathrm{polylog}(R^2/\epsilon)$ rounds, $w^{(T)}$ is updated to a desirable accuracy. However, due to the presence of noise, we are only able to guarantee that $(r_T, t_T)$ is $O(\epsilon \mathrm{polylog}(R^2/\epsilon)/\Phi(\bar{t^*}))$ close to $(r^*, t^*)$ in terms of squared norm. Fortunately, there are only two parameters we need to worry about and they are already very close to $(r^*, t^*)$; randomly selecting a pair of parameters from their neighborhoods gives us a good hypothesis with enough probability.

## C.4  Omitted Details Regarding Angle Update

In this section, we provide the full details on the subroutine for updating the direction $w^{(i)}$. Specifically, we will consider the following random vectors in $\mathbb{R}^d$. Given a ReLU activation, $h(r_i, w^{(i)}, t_i)$, define random vectors $G_i^* := \left( h(r_i, w^{(i)}, t_i)(x) - h^*(x) \right) \mathrm{proj}_{(w^{(i)})^\perp}(x)$ and its noisy version $G_i := \left( h(r_i, w^{(i)}, t_i)(x) - y(x) \right) \mathrm{proj}_{(w^{(i)})^\perp}(x)$, where $x \sim N(0, I) \mid_{w^{(i)} \cdot x > \bar{t}_i}$.

### C.4.1  Proof of Lemma 3.4

We first give an evaluation of the mean and the noise analysis of $G$. We provide the proof of Lemma 3.4 and its restatement Lemma C.4.

**Lemma C.4.** *Consider the problem of agnostic ReLU regression with queries. Let $h(r_i, \bar{w}^{(i)}, t_i)$. Write $w^* = aw^{(i)} + bu$, where $a, b > 0, a^2 + b^2 = 1, u \in \mathbb{S}^{d-1}, u \perp \bar{w}^{(i)}$. Then the following statements hold.*

*1. If $|\bar{t}_i - \bar{t}^*| \leq 1/\log(R^2/\epsilon)$ and $b \leq 1/\bar{t}_i$, then*

$$\mathbf{E}\, G_i^* = -\alpha b r^* \left\| T_a \sigma'(z - \bar{t}_i) \right\|^2 u/\Phi(\bar{t}_i)$$

*where $1/2 < \alpha < 2$.*

*2. $|(\mathbf{E}\, G^* - \mathbf{E}\, G) \cdot v| \leq \sqrt{\epsilon} \left\| \sigma'(z - t_i) \right\|_2 / \Phi(\bar{t}_i), \forall v \in \mathbb{S}^{d-1}, v \perp w^{(i)}.$*

*Proof of Lemma 3.4.* We first consider the mean of $G_i^*$. By Fact A.5 and Fact A.8, we have

$$\mathbf{E}\, G_i^* = \mathop{\mathbf{E}}_{x \sim N(0, I)|_{\{w^{(i)} \cdot x > \bar{t}_i\}}} \left( h(r_i, w^{(i)}, t_i)(x) - h^*(x) \right) \mathrm{proj}_{w^\perp} x$$

$$= \mathop{\mathbf{E}}_{x \sim N(0, I)} \mathbb{1}\{w^{(i)} \cdot x > \bar{t}_i\} \left( h(r_i, w^{(i)}, t_i)(x) - h^*(x) \right) \mathrm{proj}_{(w^{(i)})^\perp} x/\Phi(\bar{t}_i)$$

$$= -\mathrm{proj}_{(w^{(i)})^\perp} \mathop{\mathbf{E}}_{x \sim N(0, I)} \mathbb{1}\{w^{(i)} \cdot x > \bar{t}_i\} h^*(x) x/\Phi(\bar{t}_i)$$

$$= -\mathop{\mathbf{E}}_{x \sim N(0, I)} \mathbb{1}\{w^{(i)} \cdot x > \bar{t}_i\} \mathbb{1}\{w^* \cdot x > \bar{t}^*\} br^* u/\Phi(\bar{t}_i)$$

$$= -\mathop{\mathbf{E}}_{x \sim N(0, I)} \mathbb{1}\{w^{(i)} \cdot x > \bar{t}_i\} \mathbb{1}\{w^* \cdot x > \bar{t}_i\} br^* u/\Phi(t_i)$$

$$+ \mathop{\mathbf{E}}_{x \sim N(0, I)} \mathbb{1}\{w^{(i)} \cdot x > \bar{t}_i\}(\mathbb{1}\{w^* \cdot x > \bar{t}_i\} - \mathbb{1}\{w^* \cdot x > t^*\}) br^* u/\Phi(\bar{t}_i)$$

$$= -br^* \left\| T_a \sigma'(z - \bar{t}_i) \right\|_2^2 u/\Phi(\bar{t}_i) + \mathop{\mathbf{E}}_{x \sim N(0, I)} \mathbb{1}\{w^{(i)} \cdot x > \bar{t}_i\}(\mathbb{1}\{w^* \cdot x > \bar{t}_i\} - \mathbb{1}\{w^* \cdot x > t^*\}) br^* u/\Phi(\bar{t}_i)$$

By Lemma 2.7, we know that $\left\| T_a \sigma'(z - \bar{t}_i) \right\|^2 \geq \Omega(1)\Phi(\bar{t}_i)$. Furthermore, we have

$$\left| \mathop{\mathbf{E}}_{x \sim N(0, I)} \mathbb{1}\{\bar{w}^{(i)} \cdot x > \bar{t}_i\}(\mathbb{1}\{\bar{w}^* \cdot x > \bar{t}_i\} - \mathbb{1}\{\bar{w}^* \cdot x > \bar{t}^*\}) \right| \leq \left| \mathop{\mathbf{E}}_{x \sim N(0, I)} (\mathbb{1}\{\bar{w}^* \cdot x > \bar{t}_i\} - \mathbb{1}\{\bar{w}^* \cdot x > \bar{t}^*\}) \right| = o(\Phi(\bar{t}_i)).$$

We conclude that $\mathbf{E}\, G_i^* = -\alpha b r^* \left\| T_a \sigma'(z - \bar{t}_i) \right\|^2 u/\Phi(\bar{t}_i)$ where $1/2 < \alpha < 2$.

We next analyze the noise term. For every $v \in \mathbb{S}^{d-1}$ and $v \perp w^{(i)}$, by Holder's inequality, we have

$$
\begin{aligned}
|\mathbf{E}(G_i^* - G_i) \cdot v| &= \left| \mathop{\mathbf{E}}_{x \sim N(0,I)|_{\{w^{(i)} \cdot x > \bar{t}_i\}}} (y(x) - h^*(x)) (x \cdot v)) \right| \\
&= \left| \mathop{\mathbf{E}}_{x \sim N(0,I)} \mathbb{1}\{w^{(i)} \cdot x > \bar{t}_i\} (y(x) - h^*(x)) (x \cdot v)) \right| / \Phi(\bar{t}_i) \\
&\leq \sqrt{\mathop{\mathbf{E}}_{x \sim N(0,I)} (y(x) - h^*(x))^2} \sqrt{\mathop{\mathbf{E}}_{x \sim N(0,I)} \mathbb{1}\{w^{(i)} \cdot x > \bar{t}_i\}(x \cdot v)^2} \\
&\leq \sqrt{\epsilon} \, \|\sigma'(z - \bar{t}_i)\|_2 / \Phi(\bar{t}_i).
\end{aligned}
$$

$\square$

### C.4.2 Proof of Lemma 3.5

We next provide the evaluation for the variance of $G$. We provide the proof of Lemma 3.5 and its restatement Lemma C.5.

**Lemma C.5.** *Let $h(r_i, \bar{w}^{(i)}, t_i)$ be a ReLU activation. Write $w^* = aw^{(i)} + bu$, where $a, b > 0, a^2 + b^2 = 1$, $u \in \mathbb{S}^{d-1}, u \perp \bar{w}^{(i)}$. If $|\bar{t}_i - \bar{t}^*| \leq 1/\log(R^2/\epsilon), C\epsilon/((r^*)^2\Phi(t^*)) \leq b^2 \leq 1/\bar{t}_i^2$, then*

$$
\mathbf{E}(G_i \cdot v)^2 \leq \tilde{O}\left( \mathop{\mathbf{E}}_{z \sim N(0,I)} (\sigma(r_i z - t_i) - \sigma(r^* z - t^*))^2 / \Phi(\bar{t}_i) + (r^*)^2 b^2 \right),
$$

*for every $v \in \mathbb{S}^{d-1}$, $v \perp w^{(i)}$.*

*Proof of Lemma 3.5.* For every $v \in \mathbb{S}^{d-1}$, $v \perp w^{(i)}$ we have

$$
\mathbf{E}(G_i \cdot v)^2 \leq 2 \, \mathbf{E}((G_i - G_i^*) \cdot v)^2 + 2 \, \mathbf{E}(G_i^* \cdot v)^2.
$$

We bound the two terms separately.

$$
\begin{aligned}
\mathbf{E}(G_i^* \cdot v)^2 &= \mathop{\mathbf{E}}_{x \sim N(0,I)|_{\{w^{(i)} \cdot x > t_i\}}} \left( h(r_i, w^{(i)}, t_i) - h^* \right)^2 (x \cdot v)^2 \\
&= \mathop{\mathbf{E}}_{x \sim N(0,I)} \mathbb{1}\{w^{(i)} \cdot x > \bar{t}_i\} \left( h(r_i, w^{(i)}, t_i) - h^* \right)^2 (x \cdot v)^2 / \Phi(\bar{t}_i)
\end{aligned}
$$

We next expand $\left( h(r_i, w^{(i)}, t_i) - h^* \right)^2$ as follows:

$$
\begin{aligned}
&\left( h(r_i, w^{(i)}, t_i) - h^* \right)^2 \\
&= \left( h(r_i, w^{(i)}, t_i) - h(r^*, w^{(i)}, t^*) + h(r^*, w^{(i)}, t^*) - h(r^*, w^*, t^*) \right)^2 \\
&\leq 2 \left( h(r_i, w^{(i)}, t_i) - h(r^*, w^{(i)}, t^*) \right)^2 + 2 \left( h(r^*, w^{(i)}, t^*) - h(r^*, w^{(i)}, t^*) \right)^2.
\end{aligned}
$$

For the first term, we have

$$
\begin{aligned}
&\mathop{\mathbf{E}}_{x \sim N(0,I)} \mathbb{1}\{w^{(i)} \cdot x > \bar{t}_i\} \left( h(r_i, w^{(i)}, t_i) - h(r^*, w^{(i)}, t^*) \right)^2 (x \cdot v)^2 / \Phi(\bar{t}_i) \\
&\leq \mathop{\mathbf{E}}_{z \sim N(0,I)} \mathbb{1}\{w^{(i)} \cdot x > \bar{t}_i\} \left( \sigma(r_i z - t_i) - \sigma(r^* z - t^*) \right)^2 / \Phi(\bar{t}_i)
\end{aligned}
$$

For the second term, notice that

$$
\left( \sigma(w^{(i)} \cdot x - \bar{t}^*) - \sigma(w^* \cdot x - \bar{t}^*) \right)^2 \leq \left( ((1-a)w^{(i)} - bu) \cdot x \right)^2 \leq 2(1-a)^2 (w^{(i)} \cdot x)^2 + 2b^2 (u \cdot x)^2.
$$

Since $(1 - a) = O(b^2)$ and $b = O(\bar{t}_i)$, this gives

$$\underset{x \sim N(0,I)}{\mathbf{E}} \mathbb{1}\{w^{(i)} \cdot x > \bar{t}_i\} \left(\sigma(w^{(i)} \cdot x - \bar{t}^*) - \sigma(w^* \cdot x - \bar{t}^*)\right)^2 (x \cdot v)^2 / \Phi(\bar{t}_i)$$

$$\leq \underset{x \sim N(0,I)}{\mathbf{E}} \mathbb{1}\{w^{(i)} \cdot x > \bar{t}_i\} \left(2(1 - a)^2 (w^{(i)} \cdot x)^2 + 2b^2(u \cdot x)^2\right)^2 (x \cdot v)^2 / \Phi(\bar{t}_i)$$

$$\leq O(b^4 \bar{t}_i^2 + b^2) = O(b^2)$$

This gives

$$\mathbf{E}(G_i^* \cdot v)^2 \leq O\left(\underset{z \sim N(0,I)}{\mathbf{E}} \left(\sigma(r_i z - t_i) - \sigma(r^* z - t^*)\right)^2 / \Phi(\bar{t}_i) + (r^*)^2 b^2\right).$$

Next, we bound the variance of the noisy term.

$$\mathbf{E}\left((G^* - G) \cdot v\right)^2 = \left|\underset{x \sim N(0,I)|_{\{w^{(i)} \cdot x > \bar{t}_i\}}}{\mathbf{E}} \left(y(x) - h^*(x)\right)^2 (x \cdot v)^2)\right|$$

$$= \underset{x \sim N(0,I)}{\mathbf{E}} \left(y(x) - h^*(x)\right)^2 (x \cdot v)^2 \mathbb{1}\{w^{(i)} \cdot x > \bar{t}_i\} / \Phi(\bar{t}_i)$$

Let $M > 0$ be a threshold such that $\mathbf{E}_{z \sim N(0,1)} z^2 \mathbb{1}\{|z| > M\} \leq \epsilon / R^2$. Notice that if we set $y' = \text{sign}(y) \min\{|y|, r_i M\}$, then we will only introduce at most $\epsilon$ error since $|r_i - r^*| \leq r^* / \log(R/\epsilon)$. So, we can without loss of generality, assume $|y| \leq M$. In particular, for ReLU activation, $M \leq O(\sqrt{\log(R^2/\epsilon)})$. Based on this, we obtain that

$$\mathbf{E}\left((G_i^* - G_i) \cdot v\right)^2 = \underset{x \sim N(0,I)}{\mathbf{E}} \left(y(x) - h^*(x)\right)^2 (x \cdot v)^2 \mathbb{1}\{w^{(i)} \cdot x > \bar{t}_i\} \mathbb{1}\{(x \cdot v) \leq M\} / \Phi(\bar{t}_i)$$

$$+ \underset{x \sim N(0,I)}{\mathbf{E}} \left(y(x) - h^*(x)\right)^2 (x \cdot v)^2 \mathbb{1}\{w^{(i)} \cdot x > \bar{t}_i\} \mathbb{1}\{(x \cdot v) > M\} / \Phi(\bar{t}_i)$$

$$\leq \text{opt} M^2 / \Phi(t^*) + 4M^2 \underset{x \sim N(0,I)}{\mathbf{E}} (x \cdot v)^2 \mathbb{1}\{w^{(i)} \cdot x > \bar{t}_i\} \mathbb{1}\{(x \cdot v) > M\} / \Phi(\bar{t}_i)$$

$$\leq \epsilon M^2 / \Phi(\bar{t}_i) + 4\epsilon M^2 / \Phi(\bar{t}_i) \leq \sin^2 \theta M^2 = O((r^*)^2 b^2 \log(R^2/\epsilon))$$

Thus, we conclude that

$$\mathbf{E}(G_i \cdot v)^2 \leq \tilde{O}\left(\underset{z \sim N(0,I)}{\mathbf{E}} \left(\sigma(r_i z - t_i) - \sigma(r^* z - t^*)\right)^2 / \Phi(\bar{t}_i) + (r^*)^2 b^2\right).$$

$\square$

### C.4.3 Progress on Angle Update

In this section, we analyze the progress made over the update of $w^{(i)}$. We present the following lemma for measuring the progress on $w^{(i)}$.

**Lemma C.6.** *Consider the problem of agnostic ReLU regression with queries. Let $h(r_i, w^{(i)}, t_i)$ be a ReLU activation function. Write $w^* = aw^{(i)} + bu$, where $a, b > 0, a^2 + b^2 = 1, u \in \mathbb{S}^{d-1}, u \perp w^{(i)}$. Suppose $|\bar{t}_i - \bar{t}^*| \leq 1/\log(R^2/\epsilon)$, $b \leq 1/\bar{t}_i$ and $|r_i - r^*| \leq r^*/\text{polylog}(R^2/\epsilon)$. Let $B_i > 0$ be a parameter that satisfies the following property:*

*1. $\mathbf{E}_{z \sim N(0,I)} \left(\sigma(r_i z - t_i) - \sigma(r^* z - t^*)\right)^2 \leq B_i^2 \Phi(\bar{t}^*)$,*

*2. $B_i^2 \Phi(\bar{t}^*) \geq \Omega(\epsilon)$.*

*Define $B_{i+1} = (1 - 1/C)B_i$ for some universal constant $C > 0$. If $(r^*)^2 \sin^2 \theta_i \leq B_i^2 \beta^2, 0 < \beta < 1/10$, then the update $w^{(i+1)} = \text{proj}_{\mathbb{S}^{d-1}}(w^{(i)} - \mu_i \hat{G}_i)$ satisfies $(r^*)^2 \sin^2 \theta_{i+1} \leq B_{i+1}^2 \beta^2$, where $\mu_i$ is a small constant and $\hat{G}_i$ is an estimation of $\mathbf{E} G_i$ with $\tilde{O}(d/\beta^2)$ samples.*

*Proof of Lemma C.6.* Write $\bar{w}^* = a\bar{w}^{(i)} + bu$, where $a, b > 0, a^2 + b^2 = 1, u \in \mathbb{S}^{d-1}, u \perp \bar{w}^{(i)}$. Since $\sin\theta_i \le 1/\bar{t}_i$, $|\bar{t}_i - \bar{t}^*| \le 1/\log(R^2/\epsilon)$ and $|r_i - r^*| \le r^*/\text{polylog}(R^2/\epsilon)$, by Lemma 3.5 and Lemma B.10, we know that with $\tilde{O}(d)$ samples of $G_i$, we have $\left\|\hat{G}_i - \mathbf{E}\, G_i\right\| \le \beta B_i/1000$. We consider two cases for $(r^*)^2 \sin^2\theta_i$. In the first case, we assume that $\beta^2 B_i^2/2 \le (r^*)^2 \sin^2\theta_i \le \beta^2 B_i^2$. By Lemma 3.4, we know that $\left\|\hat{G}_i - \mathbf{E}\, G_i\right\| \le \beta \left\|\mathbf{E}\, G_i\right\|/C_1$, for some large enough constant $C_1 > 0$. By Lemma 2.5, we know that there is a small constant $c_2 > 0$ such that $\sin\theta_{i+1} \le (1 - c_2)\sin\theta_i$, which implies that $(r^*)^2 \sin^2\theta_{i+1} \le \beta^2 B_{i+1}^2$.

In the second case, we have $(r^*)^2 \sin^2\theta_i \le \beta^2 B_i^2/2$. Notice that

$$2(\sin(\frac{\theta_{i+1}}{2}) - \sin(\frac{\theta_i}{2})) = \left\|\bar{w}^{(i+1)} - \bar{w}^*\right\| - \left\|\bar{w}^{(i)} - \bar{w}^*\right\| \le \left\|\bar{w}^{(i+1)} - \bar{w}^*\right\| \le \left\|\bar{w}^{(i)} + \mu_i\hat{G}_i - \bar{w}^{(i)}\right\| = \left\|\mu_i\hat{G}_i\right\|.$$

Then we have

$$\beta B_{i+1} - r^* \sin(\frac{\theta_{i+1}}{2}) = \beta B_{i+1} - r^* \sin(\frac{\theta_i}{2}) - r^*(\sin(\frac{\theta_{i+1}}{2}) - \sin(\frac{\theta_i}{2}))$$

$$\ge B_{i+1}\beta - \beta B_i/2 - r^*\left\|\mu_i\hat{G}_i\right\| > 0$$

Thus, in both cases, we have $(r^*)^2 \sin^2\theta_{i+1} \le \beta^2 B_{i+1}^2$. $\qquad\square$

We discuss the implication of Lemma C.6 before providing the details for updating the parameters $(r, t)$. In our application, we will choose $\beta = 1/\text{polylog}(R^2/\epsilon)$. As we discussed in Section 3, we have initial parameters $r_0, t_0$ such that $|r_0 - r^*| \le r^*/\text{polylog}(R^2/\epsilon)$ and $|\bar{t}_0 - \bar{t}^*| \le 1/\text{polylog}(R^2/\epsilon)$. This implies that

$$\mathbf{E}_{z\sim N(0,I)} \left(\sigma(r_0 z - t_0) - \sigma(r^* z - t^*)\right)^2$$

$$\le 2\mathbf{E}_{z\sim N(0,I)} \left((r_0 - r^*)\sigma(z - \bar{t}_0)\right)^2 + \mathbf{E}_{z\sim N(0,I)} (r^*)^2(\sigma(z - \bar{t}_i) - \sigma(z - \bar{t}^*))^2$$

$$\le O((r_0 - r^*)^2 V(\bar{t}^*) + (r^*)^2(\bar{t}_0 - \bar{t}^*)^2\Phi(\bar{t}^*)) \le (r^*)^2\Phi(\bar{t}^*)/\text{polylog}(R^2/\epsilon).$$

If $\bar{t}^*$ is large enough such that for $\theta \ge 1/\text{polylog}(R^2/\epsilon)$, $(r^*)^2 \sin\theta^2\Phi(\bar{t}^*) \le C\epsilon$, then by Lemma C.6, starting from $w^{(0)}$ and updating $w^{(0)}$ at most $\log\log(R^2/\epsilon)$ steps, we obtain a hypothesis with error $O(\epsilon)$. Thus, in the rest of the section, we will assume that $w^{(0)}$ satisfies $\sin\theta_0 \le 1/\text{polylog}(R^2/\epsilon)$.

### C.5 Omitted Details in $(r, t)$ Updates

Here we provide the details for updating the parameters $(r, t)$. For a given ReLU activation $h(r_i, w^{(i)}, t_i)$, we define the following random variables:

$$U_i^* := (h(r_i, w^{(i)}, t_i) - h^*)(w^{(i)} \cdot x), \quad F_i^* := -(h(r_i, w^{(i)}, t_i) - h^*),$$

and denote by $U_i, F_i$, their noisy version, namely

$$U_i := (h(r_i, w^{(i)}, t_i) - y)(w^{(i)} \cdot x), \quad F_i := -(h(r_i, w^{(i)}, t_i) - y).$$

Here, $x \sim N(0, I)\,|_{w^{(i)}\cdot x > \bar{t}_i}$.

When the direction $w$ is fixed, we will also use $U(r, t), F(r, t)$ defined as follows for convenience:

$$U(r, t) := (h(r, w, t) - y)(w \cdot x), \quad F(r, t) := -(h(r, w, t) - y).$$

We give quantitative characterizations for these random variables. We write $w^* = aw^{(i)} + bu$, where $a, b > 0, a^2 + b^2 = 1, u \in \mathbb{S}^{d-1}, u \perp w^{(i)}$. We furthermore assume that the parameters $h(r_i, w^{(i)}, t_i)$ are close to $(r^*, w^*, t^*)$. That is to say $\theta_i := \theta(w^*, w^{(i)}) \le O(1/(\bar{t}^*)^2)$, $|r_i - r^*| \le r^*/\text{polylog}(R^2/\epsilon)$, $|\bar{t}_i - \bar{t}^*| \le 1/\text{polylog}(R^2/\epsilon)$.

**Evaluation of $\mathbf{E}\,U_i^*$ and $\mathbf{E}\,F_i^*$** We first evaluate the expectation of $U^*$. To do this, we first expand $h(r_i, w^{(i)}, t_i) - h^*$ as follows

$$h(r_i, w^{(i)}, t_i) - h^* = h(r_i, w^{(i)}, t_i) - h(r^*, w^{(i)}, t^*) + h(r^*, w^{(i)}, t^*) - h^*.$$

Since

$$\mathbf{E}\,U_i^* = \mathop{\mathbf{E}}_{x \sim N(0,I)} (h(r_i, w^{(i)}, t_i) - h^*)(w^{(i)} \cdot x)\mathbb{1}\{w^{(i)} \cdot x > \bar{t}_i\}/\Phi(\bar{t}_i)$$

$$= \mathop{\mathbf{E}}_{x \sim N(0,I)} (h(r_i, w^{(i)}, t_i) - h(r^*, w^{(i)}, t^*) + h(r^*, w^{(i)}, t^*) - h^*)(w^{(i)} \cdot x)\mathbb{1}\{w^{(i)} \cdot x > \bar{t}_i\}/\Phi(\bar{t}_i),$$

we evaluate the two components separately. For the first term, we have

$$\mathop{\mathbf{E}}_{x \sim N(0,I)} (h(r_i, w^{(i)}, t_i) - h(r^*, w^{(i)}, t^*))(w^{(i)} \cdot x)\mathbb{1}\{w^{(i)} \cdot x > \bar{t}_i\}/\Phi(\bar{t}_i)$$

$$= \mathop{\mathbf{E}}_{z \sim N(0,I)} (\sigma(r_i z - t_i) - \sigma(r^* z - t^*))\, z\mathbb{1}\{z > \bar{t}_i\}/\Phi(\bar{t}_i)$$

We next consider the second term. By Fact A.5, we have

$$\mathop{\mathbf{E}}_{x \sim N(0,I)} (h(r^*, w^{(i)}, t^*) - h^*)(w^{(i)} \cdot x)\mathbb{1}\{w^{(i)} \cdot x > \bar{t}_i\}/\Phi(\bar{t}_i)$$

$$= (r^*/\Phi(\bar{t}^*)) \mathop{\mathbf{E}}_{z \sim N(0,1)} (\sigma(z - \bar{t}^*) - T_a \sigma(z - \bar{t}^*))\, z\mathbb{1}\{z > \bar{t}_i\}$$

$$= (r^*/\Phi(\bar{t}^*)) \mathop{\mathbf{E}}_{z \sim N(0,1)} \left( \int_a^1 \frac{dT_s \sigma(z - \bar{t}^*)}{ds} ds \right) \sigma'(z - \bar{t}_i) z = (r^*/\Phi(\bar{t}^*)) \int_a^1 \mathop{\mathbf{E}}_{z \sim N(0,1)} \frac{dT_s \sigma(z - \bar{t}^*)}{ds} \sigma'(z - \bar{t}_i) z\, ds$$

$$= (r^*/\Phi(\bar{t}^*)) \int_a^1 \mathop{\mathbf{E}}_{z \sim N(0,1)} \frac{1}{s} L T_s \sigma(z - \bar{t}^*)(z\sigma'(z - \bar{t}_i))\, ds = (r^*/\Phi(\bar{t}^*)) \int_a^1 \mathop{\mathbf{E}}_{z \sim N(0,1)} \frac{1}{s}(L T_s \sigma(z - \bar{t}^*))'(z\sigma'(z - \bar{t}_i))'\, ds$$

$$= (r^*/\Phi(\bar{t}^*)) \int_a^1 \mathop{\mathbf{E}}_{z \sim N(0,1)} T_s \sigma'(z - \bar{t}^*)\, (\sigma'(z - \bar{t}_i) + z\delta(z - \bar{t}_i))\, ds$$

$$= (r^*/\Phi(\bar{t}^*)) \int_0^\theta \sin s \mathop{\mathbf{E}}_{z \sim N(0,1)} T_{\cos s} \sigma'(z - \bar{t}^*)\, (\sigma'(z - \bar{t}_i) + z\delta(z - \bar{t}_i))\, ds.$$

We notice that

$$\mathop{\mathbf{E}}_{z \sim N(0,1)} T_{\cos s} \sigma'(z - \bar{t}^*)\sigma'(z - \bar{t}_i)$$

$$= \mathop{\mathbf{E}}_{z \sim N(0,1)} T_{\cos s} \sigma'(z - \bar{t}^*)\sigma'(z - \bar{t}^*) - \mathop{\mathbf{E}}_{z \sim N(0,1)} T_{\cos s} \sigma'(z - \bar{t}^*)(\sigma'(z - \bar{t}^*) - \sigma'(z - \bar{t}_i))$$

$$= \mathop{\mathbf{E}}_{z \sim N(0,1)} \left\| T_{\sqrt{\cos s}} \sigma'(z - \bar{t}^*) \right\|_2^2 - \mathop{\mathbf{E}}_{z \sim N(0,1)} T_{\cos s} \sigma'(z - \bar{t}^*)(\sigma'(z - \bar{t}^*) - \sigma'(z - \bar{t}_i)) = \Theta(\left\| T_{\sqrt{\cos s}} \sigma'(z - \bar{t}^*) \right\|_2^2),$$

when $\sin s < 1/\bar{t}^*$ and $|\bar{t}_i - \bar{t}^*| \le O(\log(R/\epsilon))$. This implies

$$(r^*/\Phi(\bar{t}^*)) \int_0^\theta \sin s \mathop{\mathbf{E}}_{z \sim N(0,1)} T_{\cos s} \sigma'(z - \bar{t}^*)\, (\sigma'(z - \bar{t}_i)))\, ds = \Theta((r^*/\Phi(\bar{t}^*))b^2 \left\| T_{\sqrt{\cos \theta}} \sigma'(z - \bar{t}^*) \right\|_2^2).$$

On the other hand,

$$\mathop{\mathbf{E}}_{z \sim N(0,1)} T_{\cos s} \sigma'(z - \bar{t}^*) z\delta(z - \bar{t}_i) = T_{\cos s} \sigma'(\bar{t}_i - \bar{t}^*) t\psi(t) = \mathop{\mathbf{Pr}}_{\beta \sim N(0,1)} (\cos s\bar{t}_i + \sin s\beta - \bar{t}^* > 0)\bar{t}_i \psi(\bar{t}_i).$$

This implies that

$$(r^*/\Phi(\bar{t}^*)) \int_0^\theta \sin s \mathop{\mathbf{E}}_{z \sim N(0,1)} T_{\cos s} \sigma'(z - \bar{t}^*) z\delta(z - \bar{t}_i) ds$$

$$= r^* \int_0^\theta \sin s \mathop{\mathbf{Pr}}_{\beta \sim N(0,1)} (\cos s\bar{t}_i + \sin s\beta - \bar{t}^* > 0)\bar{t}_i \psi(\bar{t}_i) ds$$

$$\le O(r^* b^2 \bar{t}_i \psi(\bar{t}_i)/\Phi(\bar{t}_i)) = O(r^* b).$$

Putting everything together, we get

$$\mathbf{E}\,U_i^* = \mathop{\mathbf{E}}_{z \sim N(0,I)} (\sigma(r_i z - t_i) - \sigma(r^* z - t^*))\, z\mathbb{1}\{z > \bar{t}_i\}/\Phi(\bar{t}_i) + cr^* b$$

for some suitable constant $c > 0$.

We next evaluate $\mathbf{E}\,F_i^*$ in a similar way.

$$\mathbf{E}\,F_i^* = \underset{x \sim N(0,I)}{\mathbf{E}}(h(r_i, w^{(i)}, t_i) - h^*)\mathbb{1}\{w^{(i)} \cdot x > \bar{t}_i\}/\Phi(\bar{t}_i)$$

$$= \underset{x \sim N(0,I)}{\mathbf{E}}(h(r_i, w^{(i)}, t_i) - h(r^*, w^{(i)}, t^*) + h(r^*, w^{(i)}, t^*) - h^*)\mathbb{1}\{w^{(i)} \cdot x > \bar{t}_i\}/\Phi(\bar{t}_i)$$

$$= \underset{z \sim N(0,I)}{\mathbf{E}}\left(\sigma(r_i z - t_i) - \sigma(r^* z - t^*)\right)/\Phi(\bar{t}_i) + \underset{x \sim N(0,I)}{\mathbf{E}}(h(r^*, w^{(i)}, t^*) - h^*)\mathbb{1}\{w^{(i)} \cdot x > \bar{t}_i\}/\Phi(\bar{t}_i)$$

For the second component in $\mathbf{E}\,F_i^*$, we have

$$\underset{x \sim N(0,I)}{\mathbf{E}}(h(r^*, w^{(i)}, t^*) - h^*)\mathbb{1}\{w^{(i)} \cdot x > \bar{t}_i\}/\Phi(\bar{t}_i)$$

$$= (r^*/\Phi(\bar{t}^*)) \underset{z \sim N(0,1)}{\mathbf{E}}\left(\sigma(z - \bar{t}^*) - T_a \sigma(z - \bar{t}^*)\right)\mathbb{1}\{z > \bar{t}_i\}$$

$$= (r^*/\Phi(\bar{t}^*)) \underset{z \sim N(0,1)}{\mathbf{E}}\left(\int_a^1 \frac{dT_s \sigma(z - \bar{t}^*)}{ds}ds\right)\sigma'(z - \bar{t}_i) = (r^*/\Phi(\bar{t}^*))\int_a^1 \underset{z \sim N(0,1)}{\mathbf{E}}\frac{dT_s \sigma(z - \bar{t}^*)}{ds}\sigma'(z - \bar{t}_i)ds$$

$$= (r^*/\Phi(\bar{t}^*))\int_a^1 \underset{z \sim N(0,1)}{\mathbf{E}}\frac{1}{s}LT_s\sigma(z - \bar{t}^*)(\sigma'(z - \bar{t}_i))ds = (r^*/\Phi(\bar{t}^*))\int_a^1 \underset{z \sim N(0,1)}{\mathbf{E}}\frac{1}{s}(LT_s\sigma(z - \bar{t}^*))'(\sigma'(z - \bar{t}_i))'ds$$

$$= (r^*/\Phi(\bar{t}^*))\int_a^1 \underset{z \sim N(0,1)}{\mathbf{E}}T_s\sigma'(z - \bar{t}^*)\left(\delta(z - \bar{t}_i)\right)ds$$

$$= (r^*/\Phi(\bar{t}^*))\int_0^\theta \sin s \underset{z \sim N(0,1)}{\mathbf{E}}T_{\cos s}\sigma'(z - \bar{t}^*)\left(\delta(z - \bar{t}_i)\right)ds \leq O((r^*/\Phi(\bar{t}^*))\,b^2\psi(\bar{t}_i)) \leq O(r^* b)$$

This gives $\mathbf{E}\,F_i^* = \mathbf{E}_{z \sim N(0,I)}\left(\sigma(r_i z - t_i) - \sigma(r^* z - t^*)\right)\mathbb{1}\{w^{(i)} \cdot x > \bar{t}_i\}/\Phi(\bar{t}_i) + cbr^*$ for some suitably small constant $c$.

As a summary, we have the following:

**Proposition C.7.** *There exist small constants $c_1, c_2 > 0$ such that,*

$$\mathbf{E}\,U_i^* = \underset{z \sim N(0,I)}{\mathbf{E}}\left(\sigma(r_i z - t_i) - \sigma(r^* z - t^*)\right)z\mathbb{1}\{z > \bar{t}_i\}/\Phi(\bar{t}_i) + c_1 r^* b$$

$$\mathbf{E}\,F_i^* = \underset{z \sim N(0,I)}{\mathbf{E}}\left(\sigma(r_i z - t_i) - \sigma(r^* z - t^*)\right)\mathbb{1}\{w^{(i)} \cdot x > \bar{t}_i\}/\Phi(\bar{t}_i) + cbr^*.$$

**Evaluation for Noise Terms**   We next evaluate the noise terms. For the noise term in $U_i$, we have

$$|\mathbf{E}(U_i - U_i^*)| = \left|\underset{x \sim N(0,I)}{\mathbf{E}}(y - h^*)(w^{(i)} \cdot x)\mathbb{1}(w^{(i)} \cdot x > \bar{t}_i)/\Phi(\bar{t}_i)\right|$$

$$\leq \sqrt{\underset{x \sim N(0,I)}{\mathbf{E}}(y(x) - h^*(x))^2}\sqrt{\underset{x \sim N(0,I)}{\mathbf{E}}\mathbb{1}\{w^{(i)} \cdot x > \bar{t}_i\}(x \cdot w^{(i)})^2}/\Phi(\bar{t}_i)$$

$$\leq \left|\sqrt{\epsilon}\sqrt{\underset{z \sim N(0,1)}{\mathbf{E}}z^2\mathbb{1}(z > \bar{t}_i)}\right|/\Phi(\bar{t}_i) = \sqrt{\epsilon}\sqrt{\bar{t}_i\psi(\bar{t}_i) + \Phi(\bar{t}_i)}/\Phi(\bar{t}_i).$$

Here, in the last equation, we use the fact that $\mathbf{E}_{z \sim N(0,1)}z^2\mathbb{1}(z > \bar{t}_i) = \bar{t}_i\psi(\bar{t}_i) + \Phi(\bar{t}_i)$.

Similarly, for the noise term in $F$, we have

$$|\mathbf{E}(F_i - F_i^*)| = \left|\underset{x \sim N(0,I)}{\mathbf{E}}(y - h^*)\mathbb{1}(w^{(i)} \cdot x > \bar{t}_i)/\Phi(\bar{t}_i)\right|$$

$$\leq \sqrt{\underset{x \sim N(0,I)}{\mathbf{E}}(y(x) - h^*(x))^2}\sqrt{\underset{x \sim N(0,I)}{\mathbf{E}}\mathbb{1}\{w^{(i)} \cdot x > \bar{t}_i\}}/\Phi(\bar{t}_i)$$

$$\leq \left|\sqrt{\epsilon}\sqrt{\underset{z \sim N(0,1)}{\mathbf{E}}\mathbb{1}(z > \bar{t}_i)}\right|/\Phi(\bar{t}_i) = \sqrt{\epsilon}\sqrt{\Phi(\bar{t}_i)}/\Phi(\bar{t}_i).$$

As a summary, we have established the following:

**Proposition C.8.**

$$|\mathbf{E}(U_i - U_i^*)| \leq \sqrt{\epsilon}\sqrt{\bar{t}_i \psi(\bar{t}_i) + \Phi(\bar{t}_i)}/\Phi(\bar{t}_i)$$
$$|\mathbf{E}(F_i - F_i^*)| \leq \sqrt{\epsilon}\sqrt{\Phi(\bar{t}_i)}/\Phi(\bar{t}_i).$$

We remark that when $\bar{t}_i$ is close to $\bar{t}^*$, the noise rate of $U_i$ is larger than that of $F_i$ by a factor of $\bar{t}^*$. This is one of the central reasons why we need a small step size to update $(r, t)$, and can only guarantee that $(r_T, t_T)$ is $O(\epsilon \text{polylog}(R^2/\epsilon)/\Phi(\bar{t}^*))$ close to $(r^*, t^*)$ with gradient descent only.

**Evaluation for Variance** Finally, we examine the variance of $U_i$ and $F_i$. We start with $U_i$. Notice that $\mathbf{E}\, U_i^2 \leq 2\,\mathbf{E}(U_i^*)^2 + 2\,\mathbf{E}(U_i^* - U_i)^2$.

Let $M > 0$ be a threshold such that $\mathbf{E}_{z \sim N(0,1)}\, z^2 \mathbb{1}\{|z| > M\} \leq \epsilon/R$. Notice that if we set $y' = \text{sign}(y)\min\{|y|, r_i M\}$, then we will only introduce at most $\epsilon$ error, since $|r_i - r^*| \leq r^*/\log(R/\epsilon)$. So, we can, without loss of generality, assume that $|y| \leq M$. In particular, for the ReLU activation, $M \leq O(\sqrt{\log(R/\epsilon)})$. Based on this, we obtain that:

$$\mathbf{E}(U_i^* - U_i)^2 = \underset{x \sim N(0,I)}{\mathbf{E}}\, (y(x) - h^*(x))^2\, (x \cdot w^{(i)})^2 \mathbb{1}\{w^{(i)} \cdot x > \bar{t}_i\}\mathbb{1}\{(x \cdot w^{(i)}) \leq M\}/\Phi(\bar{t}_i)$$

$$+ \underset{x \sim N(0,I)}{\mathbf{E}}\, (y(x) - h^*(x))^2\, (x \cdot w^{(i)})^2 \mathbb{1}\{w^{(i)} \cdot x > \bar{t}_i\}\mathbb{1}\{(x \cdot w^{(i)}) > M\}/\Phi(\bar{t}_i)$$

$$\leq \epsilon M^2/\Phi(t) + 4M^2 \underset{x \sim N(0,I)}{\mathbf{E}}\, (x \cdot w)^2 \mathbb{1}\{w \cdot x > t\}\mathbb{1}\{(x \cdot w) > M\}/\Phi(t_i)$$

$$\leq \epsilon M^2/\Phi(\bar{t}_i) + 4\epsilon M^2/\Phi(\bar{t}_i) = \tilde{O}(\epsilon/\Phi(\bar{t}_i)).$$

We next consider the variance of $\mathbf{E}(U_i^*)^2$. We have

$$\mathbf{E}(U_i^*)^2 \leq 2 \underset{x \sim N(0,I)}{\mathbf{E}}\, (h(r_i, w^{(i)}, t_i) - h(r^*, w^{(i)}, t^*))^2 (w^{(i)} \cdot x)^2 \mathbb{1}\{w^{(i)} \cdot x > \bar{t}_i\}/\Phi(\bar{t}_i)$$

$$+ 2 \underset{x \sim N(0,I)}{\mathbf{E}}\, (h(r^*, w^{(i)}, t^*) - h^*)^2 (w^{(i)} \cdot x)^2 \mathbb{1}\{w^{(i)} \cdot x > \bar{t}_i\}/\Phi(\bar{t}_i)$$

We separately evaluate the two terms above. First, we have

$$\underset{x \sim N(0,I)}{\mathbf{E}}\, (h(r_i, w^{(i)}, t_i) - h(r^*, w^{(i)}, t^*))^2 (w^{(i)} \cdot x)^2 \mathbb{1}\{w^{(i)} \cdot x > \bar{t}_i\}/\Phi(\bar{t}_i)$$

$$= \underset{z \sim N(0,1)}{\mathbf{E}}\, (r_i \sigma(z - \bar{t}_i) - r^* \sigma(z - t^*))^2 z^2 \mathbb{1}\{z > \bar{t}_i\}/\Phi(\bar{t}_i)$$

For the second term, recall that we decompose $w^* = aw^{(i)} + bu$. Notice that

$$\left(\sigma(w^{(i)} \cdot x - \bar{t}^*) - \sigma(w^* \cdot x - \bar{t}^*)\right)^2 \leq \left((w^{(i)} - w^*) \cdot x\right)^2 = \left((1-a)(w^{(i)} \cdot x) + b(u \cdot x)\right)^2$$

$$\leq 2(1-a)^2 (w^{(i)} \cdot x)^2 + 2b^2 (u \cdot x)^2$$

This implies

$$\underset{x \sim N(0,I)}{\mathbf{E}}\, (h(r^*, w^{(i)}, t^*) - h^*)^2 (w^{(i)} \cdot x)^2 \mathbb{1}\{w^{(i)} \cdot x > \bar{t}_i\}/\Phi(\bar{t}_i)$$

$$= \underset{x \sim N(0,I)}{\mathbf{E}}\, \left(2(1-a)^2 (w^{(i)} \cdot x)^2 + 2b^2 (u \cdot x)^2\right)(w^{(i)} \cdot x)^2 \mathbb{1}\{w^{(i)} \cdot x > \bar{t}_i\}(r^*)^2/\Phi(\bar{t}_i)$$

$$= \underset{z,s \sim N(0,1)}{\mathbf{E}}\, \left(2(1-a)^2 (z)^2 + 2b^2 (s)^2\right)(z)^2 \mathbb{1}\{z > \bar{t}_i\}(r^*)^2/\Phi(\bar{t}_i)$$

$$\leq O\left(\underset{z \sim N(0,1)}{\mathbf{E}}\, b^4 z^4 \mathbb{1}\{z > \bar{t}_i\}(r^*)^2/\Phi(\bar{t}_i) + \underset{z \sim N(0,1)}{\mathbf{E}}\, b^2 z^2 \mathbb{1}\{z > \bar{t}_i\}(r^*)^2/\Phi(\bar{t}_i)\right)$$

We evaluate $F$ in a similar way. First, we have

$$\mathbf{E}(F_i^* - F_i)^2 \leq \epsilon/\Phi(\bar{t}_i).$$

Next, we have

$$\mathbf{E}(F_i^*)^2 \leq 2 \underset{x \sim N(0,I)}{\mathbf{E}} (h(r_i, w^{(i)}, t_i) - h(r^*, w^{(i)}, t^*))^2 \mathbb{1}\{w^{(i)} \cdot x > \bar{t}_i\}/\Phi(\bar{t}_i)$$

$$+ 2 \underset{x \sim N(0,I)}{\mathbf{E}} (h(r^*, w^{(i)}, t^*) - h^*)^2 \mathbb{1}\{w^{(i)} \cdot x > \bar{t}_i\}/\Phi(\bar{t}_i)$$

The first term can be evaluated as

$$\underset{x \sim N(0,I)}{\mathbf{E}} (h(r_i, w^{(i)}, t_i) - h(r^*, w^{(i)}, t^*))^2 \mathbb{1}\{w^{(i)} \cdot x > \bar{t}_i\}/\Phi(\bar{t}_i)$$

$$= \underset{z \sim N(0,1)}{\mathbf{E}} (r_i \sigma(z - \bar{t}_i) - r^* \sigma(z - t^*))^2 \mathbb{1}\{z > \bar{t}_i\}/\Phi(\bar{t}_i)$$

For the second term, we have

$$\underset{x \sim N(0,I)}{\mathbf{E}} (h(r^*, w^{(i)}, t^*) - h^*)^2 \mathbb{1}\{w^{(i)} \cdot x > \bar{t}_i\}/\Phi(\bar{t}_i)$$

$$= \underset{x \sim N(0,1)}{\mathbf{E}} \left(2(1-a)^2(w^{(i)} \cdot x)^2 + 2b^2(u \cdot x)^2\right) \mathbb{1}\{w^{(i)} \cdot x > \bar{t}_i\}(r^*)^2/\Phi(\bar{t}_i)$$

$$= \underset{z,s \sim N(0,1)}{\mathbf{E}} \left(2(1-a)^2(z)^2 + 2b^2(s)^2\right)(z)^2 \mathbb{1}\{z > \bar{t}_i\}(r^*)^2/\Phi(\bar{t}_i)$$

$$\leq O(\underset{z \sim N(0,1)}{\mathbf{E}} b^4 z^2 \mathbb{1}\{z > \bar{t}_i\}(r^*)^2/\Phi(\bar{t}_i) + \underset{z \sim N(0,1)}{\mathbf{E}} b^2 \mathbb{1}\{z > \bar{t}_i\}(r^*)^2/\Phi(\bar{t}_i)) .$$

As a summary, we have established:

**Proposition C.9.**

$$\mathbf{E} U_i^2 \leq \tilde{O}\left(\underset{z \sim N(0,1)}{\mathbf{E}} (r_i \sigma(z - \bar{t}_i) - r^* \sigma(z - t^*))^2 z^2 \mathbb{1}\{z > \bar{t}_i\}/\Phi(\bar{t}_i) + b^2(r^*)^2 + \epsilon/\Phi(\bar{t}_i)\right)$$

$$\mathbf{E} F_i^2 \leq \tilde{O}\left(\underset{z \sim N(0,1)}{\mathbf{E}} (r_i \sigma(z - \bar{t}_i) - r^* \sigma(z - t^*))^2 \mathbb{1}\{z > \bar{t}_i\}/\Phi(\bar{t}_i) + b^2(r^*)^2 + \epsilon/\Phi(\bar{t}_i)\right) .$$

### C.5.1 Progress on $(r,t)$ Updates

In this section, we describe how to update the parameters $(r,t)$ so that we are able to make the term $\mathbf{E}_{z \sim N(0,I)} \left(\sigma(r_i z - t_i) - \sigma(r^* z - t^*)\right)^2$ stably drop every time we choose to update the parameters. For convenience, in this section, we define the following 2-dimensional vector:

$$g^{(i)} = \left(\underset{z \sim N(0,I)}{\mathbf{E}} ((r_i z - t_i) - (r^* z - t^*)) z \mathbb{1}\{z > \bar{t}^*\}, - \underset{z \sim N(0,I)}{\mathbf{E}} ((r_i z - t_i) - (r^* z - t^*)) \mathbb{1}\{z > \bar{t}^*\}\right)^\top$$

$$= ((r_i - r^*)\delta(\bar{t}^*) - (t_i - t^*)\psi(\bar{t}^*), -(r_i - r^*)\psi(\bar{t}^*) + (t_i - t^*)\Phi(\bar{t}^*))^\top$$

and function

$$Z(r,t) := \underset{z \sim N(0,1)}{\mathbf{E}} ((rz - t) - (r^* z - t^*))^2 \mathbb{1}(z > \bar{t}^*)$$

$$= (r - r^*)^2 \delta(\bar{t}^*) + (t - t^*)\Phi(\bar{t}^*) - 2(r - r^*)(t - t^*)\psi(\bar{t}^*)$$

$$W(r,t) := \underset{z \sim N(0,1)}{\mathbf{E}} (\sigma(rz - t) - \sigma(r^* z - t^*))^2 .$$

To simplify the notation, we define

$$Q := \begin{pmatrix} \delta(\bar{t}^*) & -\psi(\bar{t}^*) \\ -\psi(\bar{t}^*) & \Phi(\bar{t}^*), \end{pmatrix}$$

and define $\Delta(r,t) = ((r - r^*), (t - t^*))^\top$. By definition, $Z(r,t) = \Delta^\top Q \Delta$. In particular, the largest eigenvalue of $Q$ is $O((\bar{t}^*)^2 \Phi(\bar{t}^*))$, while the smallest eigenvalue of $Q$ is $\Omega(\Phi(\bar{t}^*))/(\bar{t}^*)^4)$. Thus, the ratio of the largest and smallest eigenvalues of $Q$ is at most $\log^3(R^2/\epsilon)$. Furthermore, we want to mention that, when $\bar{t}^*$ becomes large, $Q$ becomes ill-conditioned; this is the reason why in [Algorithm 4](#), we update $(r_i, t_i)$ with a much smaller rate of $1/\text{polylog}(R^2/\epsilon)$ and the gradient descent stage can only guarantee that $(r_T, t_T)$ is $\epsilon \text{polylog}(R^2/\epsilon)/\Phi(\bar{t}^*)$ close to the optimal parameters $(r^*, t^*)$.

We remark that $g^{(i)}$ is exactly the gradient of $Z$ at point $(r_i, t_i)$. The motivation of using this notation is that when $(r,t)$ is close to $(r^*, t^*)$, $Z(r,t)$ and $W(r,t)$, the quantities we want to control, are different by a negligible factor. We give the following proposition.

**Proposition C.10.** *Suppose $\bar{t}^* \le \sqrt{\log(R^2/\epsilon)}$ and $r^* \le R$, given a pair of parameter $(r,t)$ such that $|r - r^*| \le r^*/\mathrm{polylog}(R^2/\epsilon)$ and $|\bar{t} - \bar{t}^*| \le 1/\mathrm{polylog}(R^2/\epsilon)$, we have*

$$|W(r,t) - Z(r,t)| = o(Z(r,t)).$$

*Proof.* Since $W(r,t)$ and $Z(r,t)$ are only different at region $\{z \mid |\mathbb{1}(z > \bar{t}_i - \mathbb{1}(z > \bar{t}^*)| > 0\}$, when $(r,t)$ are close to $(r^*, t^*)$, we have

$$|W(r,t) - Z(r,t)| \le \left((r-r^*)^2 + (t-t^*)^2\right) \mathbf{Pr}_z(z \in \{z \mid |\mathbb{1}(z > \bar{t}_i - \mathbb{1}(z > \bar{t}^*)| > 0\})$$

$$\le \left((r-r^*)^2 + (t-t^*)^2\right) \Phi(\bar{t}^*)/\mathrm{polylog}(R^2/\epsilon) \le Z(r,t)/\mathrm{polylog}(R^2/\epsilon).$$

Here, in the last inequality, we use the fact that the smallest eigenvalue of $Q$ is $\Omega(\Phi(\bar{t}^*)/(\bar{t}^{*4}))$ and thus $Z(r,t) \ge \Omega(\Delta(r,t)^2/\mathrm{polylog}(R^2/\epsilon))$ □

Such a proposition can also be generalized to the gradient of $F$ and $W$. We next give the following proposition.

**Proposition C.11.** *Suppose $\bar{t}^* \le \sqrt{\log(R^2/\epsilon)}$ and $r^* \le R$, given a pair of parameter $(r,t)$ such that $|r - r^*| \le r^*/\mathrm{polylog}(R^2/\epsilon)$ and $|\bar{t} - \bar{t}^*| \le 1/\mathrm{polylog}(R^2/\epsilon)$, we have*

$$\|\nabla Z(r,t) - \nabla W(r,t)\| \le o(\|\Delta(r,t)\| \Phi(\bar{t})).$$

*Proof of Proposition C.11.* We notice that

$$\nabla_r W(r,t) = 2 \mathop{\mathbf{E}}_{z \sim N(0,1)} (\sigma(rz - t) - \sigma(r^*z - t^*)) z\mathbb{1}(z > \bar{t})$$

$$\nabla_t W(r,t) = -2 \mathop{\mathbf{E}}_{z \sim N(0,1)} (\sigma(rz - t) - \sigma(r^*z - t^*)) \mathbb{1}(z > \bar{t})$$

Since $W(r,t)$ and $Z(r,t)$ are only different at region $\{z \mid |\mathbb{1}(z > \bar{t}_i - \mathbb{1}(z > \bar{t}^*)| > 0\}$, when $(r,t)$ are close to $(r^*, t^*)$, when

$$\|\nabla Z(r,t) - \nabla W(r,t)\|^2 \le ((r-r^*)^2 + (t-t^*)^2)\bar{t}^2 \mathbf{Pr}_z(z \in \{z \mid |\mathbb{1}(z > \bar{t}_i - \mathbb{1}(z > \bar{t}^*)| > 0\})^2$$

$$\le \Delta^2 \Phi(\bar{t})^2/\mathrm{polylog}(R^2/\epsilon) .$$

□

Before formally presenting the analysis for updating the parameters $(r,t)$, we present the following standard gradient descent analysis for analyzing quadratic minimization.

**Proposition C.12.** *Let $Q \in \mathbb{R}^{2\times 2}$ be a positive definite matrix and let $L, \mu$ be the largest eigenvalue and smallest eigenvalue of $Q$. For $x \in \mathbb{R}^2$, define $F(x) = x^\top Q x$. For $0 < c < (\mu/L)/C$, for a large constant $C$, given $g' \in \mathbb{R}^2$ such that $\|g' - \nabla F\| \le c\|\nabla F\|$, let $0 < \eta \le O(1/L)$, then the update $x' = x - \mu g'$ satisfies $F(x') \le (1 - O(\mu\eta))F(x)$ and $\|x'\| \le (1 - O(\eta\mu)) \|x\|$.*

*Proof of Proposition C.12.* The proof of Proposition C.12 follows the standard gradient descent analysis for $L$ smooth functions that satisfy the Polyak-Lojasiewicz condition with parameter $\mu$. We have

$$\|\nabla F\|^2 \ge 2\mu F(x)$$

$$F(x') \le F(x) - \eta\nabla F(x) \cdot g' + L\eta^2 \|g'\|^2 /2 .$$

Since

$$\nabla F(x) \cdot g' \ge \|\nabla F(x)\|^2 - \|\nabla F(x)\| \cdot \|g' - \nabla F(x)\| \ge (1-c)\|\nabla F(x)\|^2,$$

and

$$\|g'\| \le (1+c)\|\nabla F(x)\|,$$

we have

$$F(x') \leq F(x) - \eta(1-c)\|\nabla F(x)\|^2 + \frac{L\eta^2}{2}(1+c)^2\|\nabla F(x)\|^2$$

$$\leq \left[1 - 2\mu\left(\eta(1-c) - \frac{L\eta^2}{2}(1+c)^2\right)\right]F(x).$$

This implies that by choosing $\eta = O(1/L)$, we have $F(x') \leq (1 - O(\mu\eta))F(x)$.

We next show that the update also contracts the parameter distance.

$$\|x'\| = \|x - \eta g'\| = \|x - \eta\nabla F(x) + \eta(\nabla F(x) - g')\| \leq \|x - \eta\nabla F(x)\| + \eta\|\nabla F(x) - g'\|$$
$$\leq \|x - \eta F(x)\| + \eta c\|F(x)\| \leq (1 - \eta\mu)\|x\| + c\eta L\|x\| \leq (1 - O(\eta\mu))\|x\|.$$

Here, the last inequality follows the fact that $c \leq (\mu/L)/C$. $\qquad\square$

In our setting, we have that the parameters $\mu, L$ for $Q$ are $\Omega(\Phi(\bar{t}^*)/(\bar{t}^*)^4)$ and $O((\bar{t}^*)^2\Phi(\bar{t}^*))$. Thus, after zooming in to the region $\mathbb{1}(z > \bar{t})$, the rescaled parameters become $\mu = \Omega(1/(\bar{t}^*)^4)$ and $L = O((\bar{t}^*)^2)$. By choosing the step size $\mu = 1/\mathrm{polylog}(R^2/\epsilon)$, we are able to drop $Z(r,t)$, and thus $Z(r,t)$, by a factor of $1 - 1/\mathrm{polylog}(R^2/\epsilon)$ in each round of update. Furthermore, after the update $(r,t)$, is still close to $(r^*, t^*)$. So far, we have already stated all ingredients for analyzing the update for parameters $(r,t)$. We present the following lemma to conclude this section.

**Lemma C.13.** *Consider the problem of agnostic ReLU regression with queries. Let $h(r_i, w^{(i)}, t_i)$ be a ReLU activation function. Write $w^* = aw^{(i)} + bu$, where $a, b > 0, a^2 + b^2 = 1, u \in \mathbb{S}^{d-1}, u \perp w^{(i)}$. Suppose that $|\bar{t}_i - \bar{t}^*| \leq 1/\mathrm{polylog}(R^2/\epsilon)$, $b \leq 1/\bar{t}_i$ and $|r_i - r^*| \leq r^*/\mathrm{polylog}(R^2/\epsilon)$. Let $B_i > 0$ and define $B_{i+1} = (1 - 1/C)B_i$ for some universal constant $C$. Suppose $B_i, B_{i+1}$ satisfy the following properties:*

1. $(r^*)^2 \sin^2\theta_i \leq B_{i+1}^2$,

2. $B_i^2\Phi(\bar{t}^*) \geq \Omega(\epsilon)$,

3. $W(r_i, t_i) \leq B_i^2\Phi(\bar{t}^*)\mathrm{polylog}(R^2/\epsilon)$.

*If $W(r_i, t_i) \geq B_{i+1}^2\Phi(\bar{t}^*)\mathrm{polylog}(R^2/\epsilon)$, then there is an algorithm that makes $\mathrm{polylog}(R^2/\epsilon)$ queries, runs in $\mathrm{polylog}(R^2/\epsilon)$ time, and outputs a pair of $(r_{i+1}, t_{i+1})$ such that $W(r_{i+1}, t_{i+1}) \leq B_{i+1}^2\Phi(\bar{t}^*)\mathrm{polylog}(R^2/\epsilon)$.*

*Proof of Lemma C.13.* For a parameter $(r,t)$, if $W(r,t) \geq B_{i+1}^2\Phi(\bar{t}^*)\mathrm{polylog}(R^2/\epsilon)$, by Proposition C.10, it must be the case that $Z(r,t) \geq B_{i+1}^2\Phi(\bar{t}^*)\mathrm{polylog}(R^2/\epsilon)$, which implies that $\|\nabla Z(r,t)\|_2^2 \geq B_{i+1}^2\Phi(\bar{t}^*)\mathrm{polylog}(R^2/\epsilon)$. By Proposition C.11, this implies that $\|\nabla W(r,t)\|^2 \geq B_{i+1}^2\Phi(\bar{t}^*)\mathrm{polylog}(R^2/\epsilon)$. On the other hand, by Proposition C.7, we know that when $(r^*)^2\sin^2\theta_i \leq B_{i+1}^2$, $\|(\mathbf{E}\,U(r,t), \mathbf{E}\,F(r,t)) - \nabla W(r,t)\| \leq o(\|\nabla W(r,t)\|)$. This implies that $\|(\mathbf{E}\,U(r,t), \mathbf{E}\,F(r,t)) - \nabla Z(r,t)/\Phi(\bar{t}^*)\| \leq o(\|\nabla Z(r,t)/\Phi(\bar{t}^*)\|)$. By Proposition C.9, with $\mathrm{polylog}(R^2/\epsilon)$ queries, we are able to estimate $(\mathbf{E}\,U(r,t), \mathbf{E}\,F(r,t))$ with error $\|(\mathbf{E}\,U(r,t), \mathbf{E}\,F(r,t))\|/\mathrm{polylog}(R^2/\epsilon)$; and thus this is different from $\nabla Z(r,t)/\Phi(\bar{t}^*)$ by $\|\nabla Z(r,t)/\Phi(\bar{t}^*)\|/\mathrm{polylog}(R^2/\epsilon)$. By Proposition C.12, we know that by running gradient descent with step size $O(1/\mathrm{polylog}(R^2/\epsilon))$, we are able to drop $Z(r,t)$ be a factor of $(1 - 1/\mathrm{polylog}(R^2/\epsilon))$ in each round; and thus after $\mathrm{polylog}(R^2/\epsilon)$ rounds of iterations, we have $Z(r_{i+1}, t_{i+1}) \leq B_{i+1}^2\Phi(\bar{t}^*)\mathrm{polylog}(R^2/\epsilon)$. Therefore, by Proposition C.10, we have $W(r_{i+1}, t_{i+1}) \leq B_{i+1}^2\Phi(\bar{t}^*)\mathrm{polylog}(R^2/\epsilon)$. $\qquad\square$

## C.6   Proof of Theorem 3.1

In this section, we give the proof of Theorem 3.1.

*Proof of Theorem 3.1.* We first notice that $p < \epsilon/R^2$, otherwise the 0 function has error $O(\epsilon)$. Thus, by Lemma 3.2, with $\min\{1/p, R^2/\epsilon\}$ queries, we obtain a warm start $w^{(0)}$ such that $\theta_0 \leq O(1/\bar{t}^*)$. Furthermore, by the construction of the grid $(r,t)$, if we randomly chose a parameter $(r_0, t_0)$ from

the grid, with a non-trivial probability$(1/\text{polylog}(R^2/\epsilon))$, we have $|r_0 - r^*| \leq r^*/\text{polylog}(R^2/\epsilon)$ and $|\bar{t}_0 - \bar{t}^*| \leq 1/\text{polylog}(R^2/\epsilon)$.

With the pair of $(r_0, w^{(0)}, t_0)$, we run Algorithm 4. We prove by induction that in each round of the update $W(r_i, t_i) \leq B_i^2 \Phi(\bar{t}^*)\text{polylog}(R^2/\epsilon)$ and $(r^*)^2 \sin^2 \theta_i \leq B_{i+1}^2$. Notice that for $i = 0$ this holds automatically by the property of $(r_0, w^{(0)}), t_0$ as a warm start.

Now suppose that this holds in the $i$th round. We will show this holds for the $i + 1$th round. By induction, we have $(r^*)^2 \sin^2 \theta_{i+1} \leq B_{i+1}^2$. By Proposition C.7 and Proposition C.11, we must have $\|(\mathbf{E}\, U(r_i, t_i), \mathbf{E}\, F(r_i, t_i)) - \nabla F(r_i, t_i)/\Phi(\bar{t}^*)\| \leq o(\|\nabla Z(r, t)/\Phi(\bar{t}^*)\|)$. By Proposition C.9, with $\text{polylog}(R^2/\epsilon)$ samples, we are able to estimate $(\mathbf{E}\, U(r_i, t_i), \mathbf{E}\, F(r_i, t_i))$ up to error $\|(\mathbf{E}\, U(r_i, t_i), \mathbf{E}\, F(r_i, t_i))\|/\text{polylog}(R^2/\epsilon)$. Since $Z(r, t)/\Phi(\bar{t}^*) \geq \|\nabla Z/\Phi(\bar{t}^*)\|^2/(\bar{t}^*)^4$, this implies if $W(r_i, t_i) > B_{i+1}^2 \Phi(\bar{t}^*)\text{polylog}(R^2/\epsilon)$, we are able to verify this by looking at the length of the gradient; and thus by Lemma C.13, after making $\text{polylog}(R^2/\epsilon)$ queries, we get $W(r_{i+1}, t_{i+1}) \leq B_{i+2}^2 \Phi(\bar{t}^*)\text{polylog}(R^2/\epsilon)$. After this by Lemma C.6, by making $\tilde{O}(d)$ queries, we have $(r^*)^2 \theta_{i+1}^2 \leq B_{i+2}^2$.

Now after $T = O(\log(R^2/\epsilon))$ rounds, we have $(r^*)^2 \theta_T^2 \Phi(\bar{t}^*) \leq O(\epsilon)$, while $W(r_T, t_T) \leq O(\epsilon \text{polylog}(R^2/\epsilon))$. By Proposition C.10, this implies $\|\Delta(r_T, t_T)\|^2 \Phi(\bar{t}^*) \leq Z(r_T, t_T)\text{polylog}(R^2/\epsilon) \leq O(\epsilon \text{polylog}(R^2/\epsilon))$. Notice that if a pair of $(r, t)$ satisfies $\|\Delta(r, t)\|^2 \leq \epsilon/(\Phi(\bar{t}_i)\text{polylog}(R^2/\epsilon))$, then $W(r, t) \leq O(\epsilon)$. This implies that if we consider the ball $B \subseteq \mathbb{R}^2$ centered at $(r_T, t_T)$ with radius $O(\sqrt{\epsilon \text{polylog}(R^2/\epsilon)/\Phi(\bar{t}^*)})$, then $(r^*, t^*) \in B$. In particular, if we grid $B$ with a net $N$ of size $\text{polylog}(R^2/\epsilon)$, there must be some $(r', t')$ that is $\sqrt{\epsilon/(\Phi(\bar{t}_i)\text{polylog}(R^2/\epsilon))}$ close to $(r', t')$. By uniformly sampling from the grid, with probability at least $1/\text{polylog}(R^2/\epsilon)$, we get such $(r', t')$ with $W(r', t') \leq O(\epsilon)$. By Lemma 3.3, we know that $h(r', w^{(T)}, t')$ has noiseless error at most $O(\epsilon)$, and thus $\text{err}(h(r', w^{(T)}, t')) = O(\epsilon)$. Furthermore, the total number of queries we use is $\tilde{O}_\delta(\min\{1/p, R^2/\epsilon\} + d \cdot \text{polylog}(R^2/\epsilon))$. We remark that the current algorithm has a probability of success of $1/\text{polylog}(R^2/\epsilon)$. Thus, by running it $\text{polylog}(R^2/\epsilon)$ times, we get a list of $\text{polylog}(R^2/\epsilon)$ hypothesis, one of which has error $O(\epsilon)$. By doing a hypothesis selection procedure using Lemma 2.11, we are able to select a hypothesis with $O(\epsilon)$ error with high probability, which will only cost another $\text{polylog}(R^2/\epsilon)$ queries. $\qquad\square$

# D  Omitted Proofs from Section 4

## D.1  Proof of Theorem 4.1

For convenience, we restate the theorem below.

**Theorem D.1.** *Consider the problem of agnostic ReLU regression with queries with a restriction that the optimal ReLU satisfies $\|W^*\| \leq 1$ and has bias at least $p$. Any learning algorithm that outputs a hypothesis with error less than $O(p/\log^2(p))$ with probability $1/3$, must make at least $\tilde{\Omega}(1/p^{1-o_d(1)} + d)$ queries. Furthermore, this holds even if $\text{opt} \leq 2^{-\Omega(d^{1/4})}p$.*

*Proof.* We break down the proof into two parts. First, we show a lower bound of $\tilde{\Omega}(1/p)$. Consider two hypotheses $h_1(x) = \sigma(w^* \cdot x - t^*)$, where $w^*$ is drawn uniformly from $\mathbb{S}^{d-1}$ and $h_2(x) = 0$. Notice that

$$\mathbf{E}_{x \sim N(0,I)} (h_1(x) - h_2(x))^2 = V(t^*) = (t^*)^2 \Phi(t^*) - t^*\psi(t^*) \geq \Phi(t^*)/(t^*)^2 = \tilde{\Omega}(p).$$

Thus, any learning algorithm that can learn a hypothesis with error $\tilde{O}(p)$ can distinguish whether the target hypothesis is $h_1$ or $h_2$. We construct adversarial label noise as follows, when $h_1$ is the underlying hypothesis. For every example such that $\|x\|^2 > d + \Delta$, where $\Delta = d^\alpha$, $0 < \alpha < 1$, and $d$ large enough, the adversary changes its label by $y(x) = 0$. We first show that the noise level is small. We have

$$\mathbf{E}_{x \sim N(0,I)} (h_1(x) - y)^2 \leq \mathbf{E}_{x \sim N(0,I)} h_1^2(x)\mathbb{1}\{w^* \cdot x > t^*, \|x\|^2 > d + \Delta\}$$

$$\leq \tilde{O}(p \exp(-\Omega(d^{2\alpha-1}))),$$

where in the last inequality, we use the tail bound for $\chi^2$-distribution. Here, we choose $\alpha = 5/8$ to make $2\alpha - 1 = 1/4$.

This implies that by querying examples with norm larger than $\sqrt{d + \Delta}$, a learner will get no information. Now we consider a deterministic learner that makes $r$ queries over the ball with radius $\sqrt{d + \Delta}$. Since $w^*$ is drawn uniformly from the unit sphere, we know that for each realization of $w^*$, only examples in $R := \{x \mid w^* \cdot x > t^*\}$ have non-zero labels. This implies that the number of queries that fall into this region is

$$r \Pr_{x \sim B^d(\sqrt{d+\Delta})}(x_1 \geq t^*) \leq \tilde{O}(r \frac{1}{t^*} \exp(-\frac{(t^*)^2}{2(1 + d^{\alpha-1})})) = \tilde{O}(rp^{1-o(1)}),$$

for $\alpha \in (0, 1)$. Thus, unless $r \geq \Omega(1/p^{1-o(1)})$, no query will have a non-zero response, and thus it is impossible to distinguish whether the ground truth is $h_1$ or $h_2$.

We next establish the lower bound on $d$. We show that this even holds for learning a homogeneous ReLU with error $\Omega(1)$ in the realizable setting. We consider an even simpler model in the realizable setting, where $w^* \in \mathbb{S}^{d-1}$ and $t^* = 0$. Furthermore, for every query $x$ made by the learner, we additionally provide the information $w^* \cdot x$. Denote by $L$ the subspace spanned by $x^{(1)}, \ldots, x^{(r)}$, the queries made by the learner. We assume that $r \leq d/\log(d)$. Denote by $w_L^* = \text{proj}_L(w^*)$. Since $w^* \sim \mathbb{S}^{d-1}$, given $w_L^*$, we know that the orthogonal component $w_{L^\perp}^*$ is a random vector drawn from the ball in $L^\perp$ with radius $\sqrt{1 - \|w_L^*\|^2}$. Let $D$ be the distribution of $w_{L^\perp}^*$. Now for any fixed hypothesis $h$, we consider the expected error of $h$ when $w_{L^\perp}^* \sim D$. We have

$$\mathop{\mathbf{E}}_{w_{L^\perp}^* \sim D} \mathop{\mathbf{E}}_{x \sim N(0,I)} (h(x) - \sigma(w_L^* \cdot x + w_{L^\perp}^*))^2 = \mathop{\mathbf{E}}_{x \sim N(0,I)} \mathop{\mathbf{E}}_{w_{L^\perp}^* \sim D} (h(x) - \sigma(w_L^* \cdot x + w_{L^\perp}^*))^2$$

$$\geq \mathop{\mathbf{E}}_{x \sim N(0,I)} \text{Var}_{w_{L^\perp}^* \sim D}(\sigma(w_L^* \cdot x + w_{L^\perp}^* \cdot x))$$

$$= \mathop{\mathbf{E}}_{x \sim N(0,I)} \text{Var}_{w_{L^\perp}^* \sim D}(\sigma(w_L^* \cdot x_L + w_{L^\perp}^* \cdot x_{L^\perp})) .$$

To simplify the notation, we denote by $T = w_L^* \cdot x_L$ the threshold of the ReLU, $\sigma(w_L^* \cdot x_L + w_{L^\perp}^* \cdot x_{L^\perp})$. Notice that since $r \leq d/\log(d)$, with probability at least $2/3$, $\|w_L^*\| \leq 1/\log(d)$ and $|T| \leq 1/\log(1/d)$. Denote by $D'$ the uniform distribution over the ball in $L^\perp$ with radius $r = \left\|\sqrt{1 - \|w_L^*\|^2}\right\| \|x_{L^\perp}\|$. For every fixed $x_{L^\perp}$, we have

$$\text{Var}_{w_{L^\perp}^* \sim D}(\sigma(T + w_{L^\perp}^* \cdot x_{L^\perp})) = \text{Var}_{x \sim D'}(\sigma(T + e_1 \cdot x)),$$

where $e_1$ is the first standard basis vector in $L^\perp$. Since $L$ has dimension $d(1 - O(1/\log(d)))$, we know that when $\|x_{L^\perp}\| \geq \Omega(\sqrt{d})$, and thus $r > \Omega(\sqrt{d})$, $\text{Var}_{x \sim D'}(\sigma(T + e_1 \cdot x)) \geq \Omega(1)$. Since $x_{L^\perp}$ and $x_L$ are independent, we know that with probability at least $2/3$, $\|x_{L^\perp}\| \geq \Omega(\sqrt{d})$. Thus, for every learning algorithm, if it makes fewer than $d/\log(d)$ queries, with probability at least $2/3$, the expected error of the output hypothesis is at least $\mathbf{E}_{w_{L^\perp}^* \sim D} \mathbf{E}_{x \sim N(0,I)}(h(x) - \sigma(w_L^* \cdot x + w_{L^\perp}^*))^2 \geq \Omega(1)$. $\quad\square$

## D.2 Proof of Theorem 4.2

We restate the theorem below.

**Theorem D.2.** *For any active learning algorithm $\mathcal{A}$, there is an activation function $h^*$ that labels $S$ with bias $p$ such that if $\mathcal{A}$ makes less than $\tilde{O}(d/(p\log(m)))$ label queries over $S$, a set of $m$ i.i.d. points drawn from $N(0, I)$, then with probability at least $2/3$ the hypothesis $\hat{h}$ output by $\mathcal{A}$ has error more than $\tilde{O}(p)$ with respect to $h^*$.*

To begin with, we establish the following lemma that reduces the learning problem to a slightly easier problem of finding examples with non-zero labels from a pool of unlabeled examples.

**Lemma D.3.** *Suppose there is an active learning algorithm that can make $r$ label queries over a pool $S$ of $m \geq \text{poly}(d/p)$ examples drawn from $N(0, I)$ and learn any ReLU activation function $h^*(x) = \sigma(w^* \cdot x - t^*)$ with bias $p$ up to error $\tilde{O}(p)$ with probability at least $2/3$. Then there is*

*an algorithm such that given a pool of $2m$ random examples $S$ drawn from the standard Gaussian distribution with hidden labels by some ReLU activation function $h^*(x) = \sigma(w^* \cdot x - t^*)$ with bias $p$, it makes $r + O(d)$ queries and finds $d$ examples with non-zero labels. from $S$ with probability $1/2$.*

*Proof of Lemma D.3.* Let $\mathcal{A}$ be such a learning algorithm. We select a random set of $m$ examples $S_1$ and give it to $\mathcal{A}$. We know that with probability $1/2$, we learn a hypothesis $h$ such that $\mathbf{E}_{x \sim N(0,I)} (h(x) - h^*(x))^2 \leq \tilde{O}(p)$. We first show that if $x \sim N(0, I)$, then with probability at least $\Omega(p)$, $h(x) > 1/(2t)$. This is because otherwise

$$\mathbf{E}_{x \sim N(0,I)} (h(x) - h^*(x))^2 \mathbb{1}\{h(x) < 1/(2t), h^*(x) > 1/t\}$$

$$\geq \Omega(1/t^2) \mathbf{Pr}_{x \sim N(0,I)} (h(x) < 1/(2t), h^*(x) > 1/t) = \tilde{\Omega}(p).$$

On the other hand, if $x \sim N(0, I)$ and $h(x) > (1/2t)$, then with probability at least $1/2$, $h^*(x) > 0$. Suppose this is not correct, we have

$$\mathbf{E}_{x \sim N(0,I)} (h(x) - h^*(x))^2 \mathbb{1}\{h(x) > 1/(2t), h^*(x) \leq 0\}$$

$$\geq \Omega(1/t^2) \mathbf{Pr}_{x \sim N(0,I)} (h(x) > 1/(2t), h^*(x) \leq 0) = \tilde{\Omega}(p).$$

Since $m$ is at least $\text{poly}(d, 1/p)$, we know that with enough high probability, at least $\Omega(d)$ examples will satisfy $h(x) > 1/(2t)$ and at least a constant fraction of these examples will satisfy $h^*(x) > 0$. Thus, given such a $h$ with probability at least $3/4$, we can find $d$ examples with non-zero label in $S$ by randomly querying $O(d)$ examples with prediction $h(x) > 1/(2t)$. $\qquad \square$

Based on this, we can give the proof of Theorem 4.2.

*Proof of Theorem 4.2.* Consider the problem where an algorithm wants to find $k$ examples with non-zero labels from $m$ unlabeled examples by making $r$ queries. By Lemma D.3, it is sufficient for us to prove the hardness of such a problem by finding suitable parameters $k, r$.

Consider any deterministic learning algorithm $\mathcal{A}$. Given a pool of $m$ unlabeled examples, we describe $\mathcal{A}$ in the following way. For every $w^*$, the implementation of $\mathcal{A}$ can be described as a path $P = ((x^{(1)}, y^{(1)}), \ldots, (x^{(r)}, y^{(r)}))$, with length at most $r$. In particular, a path, along which $\mathcal{A}$ successfully finds $k$ examples with non-zero, can be uniquely represented as the indices $i_1 < \cdots < i_k$ and the corresponding label $(y^{(i_1)}, \ldots, y^{(i_k)})$. Next, we consider a fixed tuple of $k$ indices $i_1, \ldots, i_k$ and denote by $S_c$ the set of all $(x^{(i_1)}, y^{(i_1)}), \ldots, (x^{(i_k)}, y^{(i_k)})$ that make algorithm $\mathcal{A}$ succeed at positions $i_1 < \cdots < i_k$. By choosing $w^* \sim \mathbb{S}^{d-1}$ uniformly, we will show that the probability of $S_c$ is very small. Notice that for any $(x^{(i_1)}, y^{(i_1)}), \ldots, (x^{(i_k)}, y^{(i_k)}) \in S_c$, it corresponds to a unique event $(x^{(i_1)}, q^{(i_1)}), \ldots, (x^{(i_k)}, q^{(i_k)})$, where $q^{(i_j)} = w^* \cdot x^{(i_j)}$ for $j \in [k]$, since $y_{ij} > 0$ and $q_{ij} = t^* + y_{ij}$. We denote the marginal density on this event as $f((x^{(i_1)}, q^{(i_1)}), \ldots, (x^{(i_k)}, q^{(i_k)}))$. Notice that $\mathbf{Pr}(S_c) = \int_{q_{i_1}, \cdot, q_{i_k} > t^*} f((x^{(i_1)}, q^{(i_1)}), \ldots, (x^{(i_k)}, q^{(i_k)}))$. So, it is sufficient to upper bound this integral.

Denote by $L$ the subspace spanned by $\{x^{(i_1)}, \ldots, x^{(i_k)}\}$. We consider a basis $\{b_1, \ldots, b_k\}$ of $L$ as follows. $b_1 = x^{(i_1)} / \|x^{(i_1)}\|$ and $b_j = \text{proj}_{L_{j-1}} x^{(i_j)} / \|\text{proj}_{L_{j-1}} x^{(i_j)}\|$, where $L_{j-1} = span\{x^{(i_1)}, \ldots, x^{(i_{j-1})}\}$. We know that there is a unique $w_L := \text{proj}_L(w^*)$ such that $w_L \cdot x^{(i_j)} = q^{(i_j)}$ for $j \in [k]$. This implies that events $(x^{(i_1)}, y^{(i_1)}), \ldots, (x^{(i_k)}, y^{(i_k)})$ can be precisely described as a $k$-dimensional vector $v(w_L) := (w_L \cdot b_1, \ldots, w_L \cdot b_k)$. Denote by $L_0 = span(e_1, \ldots, e_k)$. We have

$$f((x^{(i_1)}, q^{(i_1)}), \ldots, (x^{(i_k)}, q^{(i_k)})) = p(\text{proj}_L(w^*) = w_L) = p(\text{proj}_{L_0}(w^*) = v(w_L)) ,,$$

where $p(\text{proj}_L(w^*) = w_L)$ is the density function of the event that $\text{proj}_L(w^*) = w_L$, $w^* \sim \mathbb{S}^{d-1}$ Consider the set $S_w$ of all possible $v(w_L)$ such that $w_L$ corresponds to a realization $((x^{(i_1)}, q^{(i_1)}), \ldots, (x^{(i_k)}, q^{(i_k)}))$. We notice that, given any $v(w_L) \in S_w$, we can uniquely recover the corresponding $(x^{(i_1)}, q^{(i_1)}), \ldots, (x^{(i_k)}, q^{(i_k)}) \in S_c$ via $\mathcal{A}$. This implies

$$\mathbf{Pr}(S_c) = \int_{q_{i_1}, \cdot, q_{i_k} > t^*} f((x^{(i_1)}, q^{(i_1)}), \ldots, (x^{(i_k)}, q^{(i_k)})) = \int_{S_w} p(\text{proj}_{L_0}(w^*) = v(w_L)) = \mathbf{Pr}_{w^*}(S_w).$$

To upper bound $\mathbf{Pr}(S_w)$, it is sufficient to upper bound the probability of a super-set of $S_w$. Since $w^* \cdot x^{(i_j)} = q^{(i_j)} > t^*$ for $j \in [k]$, we know that

$$v := (w^* \cdot x^{(i_1)}, \ldots, w^* \cdot x^{(i_k)})^\top$$

satisfies $\|v\|^2 = \sum_{j \in [k]} (q^{(i_j)})^2 \geq k(t^*)^2$. This implies that, for every realization, the square of the norm of the projection of $w^*$ onto the subspace, $\|w_L\|^2$, is

$$\begin{aligned} B &:= (w^*)^\top A^\top (AA^\top)^{-1} A w^* = v^\top (AA^\top)^{-1} v \geq \|v\|^2 / \|AA^\top\|_2 \\ &\geq k(t^*)^2 / \|AA^\top\|_2 , \end{aligned}$$

where $A \in \mathbb{R}^{k \times d}$ is the matrix with row vectors $x^{(i_1)}, \ldots, x^{(i_k)}$.

We next make use of the following structural lemma from [DKM24].

**Lemma D.4.** *Let $S \subseteq \mathbb{R}^d$ be a set of $m$ examples drawn i.i.d. from $N(0, I)$. Let $t^* > C > 0$ for a sufficiently large constant $C$ and $k = O(d/\log(m)(t^*)^4)$. Then, with probability at least $2/3$, for every $k$-tuple of examples $\{x_1, \ldots, x_k\} \subseteq S$, $\|AA^\top - dI\|_2 \leq d/(t^*)^2$, where $A \in \mathbb{R}^{k \times d}$ be a matrix with row vectors $x_1, \ldots, x_k$.*

That is to say, we are able to bound $\|AA^\top\|_2$ by $d(1 + O(1/(t^*)^2))$ and thus every vector $v(w_L)$ has squared norm at least $(k(t^*)^2/d(1 + O(1/(t^*)^2)))$. Since $w^*$ is uniformly chosen from the unit sphere, by Lemma B.1 in [KMT24c], the square norm of $w^*$ projected onto a fixed $k$-dimensional subspace is a random variable drawn from a Beta distribution $B(\frac{k}{2}, \frac{d-k}{2})$. By Lemma 2.2 in [DG03], We have

$$\begin{aligned} \mathbf{Pr}\left(\|w_{L_0}^*\|^2 \geq k(t^*)^2/d(1 + O(1/(t^*)^2)))\right) &\leq \exp\left(-\frac{k}{2}\left(\frac{d(t^*)^2}{d(1 + O(1/(t^*)^2)))} - 1 - \log\left(\frac{d(t^*)^2}{d(1 + O(1/(t^*)^2)))}\right)\right)\right) \\ &= \left(\sqrt{\frac{(t^*)^2 d}{d(1 + O(1/(t^*)^2)))}} \exp\left(-\frac{1}{2}\left(\frac{d(t^*)^2}{d(1 + O(1/(t^*)^2)))} - 1\right)\right)\right)^k \\ &\leq \left(O((t^*) \exp\left(-\frac{(t^*)^2}{2}(1 - O(1/(t^*)^2)))\right)\right)^k \\ &= \left(O((t^*)^2 \frac{1}{t^*} \exp\left(-\frac{(t^*)^2}{2}\right))\right)^k \leq (O(p \log(1/p)))^k . \end{aligned}$$

This implies that by choosing $k = O(d/\log(m)(t^*)^4)$, for every $k$ tuple of indices, the probability of success for $\mathcal{A}$ is at most $(O(p \log(1/p)))^k$. Since there are at most $\binom{r}{k}$ such tuples, by the union bound,

$$\binom{r}{k} \alpha^k \leq \left(\frac{er}{k} O(p \log(1/p))\right)^k \leq 2/3 ,$$

if $r \leq O(k/p \log(1/p)) = O(d/(p \log(m)\mathrm{polylog}(1/p))$. This implies that if we make less than $r \leq O(k/p \log(1/p)) = O(d/(p \log(m)\mathrm{polylog}(1/p))$ queries, then it is hard to find $k = O(d/\log(m)(t^*)^4)$. $\qquad\square$

