# OpenReview forum: "Robust Regression of General ReLUs with Queries"
_NeurIPS.cc/2025/Conference — NeurIPS 2025 poster_

### Official Review · Reviewer_huHJ · 2025-06-26

**Clarity:** 2
**Significance:** 4
**Originality:** 4
**Rating:** 5
**Confidence:** 2

**Summary:**

This paper considers active learning, where an algorithm can query for the label corresponding to any data instance. This is a stronger oracle model than pool-based active learning where the unlabeled dataset is fixed.

The main contribution is a computationally-efficient algorithm that achieves a label complexity that nearly matches the lower bound. The authors show that this label complexity is lower than what is required for pool-based active learning, establishing the benefit of the stronger oracle model.

**Questions:**

* Could the authors describe the intuitions underlying the proofs? It seems that each step of the algorithm could be explained/motivated separately, and how they contribute to the overall complexity can be described in isolation.

* What are the main bottlenecks in generalizing to more general marginal distributions? What is it about the Gaussian distribution that makes the analysis possible? Is there a natural extension to log-concave distributions, exponential families or some other classes of marginal distributions? While the analysis' being focused on Gaussian marginals is understandable, this would help better understand the intuition behind the improved label complexity that arises in the considered setting.

**Ethical Concerns:**

["NO or VERY MINOR ethics concerns only"]

**Final Justification:**

I assign high weight to the paper's technical novelty. The paper presents a novel and interesting result.

My primary concern was the lack of intuition behind the main proof, which is significant for a theory paper. The authors' rebuttal successfully addressed this by providing a high-level summary. Because the authors have shown that they are able to provide the needed intuition, I recommend acceptance with the expectation that this will be integrated into the camera-ready version.

**Limitations:**

Yes

**Quality:**

3

**Strengths And Weaknesses:**

**Strengths**
* This paper identifies a clear gap in the literature and closes this gap. I especially value how the authors demonstrate improved label complexity that comes with the stronger oracle than pool-based active learning.

**Weaknesses**
* The paper is very dense. The introduction is a summary of the main results and a comparison with prior work, and there is no separate related works section.

* The techniques used in this paper are specific to a Gaussian marginal distribution. The proofs rely heavily on the fact that they are Gaussian, e.g., relating to the Ornstein–Uhlenbeck semigroup. The proofs are very technical, with limited insights into how the arguments used in the paper extend to different marginals. It would be nice if the authors could describe some intuition on what the Gaussian marginals entail - is it expected that the problem becomes harder or easier for other marginal distributions?

**Some Comments**
This paper solves a technically involved and important problem, and so it is understandable that the analysis is not too easy to follow. However, I find the Appendix to be a better description of the ideas underlying the algorithm and proof. I believe the paper presentation can be improved by modifying Section 3. For example, the authors could first clearly write out the main steps and then describe each step in detail in different subsections.

---

> ### Author Rebuttal · Authors · 2025-07-31
>
> We thank the reviewer for the positive evaluation of our work and constructive suggestions. Due to the space limitations, we were not able to provide a detailed description of the algorithm in Section 3 but only the key ideas and high-level intuition. We will provide a more detailed description in the revised version of the manuscript.
>
> >Question:Could the authors describe the intuitions underlying the proofs? It seems that each step of the algorithm could be explained/motivated separately, and how they contribute to the overall complexity can be described in isolation.
>
> Here, we give a brief summary of the intuition behind our main algorithm; the detailed descriptions can be found in Section 3 and Appendix C.3.
>
> Our algorithm proceeds in a sequence of rounds, improving parameters $(r,t)$ and the direction w separately in each round of update. To start with, we will guess a pair of $(r_0,t_0)$ that is non-trivially close to $(r*,t*)$.
> Since the guessing step has a probability of success at least $1/polylog(R/\epsilon)$, we can run the same algorithm poly-logarithmically times to boost the probability of success. So, we can assume that we start with a good $(r_0,t_0)$, and we use it to get a warm start $w_0$ by making at most $1/p + d polylog(R/\epsilon)$ queries.
>
> With a good warm start $(r_0,t_0)$ and $w_0$, we improve these parameters via a gradient method in each round. Specifically, in each round, we maintain a pair of $(r_i,t_i)$ and $w_i$ and assume that $(r*)^2\theta_i^2 \le B_i^2$, where $B_i$ is some suitable constant used for measuring the quality of $w_i$.
>
> The intuition here is that if the current $(r_i,t_i)$ is not good enough, then the gradient of $w_i$ will not point in the wrong direction, but only have large variance. So, if $(r_i,t_i)$ is suitably good, we can use $d polylog(R/\epsilon)$ queries to get a relatively good estimation of the gradient of $w_i$ and improve it so that $(r*)^2(\theta_{i+1}^2) \le B_{i+1}^2$, for some $B_{i+1}<B_i$. On the other hand, if $w_i$ has already been improved well enough, we are able to estimate the gradient $(r_i,t_i)$ using $polylog(R/\epsilon)$ queries. By comparing the length of the gradient of $(r_i,t_i)$ with $B_i$, we are able to tell if the current $(r_i,t_i)$ meets the requirement for updating $w_i$. If it meets the requirement, we do not update $(r_i,t_i)$; otherwise, we run a gradient step for polylogarithmically many rounds to improve $(r_i,t_i)$ so that it meets the requirement for updating $w_i$. In summary, in each round of update we make $d polylog(R/\epsilon)$ queries.
>
> After polylogarithmically many rounds, we improve $w$ to the desired accuracy. But due to noise, if $t*$ is large, we are only able to get some $(r,t)$ so that it is within the $\epsilon polylog(R/\epsilon)$ neighborhood of $(r*,t*)$. Since there are only two parameters, by using a brute-force approach, we can find polylogarithmically many hypotheses such that one of them is guaranteed to be good enough. Finally, by a hypothesis selection procedure over the candidate hypotheses, we find a good hypothesis. This gives the final query complexity.
>
> >Question:The techniques used in this paper are specific to a Gaussian marginal distribution. The proofs rely heavily on the fact that they are Gaussian, e.g., relating to the Ornstein–Uhlenbeck semigroup.
> What are the main bottlenecks in generalizing to more general marginal distributions? What is it about the Gaussian distribution that makes the analysis possible? Is there a natural extension to log-concave distributions, exponential families or some other classes of marginal distributions?
>
> We acknowledge that the Gaussian assumption is a strong distributional assumption. On the other hand, we note that it is one of the prototypical assumptions in our setting that has been extensively studied in prior work. Specifically, the task of designing efficient and robust algorithms for robust ReLU regression under the Gaussian distribution has been studied in a sequence of recent works in the passive setting; see, e.g., [ATV23,GV24,DKTZ22]. In fact, it was only very recently that a (passive) polynomial time learner for a general ReLU with $O(opt)$ error under adversarial label noise was developed—without label complexity constraints. Moreover, to date, there is no known polynomial time algorithm known with  $O(opt)$ error beyond the Gaussian distribution.
>
> Technically, for a more general distribution $D_x$, we do not have a clean way to characterize the distance between the current hypothesis and the target hypothesis when a large threshold exists in the target hypothesis. This makes it hard to analyze the connection between the gradient in each round and the error of the current hypothesis.
>
>
> On the other hand, if we are willing to relax the final error guarantee, to say $O(\sqrt{opt})$, then it is plausible that we could design a polynomial-time algorithm for more general distribution families. In fact, prior work [DKS18] developed a polynomial time passive learner for general halfspaces (a closely related class of single-index models) under log-concave distributions to error $O(\sqrt{opt})$. Developing a query-optimal algorithm for general ReLUs in the log-concave setting with similar error guarantees is an interesting question for future work.
>
>
> >[DKZ20] Diakonikolas I, Kane D, Zarifis N. Near-optimal sq lower bounds for agnostically learning halfspaces and relus under gaussian marginals. Advances in Neural Information Processing Systems. 2020;33:13586-96.
>
> >[DKTZ22] Diakonikolas I, Kontonis V, Tzamos C, Zarifis N. Learning a single neuron with adversarial label noise via gradient descent. InConference on learning theory 2022 Jun 28 (pp. 4313-4361). PMLR.
>
> >[ATV23] Awasthi P, Tang A, Vijayaraghavan A. Agnostic learning of general relu activation using gradient descent. arXiv preprint arXiv:2208.02711. 2022 Aug 4.
>
> >[GV24] Guo A, Vijayaraghavan A. Agnostic learning of arbitrary ReLU activation under Gaussian marginals. arXiv preprint arXiv:2411.14349. 2024 Nov 21.
>
> >[DKKTZ24]Diakonikolas I, Kane DM, Kontonis V, Tzamos C, Zarifis N. Agnostically learning multi-index models with queries. In2024 IEEE 65th Annual Symposium on Foundations of Computer Science (FOCS) 2024 Oct 27 (pp. 1931-1952). IEEE.
>
> >[DKS18]Diakonikolas I, Kane DM, Stewart A. Learning geometric concepts with nasty noise. InProceedings of the 50th Annual ACM SIGACT Symposium on Theory of Computing 2018 Jun 20 (pp. 1061-1073).
>
> >[DKR23]Diakonikolas I, Kane D, Ren L. Near-optimal cryptographic hardness of agnostically learning halfspaces and relu regression under gaussian marginals. InInternational Conference on Machine Learning 2023 Jul 3 (pp. 7922-7938). PMLR.

---

> > ### Comment · Reviewer_huHJ · 2025-08-05
> >
> > Thank you for the detailed response. I highly encourage the authors to include the high level description behind the algorithm and the analysis in the main manuscript in the next revision.

---

### Official Review · Reviewer_6Jjm · 2025-07-03

**Clarity:** 2
**Significance:** 3
**Originality:** 3
**Rating:** 5
**Confidence:** 3

**Summary:**

The paper studies the problem of agnostically learning ReLUs (with arbitrary biases) under the Gaussian distribution. They work in the label query model, where the learner can query an oracle for the (possibly noisy) value $y(x)$ at a datapoint $x$. The aim is to find a ReLU hypothesis $\hat{h}$, such that the squared loss of $\hat{h}$ is at most $O(OPT) +\epsilon$ where $OPT$ is the squared loss of the best ReLU hypothesis. They solve this problem by designing a projected gradient descent (with warm start) based algorithm that uses $M= d\text{poly}\log (1/\epsilon)+\min (1/p,1/\epsilon)$ queries, and runs in time $\text{poly}(M)$ This was previously only known for the case where the ground truth ReLU was $\max (w^* \cdot x- t)$ with threshold $t\leq 0$. In this work, they handle arbitrary thresholds. Here $p= Pr_{X\sim \mathcal{N}(0,I)}[w^*\cdot x \geq t]$. This is in contrast to the setting of learning with samples, where the best algorithms draw $\text{poly}( d/\epsilon)$ samples. Here, the authors leverage the query access model to obtain poly log sample complexity in the error parameter.

They support their upper bound with an information theoretic lower bound on the query complexity. They also show that any pool based active learner (algorithm first selects a pool of unlabelled samples and is allowed to query labels only for examples in this set) that doesnt draw a large number of unlabelled samples needs more query complexity than the learner equipped with label queries. This shows a separation in the power of the two model.

**Questions:**

See Weaknesses

**Ethical Concerns:**

["NO or VERY MINOR ethics concerns only"]

**Final Justification:**

I should like to keep my score. The responses of the authors were satisfactory.

**Limitations:**

This is a theoretical paper. I don't see any negative societal impact.

**Quality:**

4

**Strengths And Weaknesses:**

## Strengths
- The paper's main algorithm (other than the initialization/ warm start) is projected gradient descent. This is a practically successful algorithm and this paper adds to the theoretical evidence supporting the empirical success of gradient descent.
- The analysis is highly non-trivial and cleverly employs various techniques from the analysis of Gaussian space and gradient descent.
- Achieving polylog query complexity requires a lot of fine grained work, for instance, even hypothesis selection is non-trivial (see Lemma 2.11)
- The question of learning ReLU activations is a fundamental question in learning theory, and this paper makes progress on it.

## Weaknesses
- The analysis only works in Gaussian space and uses many sophisticated tricks from Gaussian integration.
- The technical proofs in the appendix are very heavy in equations. I suggest adding supporting text whenever possible.
- Section 2 contains many structural lemmas (Lemma 2.2-2.5) . Are all of these novel to this paper, or have similar analyses appeared before? For instance Lemma 2.2 seems to be a natural statement linking the loss function to the angle between the vectors  with a natural Gaussian integration based proof.
- The use of the term "bias" for $p$ seems a bit confusing (as it usually refers to the threshold of the relu, or even the expectation of the function). Is this terminology standard?

---

> ### Author Rebuttal · Authors · 2025-07-31
>
> We thank the reviewer for their constructive feedback. We will incorporate the reviewer’s suggestions regarding presentation in the future version of this manuscript. We next address the comments and questions by the reviewer separately.
>
> >Question: The analysis only works in Gaussian space and uses many sophisticated tricks from Gaussian integration
>
> We acknowledge that the Gaussian assumption is a strong distributional assumption. On the other hand, we note that it is one of the prototypical assumptions in our setting that has been extensively studied in prior work. Specifically, the task of designing efficient and robust algorithms for robust ReLU regression under the Gaussian distribution has been studied in a sequence of recent works in the passive setting; see, e.g., [ATV23,GV24,DKTZ22]. In fact, it was only very recently that a (passive) polynomial time learner for a general ReLU with $O(opt)$ error under adversarial label noise was developed—without label complexity constraints. Moreover, to date, there is no known polynomial time algorithm known with  $O(opt)$ error beyond the Gaussian distribution.
>
> Technically, for a more general distribution $D_x$, we do not have a clean way to characterize the distance between the current hypothesis and the target hypothesis when a large threshold exists in the target hypothesis. This makes it hard to analyze the connection between the gradient in each round and the error of the current hypothesis.
>
>
> On the other hand, if we are willing to relax the final error guarantee, to say $O(\sqrt{opt})$, then it is plausible that we could design a polynomial-time algorithm for more general distribution families. In fact, prior work [DKS18] developed a polynomial time passive learner for general halfspaces (a closely related class of single-index models) under log-concave distributions to error $O(\sqrt{opt})$. Developing a query-optimal algorithm for general ReLUs in the log-concave setting with similar error guarantees is an interesting question for future work.
>
> >Question: Section 2 contains many structural lemmas (Lemma 2.2-2.5) . Are all of these novel to this paper, or have similar analyses appeared before? For instance Lemma 2.2 seems to be a natural statement linking the loss function to the angle between the vectors with a natural Gaussian integration based proof.
>
> While Lemma 2.5 follows a standard analysis of projected gradient descent, Lemmas 2.2-2.4 are novel. For example, for Lemma 2.2, although prior works on learning other Gaussian single index models (such as halfspaces or homogeneous ReLUs) derived relations between the angle and error, their analysis heavily relies on the specific activation functions. On the other hand, we make use of the OU-operations to give a closed form analysis between the angle, error, and the gradient, for general activation functions.
>
>
> >[DKZ20] Diakonikolas I, Kane D, Zarifis N. Near-optimal sq lower bounds for agnostically learning halfspaces and relus under gaussian marginals. Advances in Neural Information Processing Systems. 2020;33:13586-96.
>
> >[DKTZ22] Diakonikolas I, Kontonis V, Tzamos C, Zarifis N. Learning a single neuron with adversarial label noise via gradient descent. InConference on learning theory 2022 Jun 28 (pp. 4313-4361). PMLR.
>
> >[ATV23] Awasthi P, Tang A, Vijayaraghavan A. Agnostic learning of general relu activation using gradient descent. arXiv preprint arXiv:2208.02711. 2022 Aug 4.
>
> >[GV24] Guo A, Vijayaraghavan A. Agnostic learning of arbitrary ReLU activation under Gaussian marginals. arXiv preprint arXiv:2411.14349. 2024 Nov 21.
>
> >[DKKTZ24]Diakonikolas I, Kane DM, Kontonis V, Tzamos C, Zarifis N. Agnostically learning multi-index models with queries. In2024 IEEE 65th Annual Symposium on Foundations of Computer Science (FOCS) 2024 Oct 27 (pp. 1931-1952). IEEE.
>
> >[DKS18]Diakonikolas I, Kane DM, Stewart A. Learning geometric concepts with nasty noise. InProceedings of the 50th Annual ACM SIGACT Symposium on Theory of Computing 2018 Jun 20 (pp. 1061-1073).
>
> >[DKR23]Diakonikolas I, Kane D, Ren L. Near-optimal cryptographic hardness of agnostically learning halfspaces and relu regression under gaussian marginals. InInternational Conference on Machine Learning 2023 Jul 3 (pp. 7922-7938). PMLR.

---

> > ### Comment · Reviewer_6Jjm · 2025-08-05
> > **Response to rebuttal**
> >
> > Thank you for your rebuttal. I will keep my score.

---

### Official Review · Reviewer_ezZp · 2025-07-04

**Clarity:** 3
**Significance:** 3
**Originality:** 3
**Rating:** 4
**Confidence:** 4

**Summary:**

This paper addresses problem of agnostically learning a general ReLU function under the Gaussian distribution with respect to the squared loss. The authors focus on the interactive learning setting, where the learner has black-box query access to labels. The main contribution is a computationally efficient algorithm that achieves an error of O(opt) + ε with a query complexity of d*polylog(1/ε) + Õ(min{1/p, 1/ε}), where p is the bias of the optimal ReLU.  The authors complement their algorithm with a nearly-matching information-theoretic lower bound, demonstrating the near-optimality of their approach. Furthermore, they establish a strong separation result, proving that the weaker pool-based active learning model cannot achieve similar query improvements, thus justifying their use of a stronger membership query oracle.

**Questions:**

- How can the results be generalized to general sub-gaussian distributions over the input space Dx?
- What are the challenges in obtaining an optimality guarantee on $opt +\epsilon$ rather than O(opt) + \epsilon?
- Optimality Gap (Theorems 1.1 & 1.2): There is a small gap between the upper bound's Õ(1/p) and the lower bound's Ω(1/p^(1-o_d(1))) complexity. Do you believe this gap is an artifact of the analysis, or could there be a fundamental reason for the p^(-o_d(1)) factor?
- What is the dependence on $\delta$ on the sample complexity?

**Ethical Concerns:**

["NO or VERY MINOR ethics concerns only"]

**Final Justification:**

I would like to keep my score and recommend this paper for acceptance.

**Limitations:**

Yes

**Quality:**

3

**Strengths And Weaknesses:**

Strength:
- The paper presents near optimal sample complexity in the interactive learning setting, along with an almost matching lower bound.
- The paper also shows the importance of the interactive setup showing a lower bound on pool-based active learning algorithm.
- The authors start off with the simpler of setup of learning a spherical GLM and generalize to the ReLU setup later.

Weaknesses:
- The bias p, defined on line 101, is used extensively in the introduction  to showcase the main results. It would be better to define it earlier in the introduction. Also, the term ‘bias’ is used confusingly to denote both p and t in the paper.
- Despite the simpler setup, the overall presentation is a bit too dense and it is challenging to verify the correctness of each of the claims.
- The oracle model with the interactive querying can be a pretty strong. Would a slightly stronger pool based membership query work? For example, a local pool based membership oracle which allows the learner to query in a neighborhood of the points sampled from the input distribution Dx?

---

> ### Author Rebuttal · Authors · 2025-07-31
>
> We thank the reviewer for the constructive feedback. We will incorporate the reviewer’s suggestions regarding presentation and be more explicit about the use of notations in the future version of this manuscript. We next address the comments and questions by the reviewer separately.
>
> >Question: The oracle model with interactive querying can be pretty strong. Would a slightly stronger pool based membership query work? For example, a local pool based membership oracle which allows the learner to query in a neighborhood of the points sampled from the input distribution $D_x$?
>
> We first want to point out that although the membership query model is a strong model in the realizable setting, in the agnostic setting, the response to a query could be arbitrary and thus captures many interesting practical applications. We also want to mention that the reviewer’s question on learning under the pool-based membership query model makes a very interesting point.  The model of local membership queries is related to the high-level idea behind our algorithm. Specifically, given a warm start, our algorithm can be implemented in the standard pool-based active learning setting. The use of the membership query oracle is only required in the stage of finding a warm-start. Intuitively, finding such a warm-start requires at least $d$ examples with non-zero labels, which in turn requires many unlabeled examples (as demonstrated in our lower bound). Using membership queries, given a random example with non-zero labels, by querying within a small neighborhood of this example, we can efficiently find the other $d-1$ examples with non-zero labels. (Such an intuition is also mentioned in the introduction of our paper for realizable ReLU regression as a motivation for our robust algorithm.) That said, a formal analysis in this setting would depend on the precise definition of the local query model. Specifically, if we are only able to query an extremely small neighborhood of a point in the pool, this would not make much difference from the pool-based active learning setting (for which we have already obtained a lower bound). It would be an interesting direction to understand the tradeoff between the query complexity and the size of the neighborhood we are allowed to query.
>
> >Question: How can the results be generalized to general sub-gaussian distributions over the input space $D_x$?
>
> To properly answer this question, we provide the relevant context.
>
> The goal of this paper is to get an efficient algorithm that achieves $O(opt)+\epsilon$ error under adversarial label noise with an optimal label complexity under the Gaussian distribution. This is a fundamental problem, even if we ignore the label complexity requirement, and a line of work has focused on it in the standard passive setting. In particular, the passive learning version of the problem has been studied in several prior works, such as [ATV23,GV24,DKTZ22]. Perhaps surprisingly, it was only very recently that a (passive) polynomial-time algorithm for a general ReLU with $O(opt)$ error under adversarial label noise was developed. To date, there is no known polynomial-time algorithm  that achieves $O(opt)$ error beyond the case of the Gaussian distribution. One of the technical obstacles is that for a more general distribution $D_x$, we do not have a clean way to precisely characterize the distance between the current hypothesis and the target hypothesis when a large threshold exists in the target hypothesis. This makes it challenging to analyze the connection between the gradient in each round and the error of the current hypothesis.
>
> On the other hand, if we are willing to relax the final error guarantee, to say $O(\sqrt{opt})$, then it is plausible that we could design a polynomial-time algorithm for more general distribution families. In fact, prior work [DKS18] developed a polynomial time passive learner for general halfspaces (a closely related class of single-index models) under log-concave distributions to error $O(\sqrt{opt})$. Developing a query-optimal algorithm for general ReLUs in the log-concave setting with similar error guarantees is an interesting question for future work.
> Finally, regarding the case of arbitrary sub-Gaussian distributions, we believe that robust learning with queries to error $O(opt)+\epsilon$ with the optimal label complexity is challenging, in part due to the fact that an interactive learner usually needs to take advantage of prior knowledge of the marginal distribution.
>
> >Question: What are the challenges in obtaining an optimality guarantee on opt+\epsilon  rather than O(opt) + \epsilon?
>
>  As mentioned in the introduction of our paper (line 28), prior work such as [DKZ20, DKR23] have shown that even for the task of agnostically learning a homogeneous ReLU, achieving $opt+\epsilon$ error is computationally hard (specifically, it requires time exponential in $1/\epsilon$). Furthermore, more recent work [DKKTZ24] shows that even in the membership query model, achieving $opt+\epsilon$ error is computationally hard for proper learning (recall that our algorithm is proper). In summary, there are fundamental computational limitations to obtaining error $opt+\epsilon$.
> Intuitively, for the task of ReLU regression, directions $w$ that give a hypothesis with $O(opt)$ error all fall in a small cone around $w^*$. So, as long as we detect this cone, we can solve the learning task. But within this cone, due to the adversarial label noise, the structure can be quite complicated and finding a direction within the cone with the smallest error can be computationally hard.
>
> >Question: Optimality Gap (Theorems 1.1 & 1.2): There is a small gap between the upper bound's Õ(1/p) and the lower bound's Ω(1/p^(1-o_d(1))) complexity. Do you believe this gap is an artifact of the analysis, or could there be a fundamental reason for the p^(-o_d(1)) factor?
>
> We provide a high-level intuition explaining why there is a $p^{-o_{d(1)}}$ factor in our statement. Intuitively, we want to prevent an algorithm from querying examples that are too far from the origin.
> In the realizable setting, an algorithm may only need $2$ queries to find an example with a non-zero label by querying points that are very far from the origin. To avoid this, we add a tiny fraction of noise so that examples with large norm must have $0$ labels, so that only queries within a ball with a bounded radius may have non-zero response. However, truncated within this ball, the fraction of examples with non-zero labels becomes slightly larger—since within the ball, the probability measure is slightly different from a Gaussian, and this is why we have a lower bound of $1/p^{1-o(1)}$. For large $d$, such a lower bound suffices to show the optimality of our query complexity.
>
> >Question: What is the dependence on $\delta$ the sample complexity?
>
>  It is $log(1/\delta)$
>
> >[DKZ20] Diakonikolas I, Kane D, Zarifis N. Near-optimal sq lower bounds for agnostically learning halfspaces and relus under gaussian marginals. Advances in Neural Information Processing Systems. 2020;33:13586-96.
>
> >[DKTZ22] Diakonikolas I, Kontonis V, Tzamos C, Zarifis N. Learning a single neuron with adversarial label noise via gradient descent. InConference on learning theory 2022 Jun 28 (pp. 4313-4361). PMLR.
>
> >[ATV23] Awasthi P, Tang A, Vijayaraghavan A. Agnostic learning of general relu activation using gradient descent. arXiv preprint arXiv:2208.02711. 2022 Aug 4.
>
> >[GV24] Guo A, Vijayaraghavan A. Agnostic learning of arbitrary ReLU activation under Gaussian marginals. arXiv preprint arXiv:2411.14349. 2024 Nov 21.
>
> >[DKKTZ24]Diakonikolas I, Kane DM, Kontonis V, Tzamos C, Zarifis N. Agnostically learning multi-index models with queries. In2024 IEEE 65th Annual Symposium on Foundations of Computer Science (FOCS) 2024 Oct 27 (pp. 1931-1952). IEEE.
>
> >[DKS18]Diakonikolas I, Kane DM, Stewart A. Learning geometric concepts with nasty noise. InProceedings of the 50th Annual ACM SIGACT Symposium on Theory of Computing 2018 Jun 20 (pp. 1061-1073).
>
> >[DKR23]Diakonikolas I, Kane D, Ren L. Near-optimal cryptographic hardness of agnostically learning halfspaces and relu regression under gaussian marginals. InInternational Conference on Machine Learning 2023 Jul 3 (pp. 7922-7938). PMLR.

---

### Official Review · Reviewer_Jem3 · 2025-07-05

**Clarity:** 3
**Significance:** 3
**Originality:** 3
**Rating:** 4
**Confidence:** 3

**Summary:**

This paper studies the problem of agnostically learning general (non-homogeneous) ReLU functions under the Gaussian distribution with respect to squared loss. While recent work has shown that, in the passive learning setting, one can learn with error O(opt) + ε using a polynomial number of labeled examples, this paper focuses on the interactive setting, where the learner can query labels of selected unlabeled examples.

The main contribution is a computationally efficient algorithm that achieves error O(opt) + ε using roughly d·polylog(1/ε) + O(min{1/p, 1/ε}) black-box label queries, where p is the bias of the target function.

**Questions:**

Can this technique be generalized for learning more than one neuron?

**Ethical Concerns:**

["NO or VERY MINOR ethics concerns only"]

**Quality:**

2

**Strengths And Weaknesses:**

The main strength of the paper is to show a sample complexity upper bound (Theorem 3.1) and an almost matching information theoretic lower bound (theorem 4.1). They also provide a separation between pool-based active learning and query learning in this setting.

---

> ### Author Rebuttal · Authors · 2025-07-31
>
> We thank the reviewer for their time and effort in providing feedback, and we are encouraged by the positive assessment. We next address the reviewer’s question.
>
>
> >Question: Can this technique be generalized for learning more than one neuron?
>
> The main contribution of this paper is to develop a computationally efficient, robust learning algorithm for learning general ReLU activations with optimal label complexity.
>
> The problem of learning a linear combination of multiple ReLUs with queries is also of interest and has been studied by some of the prior works, such as [MSDH19,CKM21,DG21] under certain additional assumptions---importantly, in the realizable (aka noiseless) setting. Roughly speaking, these works first use queries to find a point where only one neuron is active and use queries to recover that neuron. While there exist some potential connections between our work and these prior works (in particular, some of our techniques could possibly be leveraged to solve the problem with more than one neuron), we want to remark that the aforementioned algorithms heavily rely on the realizable assumption and are not even robust to Gaussian random noise. So, these works on learning a sum of ReLUs in the realizable setting are not comparable to our work. We believe that developing an efficient, robust learning algorithm for more than one neuron with queries is an interesting open question that would also require significant effort and new ideas.
>
> >[CKM21] Chen, Sitan, Adam R. Klivans, and Raghu Meka. "Efficiently learning any one hidden layer relu network from queries, 2021." URL https://arxiv. org/abs/2111.04727.
>
> >[DG21] Daniely A, Granot E. An exact poly-time membership-queries algorithm for extraction a three-layer relu network. arXiv preprint arXiv:2105.09673. 2021 May 20.
>
> >[MSDH19] Milli S, Schmidt L, Dragan AD, Hardt M. Model reconstruction from model explanations. InProceedings of the Conference on Fairness, Accountability, and Transparency 2019 Jan 29 (pp. 1-9).

---

### Decision · Program_Chairs · 2025-09-17

**Decision:**

Accept (poster)

**Comment:**

The paper presents query-based algorithms for learning ReLUs, and provide matching lower bounds. All reviewers acknowledge the technical novelty of this paper, and the intuition of the proof was explained in the rebuttal stage. The authors are encouraged to include the explanations in the final version.